# PRC2-EZH1 contributes to circadian gene expression by orchestrating chromatin states and RNA polymerase II complex stability

Peng Liu [1,10] ✉, Seba Nadeef [1,10], Maged F Serag[2], Andreu Paytuví-Gallart[3], Maram Abadi[2], Francesco Della Valle[1,4], Santiago Radío[3], Xènia Roda [3], Jaïr Dilmé Capó[3], Sabir Adroub[1], Nadine Hosny El Said [1], Bodor Fallatah[1], Mirko Celii[1], Gian Marco Messa[1], Mengge Wang[2], Mo Li [2], Paola Tognini [5,6], Lorena Aguilar-Arnal [5,7], Satoshi Habuchi[2], Selma Masri[8], Paolo Sassone-Corsi[5,9,11] & Valerio Orlando [1,11]✉

## Abstract

**Circadian rhythmicity of gene expression is a conserved feature of cell physiology. This involves fine-tuning between transcriptional and post-transcriptional mechanisms and strongly depends on the metabolic state of the cell. Together these processes guarantee an adaptive plasticity of tissue-specific genetic programs. However, it is unclear how the epigenome and RNA Pol II rhythmicity are integrated. Here we show that the PcG protein EZH1 has a gateway bridging function in postmitotic skeletal muscle cells. On the one hand, the circadian clock master regulator BMAL1 directly controls oscillatory behavior and periodic assembly of core components of the PRC2–EZH1 complex. On the other hand, EZH1 is essential for circadian gene expression at alternate Zeitgeber times, through stabilization of RNA Polymerase II preinitiation complexes, thereby controlling nascent transcription. Collectively, our data show that PRC2–EZH1 regulates circadian transcription both negatively and positively by modulating chromatin states and basal transcription complex stability.**

**Keywords** Circadian Rhythm; EZH1; H3K27me3; Transcription
**Subject Category** Chromatin, Transcription & Genomics

## Introduction

Circadian rhythms encompass a variety of behavioral, physiological, and metabolic functions in complex organisms. These rhythms are largely controlled by the circadian clock, an endogenous, time-keeping molecular machinery that regulates multiple metabolic and physiological functions (Reinke and Asher, 2019; Schibler and Sassone-Corsi, 2002). In mammals, the organization of the molecular circadian oscillator relies on well-defined transcriptional-translational feedback loops (TTFLs), composed of several clock genes (Takahashi, 2017). The transcriptional activators CLOCK and BMAL1 form a heterodimer and direct the expression of numerous clock-controlled genes (CCGs) through binding to their E-box motif. Among the CCGs, the repressors period (PER) and cryptochrome (CRY) are encoded and accumulate. Following their accumulation, PER and CRY proteins interact with each other and translocate into the nucleus, where they associate with the CLOCK: BMAL1 complex and suppress its transcriptional activity (Chiou et al, 2016; Kume et al, 1999; Sato et al, 2006). Two families of nuclear receptors, nuclear receptor subfamily 1 group D member 1/2 (NR1D1/2), also known as Rev-Erb alpha/beta (REV-ERBα/β), and RAR-related orphan receptors (RORs), are also direct targets of CLOCK: BMAL1. Those nuclear receptors function to stabilize the core loop through competitively binding to ROR/REV-ERB-response elements (RORE) and as a result antagonistically regulate *Bmal1* expression (Akashi and Takumi, 2005; Preitner et al, 2002; Sato et al, 2004). These interlocking transcriptional feedback loops work together to orchestrate circadian transcription (Patke et al, 2020b).

Circadian transcriptional programs are accompanied by dynamic changes in chromatin modifications, the epigenetic

[1]King Abdullah University of Science and Technology, KAUST, Biological and Environmental Sciences and Engineering Division, KAUST Environmental Epigenetics Program, Thuwal 23955-6900, Kingdom of Saudi Arabia. [2]King Abdullah University of Science and Technology, KAUST, Biological and Environmental Sciences and Engineering Division, Bioscience Program, Thuwal 23955-6900, Kingdom of Saudi Arabia. [3]Sequentia Biotech SL, Carrer Comte D'Urgell 240, Barcelona 08036, Spain. [4]Altos Labs, Institute of Science, San Diego, CA 92121, USA. [5]University of California, Irvine, Department of Biological Chemistry, School of Medicine, Center for Epigenetics and Metabolism, Irvine, CA 92697, USA. [6]Health Science Interdisciplinary Center, Scuola Superiore Sant'Anna, Pisa 56126, Italy. [7]Universidad Nacional Autónoma de México, Instituto de Investigaciones Biomédicas, Mexico City 04510, Mexico. [8]University of California Irvine, Department of Biological Chemistry, Chao Family Comprehensive Cancer Center, Irvine, CA 92697, USA. [9]Deceased: Paolo Sassone-Corsi. [10]These authors contributed equally: Peng Liu, Seba Nadeef. [11]These authors contributed equally as senior authors: Paolo Sassone-Corsi, Valerio Orlando. ✉E-mail: peng.liu@kaust.edu.sa; valerio.orlando@kaust.edu.sa

landscape (Etchegaray et al, 2003; Katada and Sassone-Corsi, 2010; Koike et al, 2012; Valekunja et al, 2013; Vollmers et al, 2012), and RNA Polymerase II (RNAPII) occupancy (Koike et al, 2012; Le Martelot et al, 2012). Moreover, clock machinery associates with diverse chromatin modifiers and remodelers which govern the circadian chromatin epigenome landscape to support rhythmic transcription (DiTacchio et al, 2011; Etchegaray et al, 2003; Kim et al, 2014a; Nakahata et al, 2008; Nam et al, 2014; You et al, 2013). Although the temporal coordination between RNAPII and multiple activation histone marks like H3K4me3, H3K27ac, and H3K36me3 have been extensively studied (Koike et al, 2012), contributions from repressive histone modifications H3K27me3 and PRC2 in the regulation of circadian transcription has not been fully explored. Polycomb group (PcG) proteins are a family of proteins that maintain cell identity during development by conveying a repressed state of target genes through histone H3K27me3 (Blackledge and Klose, 2021; Kim and Kingston, 2022; Lanzuolo and Orlando, 2012; Schuettengruber et al, 2017; Yu et al, 2019). Polycomb repressive complex 2 (PRC2) contains H3K27me3-specific histone methyl transferase (HMT) activity and is present in mammals in two forms: PRC2-EZH2, which is prevalent in cycling cells, and PRC2–EZH1, which replaces EZH2 in somatic postmitotic cells (Margueron et al, 2008; Shen et al, 2008). At the transcriptional level, *Ezh2* and *Ezh1* are oppositely expressed during development; *Ezh2* is highly expressed in proliferating cells but barely detectable in adult tissues, whereas *Ezh1* is expressed mainly in postmitotic tissues (Caretti et al, 2004; Ezhkova et al, 2009; Margueron et al, 2008; Shen et al, 2008; Su et al, 2005). A role for EZH2 in regulating the mammalian molecular clock was reported in different contexts (Etchegaray et al, 2006; Zhong et al, 2018). An additional report revealed that the CLOCK–BMAL1 complex is required for EZH2 recruitment, leading to the trimethylation of H3K27 at targeted genes of chemokine expression in monocytes, thereby inhibiting their transcription (Nguyen et al, 2013). However, the overall effects on gene expression remained contradictory, suggesting the intricate nature of PRC2 and circadian transcription, especially in postmitotic tissues (Etchegaray et al, 2006; Zhong et al, 2018).

The regulation and function of EZH1 are distinct from that of EZH2. We previously reported a novel mechanism in which a cytoplasmic short isoform of EZH1 (EZH1β) modulates shuttling of Embryonic Ectoderm Development (EED) into the nucleus, facilitating adaptive and reversible PRC2–EZH1α complex assembly and H3K27me3 deposition in response to oxidative stress (Bodega et al, 2017). Moreover, EZH1 was also reported to exert activating functions during the differentiation of skeletal muscle cells, either through mediating RNA Pol II elongation or MyoD recruitment (Mousavi et al, 2012; Stojic et al, 2011). The activation mechanism of EZH1 potentially involves an EZH1-SUZ12 subcomplex that does not contain EED. This subcomplex has the ability to bind to active chromatin and positively regulate transcription (Su et al, 2016; Xu et al, 2015). EZH1's activation is implicated in the initial release of RNAPII pausing, instead of H3K27me3 removal, which activates bivalent genes (Chen et al, 2020). In addition, EZH1 plays a role in maintaining gene expression of the Notch signaling pathway in muscle stem cells (Feng et al, 2023). Collectively, these studies confirm a primary positive regulatory role of EZH1 in transcriptional regulation. However, the mechanistic impact on RNA Pol II and the coordination with canonical silencing function in homeostatic conditions remain to be elucidated.

Research on circadian rhythm in skeletal muscle has revealed that the circadian clock influences various aspects of skeletal muscle physiology, metabolism, and gene expression (Dyar et al, 2014; Dyar et al, 2018; Jordan et al, 2017; Perrin et al, 2018). However, the question remains regarding whether and how the rhythmic activity of RNA Pol II in skeletal muscle is regulated. In this study, we present evidence supporting the involvement of PRC2–EZH1 as a novel regulator of the intrinsic clock. Our findings reveal that EZH1 plays a dual role in chromatin modification and transcription machinery, being cyclically associated with both PRC2 components and RNA Pol II machinery. Interestingly, the two interactions exhibited distinct rhythmic patterns with different peak phases. Furthermore, we identified that the shuttling of EED is synchronized with the assembly of the PRC2–EZH1 complex and the deposition of H3K27me3 in a circadian pattern. In addition, we discovered that EZH1 is a crucial positive player in maintaining the integrity of the RNA Pol II preinitiation complex. Taken together, this study sheds light on the intricate orchestration of transcriptional rhythmicity between epigenome dynamics and RNA Pol II intrinsic regulation, revealing EZH1 as a key player in the formation of respective complexes at different phases, thereby contributing to circadian gene expression regulation.

## Results

### EZH1 expression is rhythmic in skeletal muscle tissue and synchronized C2C12 myotubes

In order to investigate whether EZH1 might be involved in skeletal muscle peripheral clock regulation, we first examined the temporal expression of *Ezh1* in mouse skeletal muscle tissue. Mice were sacrificed every 4 h over a 24-h period, which was followed by RNA and protein extraction. As shown, both *Ezh1α* and *Ezh1β* exhibited significant oscillations, along with circadian transcriptional regulators such as *Bmal1* and *Per1* (Fig. 1A) and other clock genes such as *Per2*, *Cry1, Dbp*, and *Rev-erbα* (Appendix Fig. S1A). Interestingly, *Ezh1α* and *Ezh1β* expression follows a similar circadian pattern to *Per1* and is in anti-phase with *Bmal1* (Fig. 1A). At the protein level, EZH1α accumulated in a circadian pattern, with maximal and minimal levels phase around *zeitgeber* time (ZT) ZT4 and ZT12, respectively. Intriguingly, EZH1β protein exhibited a 12-h rhythm, which has been previously involved in coordinating metabolic and stress rhythms (Zhu et al, 2017). In addition, circadian changes in BMAL1 and PER1 protein levels were uncovered in skeletal muscle (Fig. 1B,C). The discrepancy of rhythmicity between *Ezh1* transcripts and EZH1 proteins might implicate additional post-transcriptional and post-translational mechanisms. In addition to skeletal muscle tissue, we also utilized horse serum-synchronized mouse skeletal muscle C2C12 cells (Peek et al, 2015) to examine oscillation of *Ezh1*. Synchronized cells were collected at the late myotube stage after changing the differentiation medium for 4 days, a more physiological stage recapitulating features of intact muscle (Yaffe and Saxel, 1977). Successful synchronization of C2C12 was validated by measuring *Bmal1* and *Per1* circadian transcription levels (Fig. 1D) together with other core clock genes (Appendix Fig. S1b). Consistent with the findings in skeletal muscle tissue, EZH1 displayed a significant circadian pattern at the mRNA and protein level in

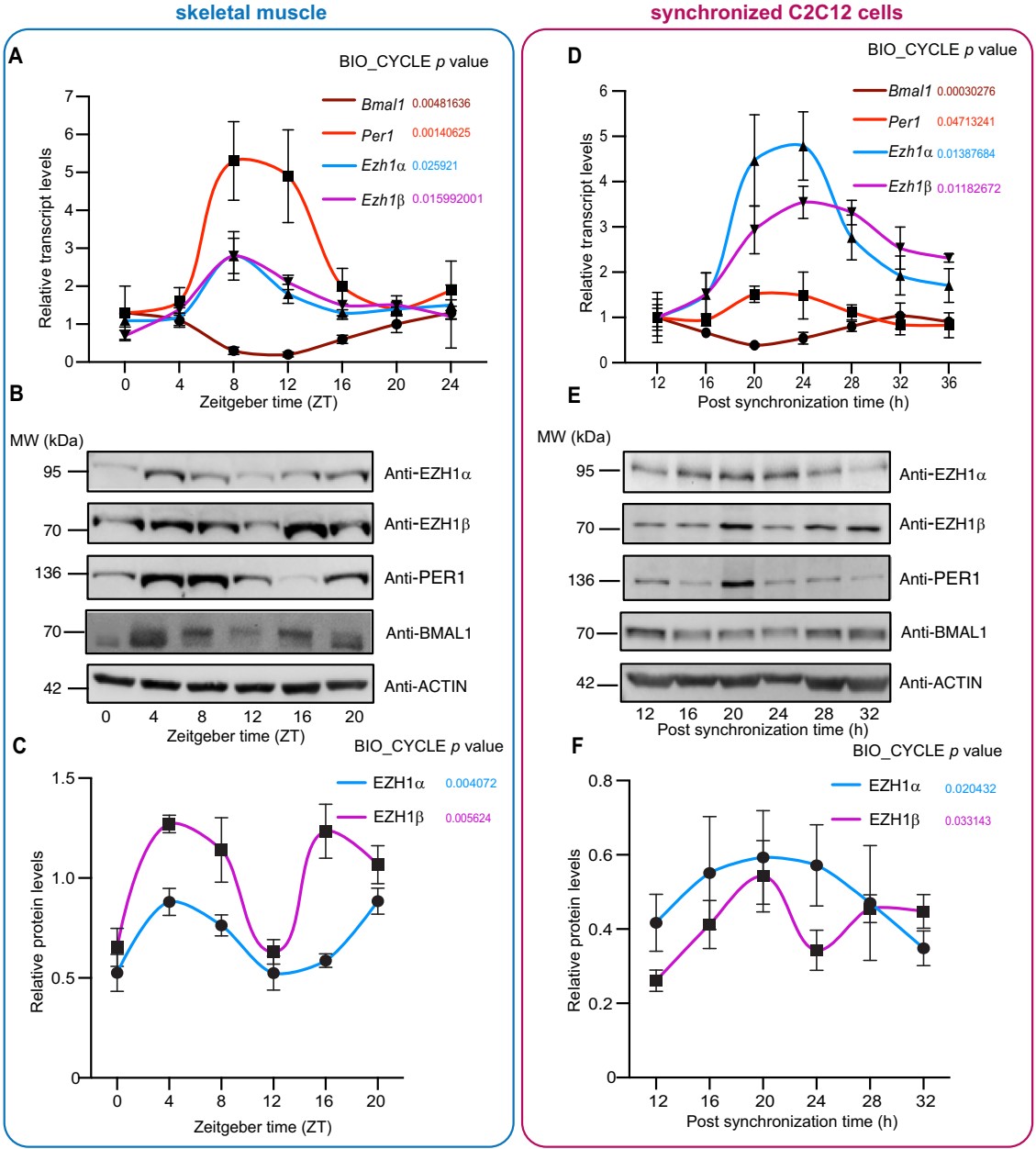

**Figure 1. Circadian profile of both *Ezh1* and EZH1 in skeletal muscle tissue and synchronized C2C12.**

(A) Mice were sacrificed at 4-hour intervals. Circadian expression of *Ezh1α* and *Ezh1β* mRNA along with clock genes *Bmal1* and *Per1* from mice gastrocnemius muscle tissue was analyzed by qPCR using specific oligos ($n = 5$ per time point). Error bars represent ± SEM from five independent experiments. (B) Rhythmic protein levels of EZH1α, EZH1β, BMAL1, and PER1 in mouse gastrocnemius muscle tissue was analyzed by immunoblotting. Three independent experiments were performed, and representative data was shown. ACTIN was used as an internal control. (C) Quantitative analysis of protein abundance of EZH1α and EZH1β shown in (B), respectively, in skeletal muscle tissue. Error bars represent ± SEM from three independent experiments. (D) C2C12 myotube cells at day 4 were synchronized with 50% horse serum and collected every 4 h. Rhythmic expression of *Ezh1α*, *Ezh1β*, *Bmal1*, and *Per1* transcripts from synchronized C2C12 was analyzed by quantitative real-time PCR using specific oligos ($n = 3$ per time point). Error bars represent ± SEM from three independent experiments. (E) Rhythmic protein expression of EZH1α, EZH1β, BMAL1, and PER1 from synchronized C2C12 at indicated time points were analyzed by immunoblotting. Three independent experiments were performed, and representative data was shown. ACTIN was used as an internal control. (F) Quantification of protein abundance of EZH1α, EZH1β, and PER1 shown in (E), respectively. Error bars represent ± SEM from three independent experiments. Circadian $P$ values indicated from (A) to (F) represent rhythmic transcripts or proteins using the nonparametric test Bio-cycle, where $P < 0.05$ is considered statistically significant, incorporating a window of 24 h (muscle tissue) and 32 h (C2C12). Source data are available online for this figure.

synchronized cells (Fig. 1D–F). Finally, we measured H3K27me3 at various time points. Interestingly, we did not observe any significant circadian variations in H3K27m3 global levels, either skeletal muscle or synchronized C2C12 myotube cells. (Appendix Fig. S1C–F). Taken together, our results demonstrate that *Ezh1* displays an oscillatory profile in anti-phase with *Bmal1*, both in vivo and in vitro, suggesting a potential role for EZH1 in muscle clock regulation.

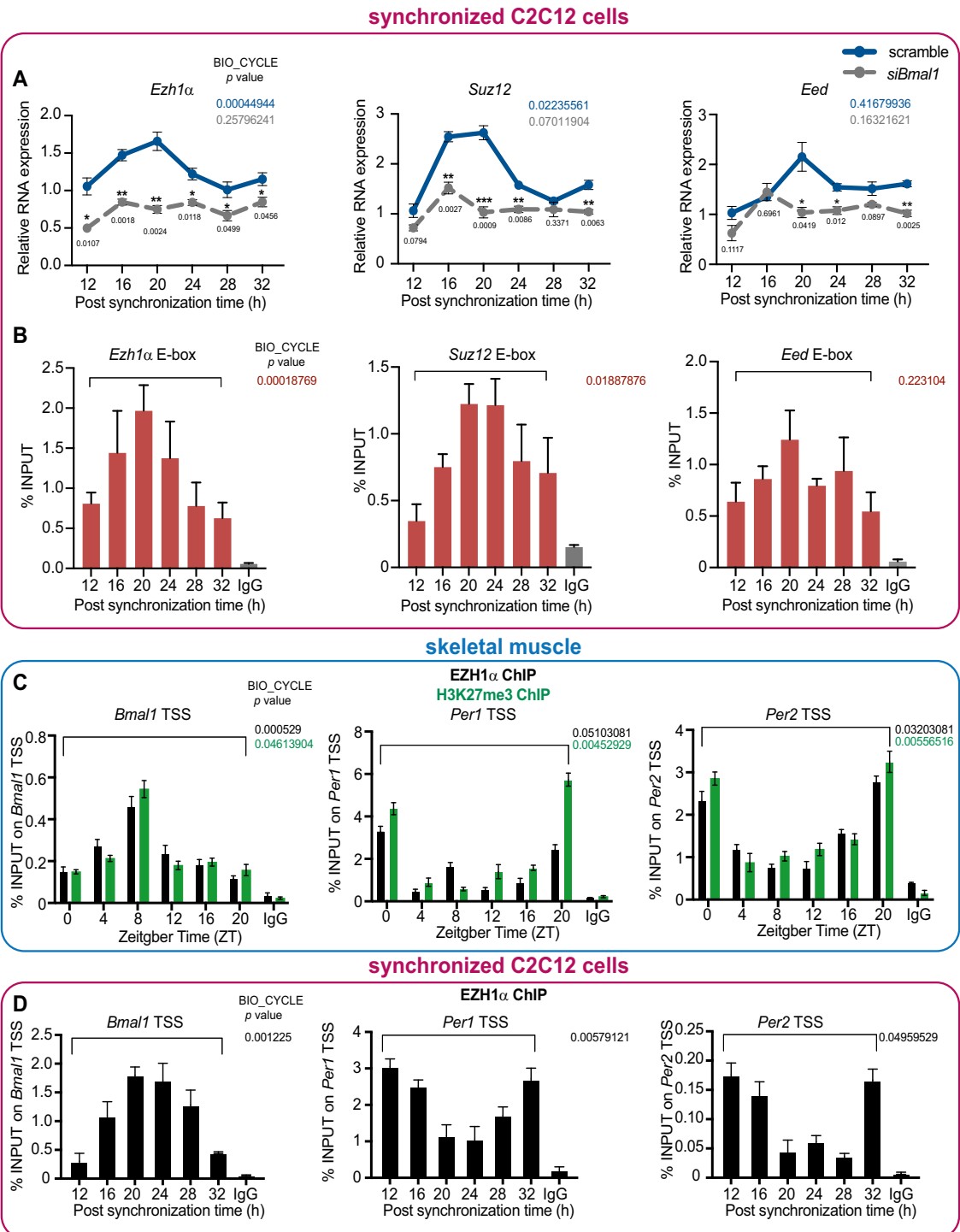

## Reciprocal regulation between PcG complex and core clock machinery

As previously reported (Bass and Lazar, 2016), the clock components CLOCK and BMAL1 are known to be involved in regulating rhythmic gene expression. We thus wondered whether *Ezh1* circadian oscillation is BMAL1-dependent. We investigated the oscillatory profile of *Ezh1α* in *Bmal1*-depleted C2C12 cells

treated by different siRNAs targeting *Bmal1* (Appendix Fig. S2A,B). Similarly, to what we saw in wild-type C2C12, *Ezh1α* showed a significant oscillation profile; however, both circadian phase pattern and amplitude were significantly disturbed in the absence of *Bmal1* (Fig. 2A; Appendix Fig. S2C–E). Not only *Ezh1*, rhythmicity and amplitude of *Suz12* and *Eed*, another two core components of PRC2–EZH1 complex, were also compromised due to *Bmal1* deficiency (Fig. 2A). To explore whether BMAL1-dependent *Ezh1*

◄ **Figure 2. Reciprocal regulation between PRC2–EZH1 complex and core clock components.**

(A) Oscillatory profile of *Ezh1α*, *Suz12*, and *Eed* transcripts in scramble and *Bmal1* siRNA (#1) knockdown C2C12 was analyzed by RT-qPCR using specific oligos (*n* = 3 per time point). Error bars represent ± SEM from three independent experiments. \**P* < 0.05; \*\**P* < 0.01; \*\*\**P* < 0.001; by two-tailed *t* test. (B) ChIP of BMAL1 at predictive E-box region around *Ezh1*, *Suz12*, and *Eed*. ChIP was performed using cross-linked chromatin from synchronized C2C12 myotube cells at day 4and quantified by qPCR (*n* = 3 per time point). IgG represents ChIP experiment performed with an isotype-matched control immunoglobulin (normal rabbit IgG) to BMAL1. (C) ChIP experiment of EZH1α and H3K27me3 at transcription start site (TSS) of *Bmal1*(left panel), *Per1* (middle panel), and *Per2* (right panel) was performed using cross-linked chromatin from gastrocnemius muscle tissue. Immunoprecipitated DNA was quantified by qPCR at the indicated zeitgeber times (ZT, hours after lights on) (*n* = 3 per time point). (D) ChIP experiment of EZH1α at transcription start site (TSS) of *Bmal1* (left panel), *Per1* (middle panel), and *Per2* (right panel) was performed using cross-linked chromatin from wild-type synchronized C2C12 myotubes. Immunoprecipitated chromatin was quantified by qPCR at the indicated different time points after horse serum shock (*n* = 3 per time point). Error bars represent ± SEM from three independent experiments. Circadian *P* values within each panel indicate statistical significance of rhythmicity of transcripts expression or occupancy profile using the nonparametric test Bio-cycle, where *P* < 0.05 consider statistically significant, incorporating a window of 24 h (muscle tissue) and 32 h (C2C12). Source data are available online for this figure.

circadian oscillation is conserved in other cell-autonomous systems, a similar experiment was performed using Mouse Embryonic Fibroblasts (MEFs) derived from *Bmal1*(−/−) and *Clock*(−/−) deficient mice. Interestingly, the cyclic transcription profile of *Ezh1α* was also significantly compromised in synchronized *Clock*(−/−) and *Bmal1*(−/−) MEF cells alongside different circadian time points (Appendix Fig. S2F,G). These findings collectively demonstrate the ubiquitous role of BMAL1 as a positive regulator for the circadian expression of PRC2–EZH1 core components.

Then, we sought to investigate whether BMAL1 regulates circadian oscillation of *Ezh1α* through direct association with the E-box of PRC2 components. Chromatin immunoprecipitation (ChIP) was performed using BMAL1 antibody to assess its occupancy on E-boxes, which were predicted using Genomatix software suite V3.10 Matlnspector, around transcriptional start site (TSS) of PRC2–EZH1 complex components: *Ezh1α*, *SUZ12*, and *EED*. ChIP-qPCR results showed significant circadian binding of BMAL1 at E-boxes of *Ezh1* and *Suz12*, which could explain their rhythmic transcriptional pattern (Fig. 2B). Collectively, we conclude that BMAL1 positively and directly modulates the oscillatory expression profile of core components of PRC2–EZH1 in mouse skeletal muscle tissue and in synchronized C2C12 cells.

Given that the peak of EZH1 protein levels coincides with the lowest *Bmal1* mRNA levels (ZT8) (Fig. 1B,C), we hypothesized a role of EZH1 in the regulation of core clock transcript expression. In particular, we tested whether EZH1α, the nuclear isoform associated with chromatin could directly bind the promoter region of core clock genes. Chromatin immunoprecipitation (ChIP) of EZH1α was performed using skeletal muscle tissue alongside the circadian cycle. Precisely, EZH1α displayed significant circadian enrichment at the TSS regions of *Bmal1*, *Per1*, and *Per2* (Fig. 2C). However, no significant circadian occupancy of EZH1 around TSS regions of *Cry1*, *Dbp* and *Rev-erbα* (Appendix Fig. S3A–C) was found. These data suggest functional and target specificity in terms of circadian occupancy of EZH1α. In addition, we performed ChIP-qPCR assay to measure EZH1α oscillational binding around genomic loci of *Bmal1*, *Per1*, and *Per2* in synchronized C2C12 cells. Similar findings were observed and demonstrated that EZH1α exhibits oscillatory occupancy around TSS region of *Bmal1*, *Per1*, and *Per2* in cell-autonomous system (Fig. 2D). In order to validate the specificity of EZH1 binding, we assessed the EZH1α intensity surrounding the transcription start site (TSS) loci of *Hoxd10* and *Neurog1*, which are well-established canonical PcG targets. We discovered strong and specific enrichment of EZH1α around

*Hoxd10* and *Neurog1* loci. However, we did not observe any significant circadian pattern around the TSS regions of *Hoxd10* and *Neurog1* in conjunction with *Rev-erbα*. (Appendix Fig. S3D–F). Taken together, these results indicate a strong association between EZH1α and clock regulation, possibly mediated by rhythmic H3K27me3 modification status. Indeed, H3K27me3 ChIP-qPCR assay demonstrated that oscillatory occupancy of EZH1 exhibits high correlation with H3K27me3 circadian deposition profile around the same genomic loci (Fig. 2C).

## Integrated mode of EZH1α in the regulation of circadian transcription

To further investigate whether circadian enrichment of EZH1α on the promoters of core clock genes is required for cyclic H3K27me3 deposition levels on core clock genomic loci, stable C2C12 cell lines (*shEzh1*) with impaired EZH1 function were constructed using specific shRNA targeting the region shared with both *Ezh1α* and *Ezh1β*. To exclude off-target effects of shRNA technology, we further generated another cell line (*shEzh1*^EZH1α) through ectopic expression of *Ezh1* siRNA-resistant EZH1α in the background of *shEzh1* knockdown cell line. Another study has employed a similar strategy and demonstrated that this rescue approach effectively eliminates the off-target effects associated with shRNA technology (Vo et al, 2018). Knockdown efficiency was confirmed by measuring *Ezh1* transcript and EZH1 protein level in wild-type (WT), knockdown (*shEzh1*) and *shEzh1*^EZH1α genetic backgrounds (Appendix Fig. S4A,B). As the core catalytic enzyme of PRC2–EZH1 complex, we firstly checked EZH1 mediated H3K27me3 levels around *Bmal1*, *Per1*, and *Per2* genomic loci during the circadian cycle. The rhythmic pattern of H3K27me3 deposition around those loci in WT background was significantly diminished upon *Ezh1* depletion, which further was partially recovered in the *shEzh1*^EZH1α condition (Fig. 3A). Given the well-established concept that H3K27me3 is a repressive histone mark for transcription (Yu et al, 2019), we then tried to determine to what extent EZH1 can affect the circadian expression of *Bmal1*, *Per1*, and *Per2*. Surprisingly, compared with the WT background, expression of *Bmal1*, *Per1*, and *Per2* was highly reduced upon the depletion of *Ezh1*, and largely restored in *shEzh1*^EZH1α condition (Fig. 3B). H3K27me3 reduction status in *Ezh1* knockdown background (Fig. 3A) could not explain the reduced expression pattern (Fig. 3B). EZH1 could instead impact RNA Pol II function as a positive regulator (Mousavi et al, 2012). Indeed, RNA Pol II recruitment and activity exhibit time-dependent regulation to accompany

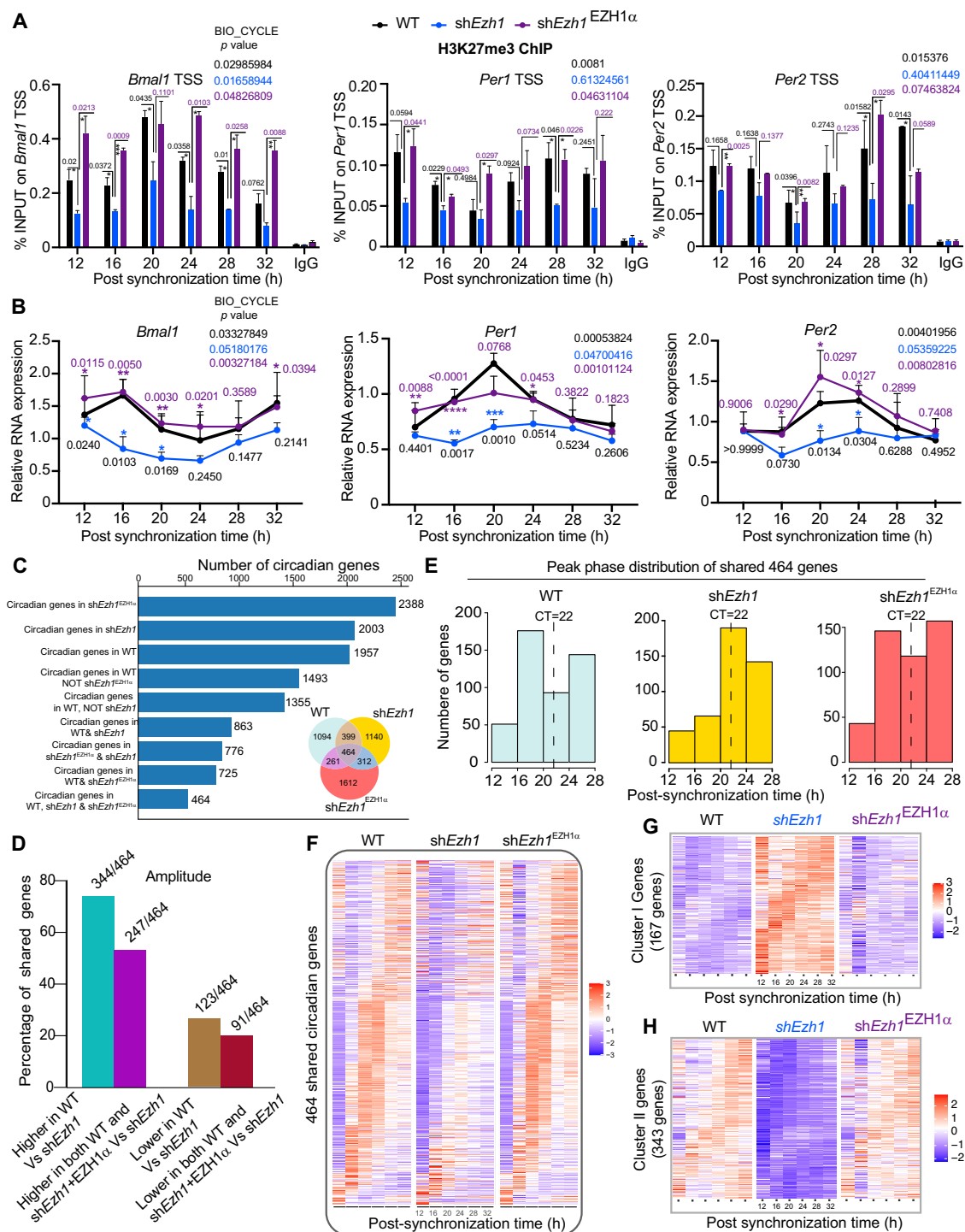

genome-wide circadian gene transcription (Koike et al, 2012; Le Martelot et al, 2012). Thus, we examined whether the dampened expression of core clock gene expression in the *Ezh1* knockdown background was due to the compromised RNA Pol II temporal dynamics. As described, serine 5 phosphorylation (S5P) and serine 2 phosphorylation (S2P) of the C-terminal domain (CTD) of RNA Pol II are recognized as the initiation and elongation forms of RNA Pol II, respectively (Mousavi et al, 2012). We asked whether EZH1

has a role in the modulation of RNA Pol II occupancy profile around core clock genes loci. We then performed ChIP-qPCR with RNA Pol II antibodies specific for S5P and S2P to check the RNA Pol II initiation and elongation forms and circadian status around *Bmal1*, *Per1*, and *Per2* genomic loci during the circadian cycle. ChIP-qPCR assay showed that both RNA Pol II S5P and S2P forms exhibit oscillatory profile at the TSS and gene body region of *Bmal1*, *Per1*, and *Per2* loci. In the absence of EZH1, the circadian

Figure 3. Integrated mode of EZH1α in the regulation of circadian transcription at a genome-wide scale.

(A) ChIP experiment of H3K27me3 at transcription start site (TSS) of *Bmal1*(left panel), *Per1* (middle panel), and *Per2* (right panel) was performed using cross-linked chromatin from wild-type (WT), sh*Ezh1* and sh*Ezh1*$^{EZH1\alpha}$ synchronized C2C12 myotubes at day 4. Immunoprecipitated DNA was quantified by qPCR at the indicated different time points after 50% horse serum synchronization ($n = 3$ per time point). Error bars represent ± SEM from three independent experiments. IgG represents ChIP experiment performed with an isotype-matched control immunoglobulin (normal rabbit IgG) to EZH1α or H3K27me3. *$P < 0.05$; **$P < 0.01$; ***$P < 0.001$; by two-tailed $t$ test. (B) Expression of core clock component genes *Bmal1* (left panel), *Per1* (middle panel), and *Per2* (right panel) were measured by quantitative real-time PCR using specific primers ($n = 3$ independent biological experiments per time point) in synchronized WT, sh*EZH1* and sh*EZH1*$^{Ezh1a}$ cells. Error bars represent ± SEM from three independent experiments. *$P < 0.05$; **$P < 0.01$; ***$P < 0.001$; by two-tailed $t$ test. Circadian $P$ values within (A, B) Represent rhythmic transcripts using the nonparametric test Bio-cycle, where $P < 0.05$ consider statistically significant, incorporating a window of 32 h (C2C12). (C) Venn diagram represents circadian transcripts in WT, sh*Ezh1*, and sh*Ezh1*$^{EZH1\alpha}$ conditions. Bar plot shows the number of circadian genes within specifically defined condition. (D–F) Amplitude (D), peak phase (E), and heatmap (F) analyses of shared transcripts with the oscillatory profile in WT, sh*Ezh1*, and sh*Ezh1*$^{EZH1\alpha}$ conditions. (G, H) Heatmap of circadian transcripts under WT condition, disturbed in sh*Ezh1* and partially restored in sh*Ezh1*$^{EZH1\alpha}$ condition ($n = 3$, JTK_cycle, $P < 0.05$). Cluster I and II genes correspond to circadian genes in the wild-type (WT) samples, exhibiting either upregulation (G) or downregulation (H) in expression across all time points when compared to both WT and sh*Ezh1* samples. The statistical analysis in (G, H) were performed using DESeq2 as described in the "Methods" section. Source data are available online for this figure.

occupancy profile of both forms was impaired in *shEzh1* and re-established in *Ezh1* rescue experiments (Appendix Fig. S4C,D). Taken together, both H3K27me3 circadian deposition pattern and oscillatory occupancy of RNA Pol II were dampened in *Ezh1* knockdown condition, indicating the integrated nature of regulation of H3K27me3 status and RNA Pol II processivity, where temporal RNA Pol II dynamics appears to play a specific and more relevant role in orchestrating circadian transcriptional program of *Bmal1*, *Per1*, and *Per2* genes. Earlier study has revealed the simultaneous presence of H3K4me3 and EZH1α (Mousavi et al, 2012). Thus, we also checked the state of H3K4me3 around the same loci of *Bmal1*, *Per1*, and *Per2* genes. Through ChIP-qPCR analysis, we observed an increase in the H3K4me3 signal upon EZH1 depletion. This finding suggests a coordinated interplay between H3K27me3 and H3K4me3 in the regulation of expression of *Bmal1*, *Per1*, and *Per2* (Appendix Fig. S5).

To verify the integrated function of EZH1α-dependent transcriptional rhythmicity at the genome-wide scale, total RNAs from WT, *shEzh1*, and *shEzh1*$^{EZH1\alpha}$ backgrounds were extracted for transcriptome analyses during a circadian cycle. Transcripts showing circadian rhythmicity were identified using the JTK_CYCLE algorithm for the determination of diurnal periodicity (Hughes et al, 2010). Out of total 2218 circadian transcripts identified in WT cells; rhythms of 1355 transcripts were impaired in the *shEzh1* condition (Fig. 3C). Intriguingly, expression of siRNA-resistant EZH1α in the *Ezh1* knockdown cell line could restore the circadian rhythm of 261 targets out of 1355 genes, hinting the importance of relative temporal stoichiometry of EZH1 pacing circadian rhythm of cyclic genes (Fig. 3C) (Bodega et al, 2017; Xu et al, 2015). We detected 464 genes exhibiting circadian expression among all three conditions. Amplitude analysis of common 464 circadian genes across all conditions showed that EZH1 positively impacts the amplitude of 344 (74%) common oscillating genes in synchronized C2C12 WT cells (Fig. 3D). Conversely, the amplitude of 123 (26%) common circadian genes was negatively regulated by EZH1 (Fig. 3D). Of note, both increased (247 out of 344) and decreased (91/123) amplitude of common circadian genes due to depletion of *Ezh1* could be efficiently restored when EZH1α was introduced back into sh*Ezh1* knockdown background (Fig. 3D). Moreover, peak phase and heatmap analyses of common circadian genes revealed similar peak phase distribution in both WT and *shEzh1*$^{EZH1\alpha}$ (Fig. 3E,F), while in sh*Ezh1* knockdown, peak phase of common circadian genes

redistributed with the main peak around 22 h after synchronization (Fig. 3E,F). Concomitantly, the peak phase and rhythmicity of unique oscillating genes in specific conditions exhibit unique profiles (Appendix Fig. S6). To further assess the function of EZH1 in the modulation of expression of oscillating genes as an activator or repressor, we identified two different sets of circadian genes whose expression was either upregulated or downregulated at all time points in the knockdown condition. We defined classes of upregulated or downregulated transcripts as cluster I and cluster II gene set, respectively (Fig. 3G,H; Dataset EV2). Cluster I gene set contains 167 genes, which correlate with the repressive function of EZH1 (Fig. 3G), and a total of 313 genes in cluster II gene set, indicating activation function of EZH1 (Fig. 3H). Collectively, these data demonstrate the integrated role of EZH1 in regulation of circadian gene expression at a genome-wide scale, particularly through modulating the amplitude of circadian genes.

## Repressive role of EZH1α in the regulation of circadian transcription requires temporal assembly of PRC2–EZH1 complex

The cluster I gene set, identified through total RNA-seq analysis, suggests a potential repressive function of EZH1α (Fig. 3G). Stoichiometry of Polycomb complex components is crucial for PcG repressive function (Bodega et al, 2017). Thus, we hypothesized that the assembly of PRC2–EZH1 components may also undergo rhythmic regulation during circadian cycle. To assess the rhythmic interaction pattern among PRC2–EZH1 components, we performed immunoprecipitation coupled with quantitative mass spectrometry analyses (IP-qMS) using C2C12 stable cell line expressing EZH1α-Flag-HA chimeric proteins (Liu et al, 2019). Our IP-qMS assay identified all previously well-defined PRC2–EZH1 components (Xu et al, 2015), including SUZ12, EED, JARID2 etc. In support of our hypothesis, both core components and accessory proteins were found to be associated with EZH1α in a rhythmic pattern (Fig. 4A–C; Dataset EV3). More intriguingly, this interaction displayed the highest peak at 24–28 h period, where protein levels of EZH1α also reached the highest peak (Fig. 1E,F). Those data hinted that more complicated signals and mechanisms are required to maintain this temporal assembly behavior, possibly through the previously described Ezh1β dependent EED cytoplasm to nucleus shuttling (Bodega et al, 2017). Therefore, we examined dynamic EED association with

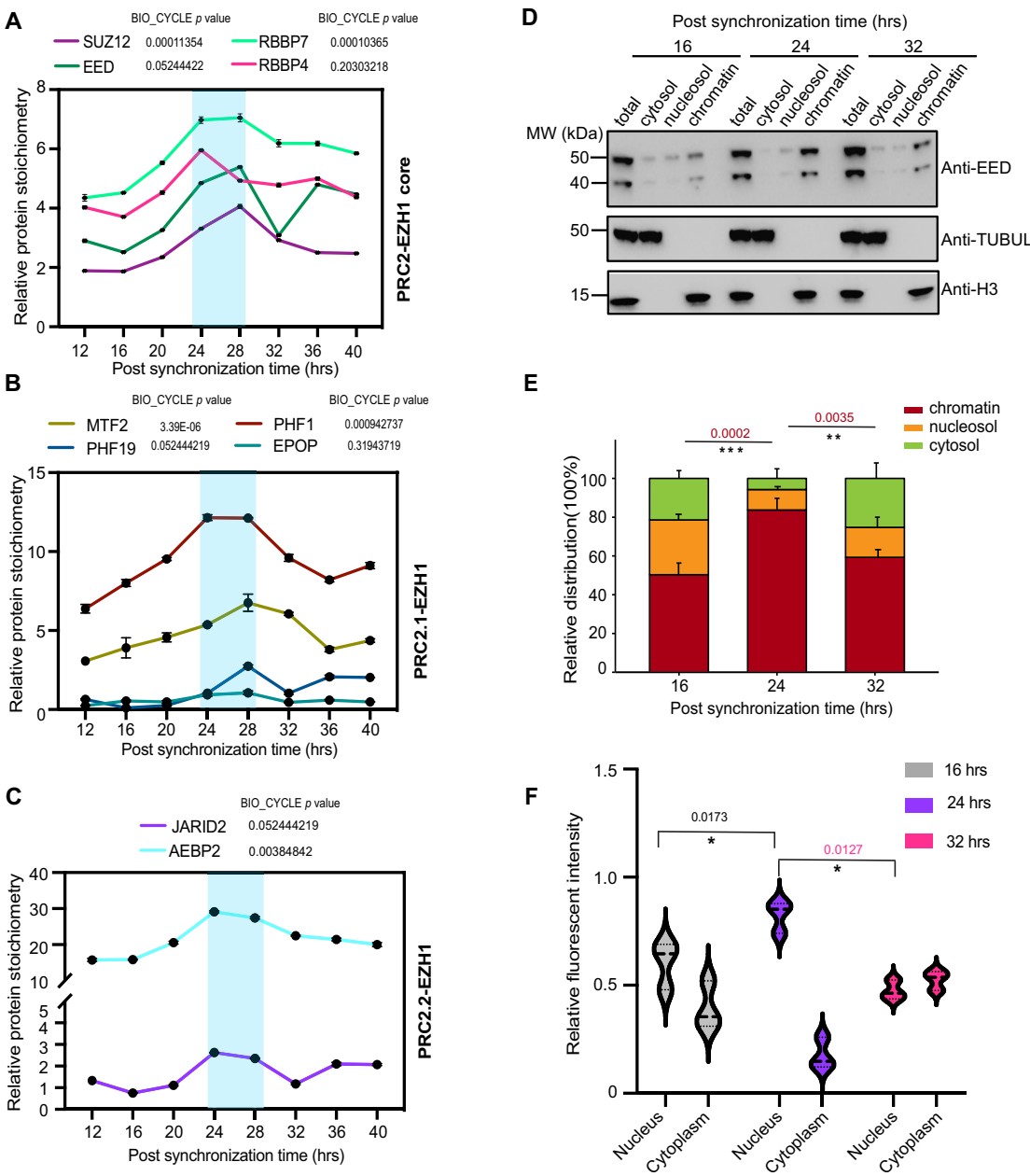

**Figure 4. Shuttling of EED facilitates PRC2‒EZH1α complex assembly.**

(**A**) Immunoprecipitation coupled with label-free quantitative mass spectrometry analysis indicating circadian interaction between EZH1α and core polycomb components: SUZ12, EED, RBBP4/7. (**B**, **C**) Immunoprecipitation coupled with label-free quantitative mass spectrometry analysis indicating circadian interaction between EZH1α and two major polycomb subcomplexes: PRC2.1-EZH1 (**B**) and PRC2.2-EZH1 (**C**), which contains distinct subunits at indicated circadian time points ($n = 3$ independent biological experiments). Relative stoichiometry intensity of EZH1α-HA from each time point has been normalized. Error bars represent ± SEM from three independent experiments. (**D**) Representative immunoblot analysis of EED distribution in the cytosol, nucleosol and chromatin-bound fractions at indicated circadian time points in wild-type C2C12 myotube cells at day 4. Tubulin or H3 was used as cytosol and chromatin fraction control, respectively. (**E**) Quantitative analyses of EED distribution as indicated condition in (**D**). ($n = 3$). Error bars represent ± SEM from three independent experiments. $P$ value indicates the relative difference between chromatin-bound EED intensity. **$P < 0.01$; ***$P < 0.001$; by two-tailed $t$ test. (**F**) Relative fluorescent intensity of EED inside nucleus and cytoplasm in myotube was calculated respectively. The Myotube region was defined through the MyHC staining signal. All quantitative analyses were analyzed using ImageJ software. A minimum of 100 nuclei inside the myotube were counted for statistical analysis. *$P < 0.05$; by two-tailed $t$ test. Error bars represent ± SEM from three independent experiments. Circadian $P$ values within each panel indicate statistical significance of rhythmicity of transcripts expression or occupancy profile using the nonparametric test Bio-cycle, where $P < 0.05$ consider statistically significant, incorporating a window of 32 h (C2C12). Source data are available online for this figure.

chromatin fraction at different circadian time points using cell fractionation assay (Fig. 4D). By analyzing chromatin fraction at different time points, we found that the majority of EED bound aligns with the maximum stoichiometry value of PRC2–EZH1 complex (Fig. 4D,E). These data suggest that the distribution rate of EED between cytosol and chromatin fraction follows a dynamic temporal pattern. Interestingly, the association of SUZ12 with chromatin fraction exhibits no significant time-dependent pattern (Appendix Fig. S7). We further utilized immunostaining approach for quantitative measurement of EED distribution at different fractions during the circadian cycle. Consistent with cell fractionation assay, our immunostaining-based approach proved that EED levels in in postmitotic myotube nuclei reach the highest peak at the same time point of the maximum PRC2–EZH1 stoichiometry (Fig. 4F; Appendix Fig. S7). Hence, time-dependent shuttling of EED components may facilitate cyclic assembly of PcG complex in order to orchestrate rhythmicity of H3K27me3 deposition around Polycomb direct targets and drive their circadian expression pattern in postmitotic skeletal muscle.

## EZH1 is crucial to maintain the rhythmicity of nascent transcripts

In addition to EZH1 function in repression of circadian genes, cluster II genes reflect the positive function of EZH1 in the regulation of another set of circadian genes globally (Fig. 3H). Previous reports indicated the interaction between EZH1 and RNA Pol II (Mousavi et al, 2012; Stojic et al, 2011). Here, through immunoprecipitation coupled with quantitative mass spectrometry (IP-qMS) analysis associated with EZH1α we also found members of the mediator complex and several other RNA Pol II machinery subunits (Appendix Fig. S8). Intriguingly, we found that the strongest interaction between RNA Pol II machinery-associated proteins and EZH1 occurs at around the time between 16 h and 20 h, which is distinct from PRC2–EZH1 assembly peak phase around 24 h (Appendix Fig. S8).

To further investigate the impact of EZH1α on RNA Pol II activity, we utilized TT-seq (Schwalb et al, 2016) analysis to track alterations in nascent RNA synthesis following *Ezh1* depletion. This analysis was conducted at three distinct time points after synchronization, including the peak time point at 16 h, which corresponds to the strongest interaction between EZH1 and the RNA Pol II machinery, as well as the canonical assembly of the PRC2–EZH1 complex at 24 h (Fig. 5A). More than eighty percent of uniquely mapped TT-seq reads are mapped to intronic regions at all biological samples, indicating high quality of TT-seq library (Appendix Fig. S9A). Further principal component analysis (PCA) shows a high degree of correlation between two biological replicates of each condition (Appendix Fig. S9B). Differential gene expression analysis revealed that the transcription process of majority nascent transcripts was attenuated due to depletion of *Ezh1* at all time points (Appendix Fig. S9C,D). Absolute fold changes of significantly altered genes exhibit remarkable differences when we compare sh*Ezh1* with WT or sh*Ezh1*^EZH1α condition (Fig. 5B). However, minor changes were observed when comparing the WT and sh*Ezh1*^EZH1α conditions at three time points, indicating that the majority of differentially affected genes' expression levels could be restored to normal levels upon rescuing the function of EZH1α. (Fig. 5B–D). Next, to assess to what extent EZH1 affects nascent

transcription state of circadian genes, we focused on the cluster II gene set, previously defined from total RNA-seq analysis (Fig. 3H). In agreement with mature RNA transcriptome analysis, metaplot and heatmap analyses show that nascent transcription profile of these circadian genes was perturbed in the absence of *Ezh1*, indicating the crucial role of EZH1 in the maintenance of amplitude robustness of these circadian genes (Fig. 5E). Of note, the rhythmicity of nascent transcripts could not fully be restored in sh*Ezh1*^EZH1α condition, indicating accurate dosage dependence of EZH1 in the modulation of circadian transcription activity (Fig. 5E).

In order to further investigate whether the rescued phenotype was attributed to the constitutive expression of EZH1, we generated an additional inducible rescue cell line, sh*Ezh1*^TET-HA-EZH1α (Appendix Fig. S10A,B), and induced the expression of EZH1α-HA for 24 h prior to synchronization. Following synchronization, we performed TT-seq analysis at the peak phase, which corresponds to 16 h after synchronization when the interaction between EZH1α and the RNA Pol II machinery reaches its maximum level (Appendix Fig. S10C). In this experiment, both the transcript and protein levels of the exogenous EZH1α-HA were found to be similar to the endogenous counterpart (Appendix Fig. S10A,D). Notably, the TT-seq assay revealed that the expression of the majority of significantly altered genes observed in the sh*Ezh1* cells, as compared to the wild-type (WT) cells, was restored to normal levels upon transient induction of EZH1α-HA using doxycycline (Appendix Fig. S10E,F). This independent rescue experiment provides compelling evidence for the specific and primary role of EZH1 in transcriptional regulation. To further strengthen the evidence supporting the direct involvement of EZH1α in the regulation of circadian targets, we generated EZH1α ChIP-seq data at different time points using EZH1α tagged with the HA epitope. This analysis shows direct association of EZH1α with core clock genes *Bmal1*, *Per1*, and *Per2* (Appendix Fig. S11). Furthermore, to assess the transcriptional activity of these clock genes, TT-seq results clearly demonstrate that the nascent transcription of these clock genes is compromised upon EZH1α depletion. However, this can be partially restored by the reintroduction of inducible EZH1α (Appendix Fig. S12). Collectively, these data support the direct and specific role of EZH1α in modulating the nascent transcription process.

## EZH1α regulates circadian transcription through stabilization of RNA Pol II complex machinery

The above genomic approaches TT-seq revealed positive effect of EZH1 in regulation of RNA Pol II activity at population levels. We next wanted to quantitatively characterize spatiotemporal dynamics of RNA Pol II at a single-cell resolution in live cells using time-correlated photoactivated localization microscopy (tcPALM) (Cho et al, 2018; Cisse et al, 2013). We engineered WT, sh*Ezh1*, and sh*Ezh1*^EZH1α stable cell lines to express α-amanitin resistant mutant of RPB1, the catalytic subunit of RNA Pol II fused with the photoconvertible fluorescent protein Dendra2, which has been widely used for the characterization of RNA Pol II organization and dynamics using tcPALM (Cho et al, 2016). In alignment with previous findings (Cho et al, 2018; Mylonas et al, 2021), we also observed two types of RNA Pol II clusters, small (<200 nm) and large (>200 nm) clusters, in postmitotic C2C12 myotube cells

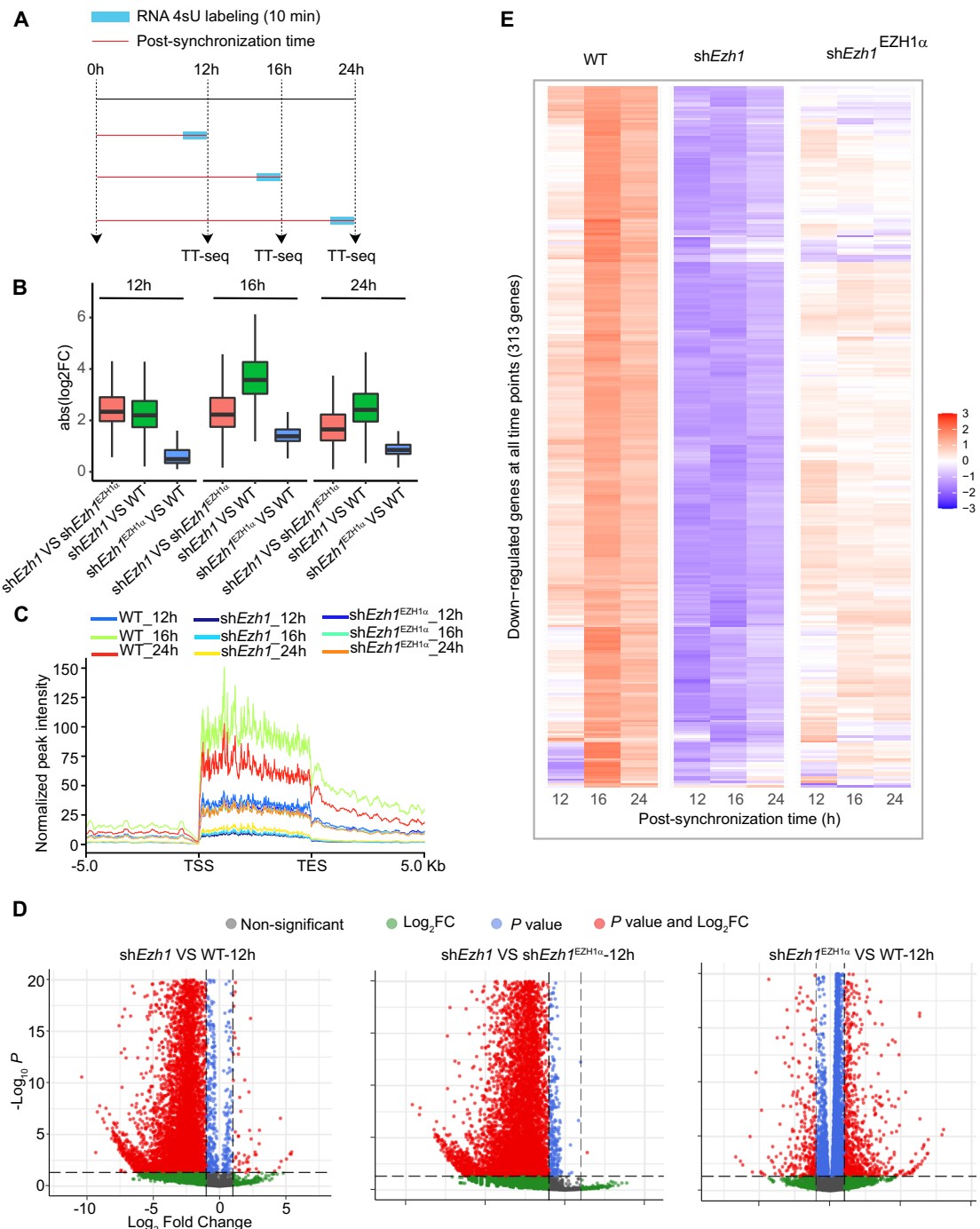

**Figure 5. Global impairment of nascent transcription activity due to EZH1 depletion.**

(A) Experimental design illustrating TT-seq samples collection at indicated time points. (B) Absolute fold changes of significantly differential genes in comparison with indicated biological conditions at specific time point. $N = 15{,}868$, $16{,}582$, $7282$, $18{,}433$, $16{,}685$, $14{,}100$, $16{,}584$, $15{,}491$, and $11{,}659$ genes were analyzed in comparison with within indicated biological conditions and time points from left to right. Detailed minima, maxima, center, bounds of box and whiskers, and percentile value have been provided in Dataset EV8. (C) Metaplot analysis of normalized TT-seq read distribution at centered gene bodies of total RNA-seq cluster II genes in comparison with indicated conditions at different time points, spanned upstream/downstream $+/- 5$ kb. (D) Volcano plot representation of differential expression profile of nascent transcripts in comparison with different biological conditions at indicated post-synchronization time points. The statistical analysis in panel (D) was performed using DESeq2 as described in "Method" section. (E) Heatmap representing nascent transcriptome profile of total RNA-seq cluster I genes downregulated in *Ezh1* knockdown (sh*Ezh1*) condition in comparison with wild-type (WT) and rescue condition (sh*Ezh1*^EZH1α) at indicated post-synchronization time points.

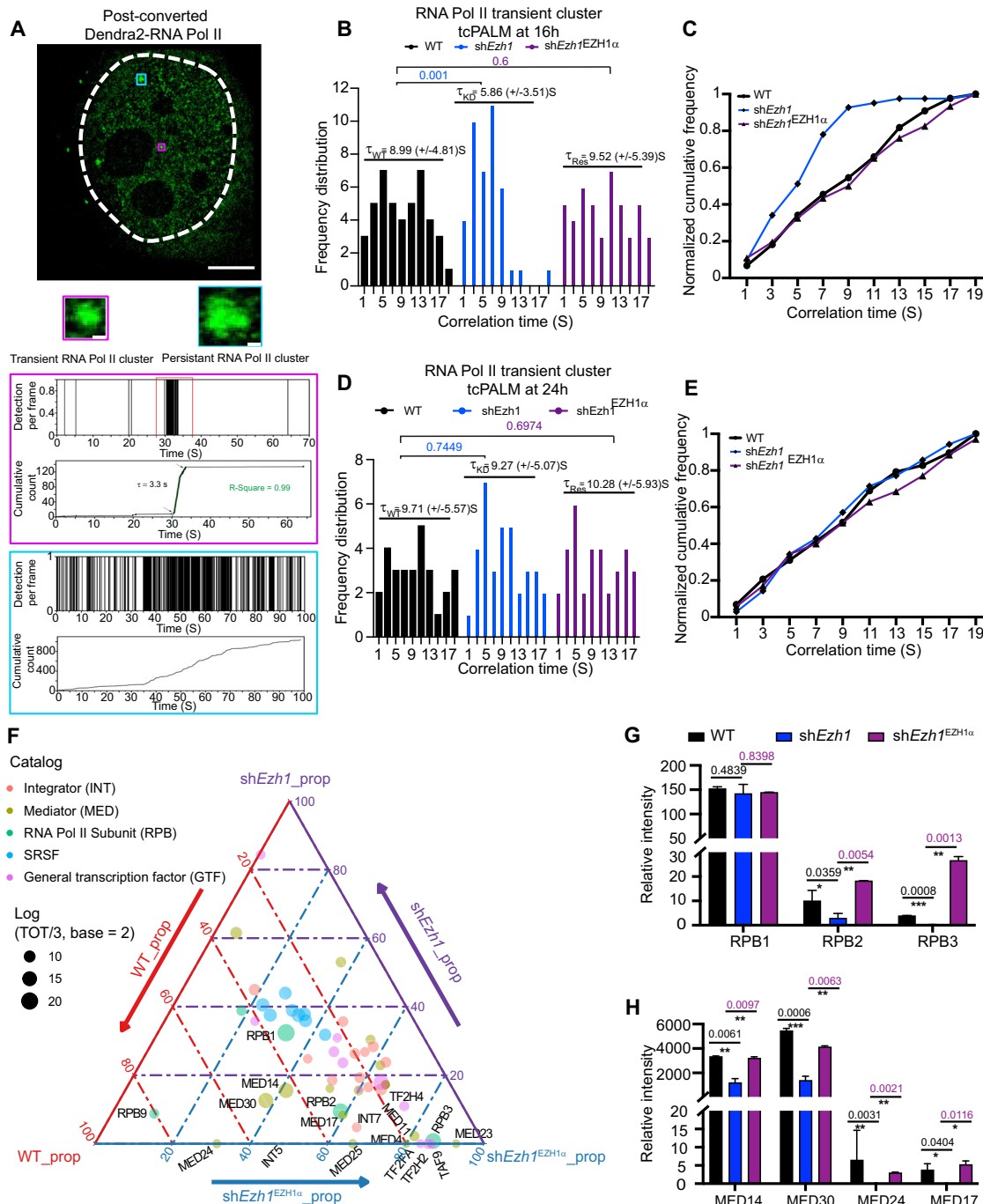

(Fig. 6A). We then tested the extent level to which the dynamics of these clusters would be depending on the function of EZH1. Given the strong interaction between EZH1 and RNA Pol II subunits, we firstly measured numbers of small and large clusters in WT, *shEzh1* and sh*Ezh1*[EZH1α] genetic backgrounds at the 16 h time point when highest interaction between EZH1 and RNAPII occurs. Impaired EZH1 function had no apparent effect on the population of larger clusters. However, significant reduction of small clusters population was identified upon the depletion of *Ezh1* (Appendix Fig. S13A,B; Fig. S14). It has been suggested that the lifetime of the RNA Pol II

small cluster positively correlates with the rate of RNA Pol II initiation (Mylonas et al, 2021). We thus focused on the potential role of EZH1 in the regulation of the lifetime of the small clusters (Appendix Fig. S13). Interestingly, the median lifetime of small clusters decreased to 5.86 (+/−3.51) s upon *Ezh1* depletion (P = 0.001), which could be restored at 9.52 (+/−5.39) s in rescue condition (Fig. 6B,C). Surprisingly, when we examined population of clusters and lifetime of small clusters at the peak phase of PRC2–EZH1 complex assembly (24 h), nor cluster numbers and lifetime of small clusters at 24 h exhibit significant alterations

◀ **Figure 6. EZH1 stabilizes RNA Pol II machinery.**

(A) Upper panel, representative super-resolution reconstruction image of post-converted Dendra2-RNA Pol II left panel. Middle panel, small and large RNA Pol II clusters are shown as indicated color squares from left panel. Lower panel, representative time traces for small and large Pol II clusters, respectively. In the traces, time $t = 0$ means the start of acquisition. Scale bar, 5 μm (5 nm in zoom-in). (B, C) The distribution frequency (B) and normalized cumulative distribution analyses (C) of RNA Pol II lifetime in WT ($n = 45$ clusters from 10 cells measured over three independent experiments), shEZH1 ($n = 40$ clusters from 10 cells measured over three independent experiments) and shEZH1$^{Ezh1a}$ ($n = 47$ clusters from 10 cells measured over three independent experiments) at 16 h. Average RNA Pol II lifetime τ is calculated and shown in each condition. Errors represent standard error of the mean. Statistical analyses are performed using an unpaired two-sided Wilcoxon rank test. (D, E) the distribution frequency (D) and normalized cumulative distribution analyses (E) of RNA Pol II lifetime in WT ($n = 30$ clusters from 10 cells measured over three independent experiments), shEZH1 ($n = 36$ clusters from 10 cells measured over three independent experiments) and shEZH1$^{Ezh1a}$ ($n = 36$ clusters from 10 cells measured over three independent experiments) at 24 h. The average RNA Pol II lifetime τ is calculated and shown in each condition. Errors represent the standard error of the mean. Statistical analyses are performed using an unpaired two-sided Wilcoxon rank test. (F) Ternary plot shows differentially associated proteins with RPB1: RNA Pol II subunits, Mediators (MEDs), General transcription factors (GTFs), Serine/arginine-rich splicing factors (SRSFs) and Integrators (INTs). Proteins were colored based on functional groups, and their size is based on relative amount value in each pairwise comparison. Comparisons were made between WT, shEzh1, and shEzh1$^{EZH1a}$. Arrows represent the direction of the enrichment. (G, H) Normalized intensity of RNA Pol II subunits and indicated MEDs immunoprecipitated together with RPB1 in WT, shEzh1, and shEzh1$^{EZH1a}$ conditions. Error bars represent ± SEM from three independent experiments. *$P < 0.05$; **$P < 0.01$; ***$P < 0.001$; by two-tailed $t$ test. Source data are available online for this figure.

(Fig. 6D,E; Appendix Fig. S13C). These data suggest a crucial role of EZH1 in the modulation of RNA Pol II initiation process in a time-dependent manner.

Based on the TT-seq result (Fig. 5) and the observed periodicity between EZH1 and RNA Pol II subunits and associated components (Appendix Fig. S8), we focused our attention more on the mechanistic consequences and hypothesizing a role of EZH1 in cyclic manner of RNA Pol II complex integrity. Lifetime of RNA Pol II cluster positively correlates with the rate of RNA Pol II initiation process (Mylonas et al, 2021). We reasoned that EZH1 depletion may affect the stability of RNA Pol II machinery. To measure the integrity of RNA Pol II machinery, we immunoprecipitated the endogenous RNA Pol II complex and analyzed co-immunoprecipitated proteins by quantitative proteomic analysis in WT, shEzh1, and shEzh1$^{EZH1a}$ backgrounds. We focused on 16 h after synchronization, which is the same phase defined as active phase of transcriptional activity. Consistently with previous report (Aoi et al, 2021), we observed that other RNA Pol II subunits, Mediators (MEDs), General transcription factors (GTFs), integrators (INTs) and Serine/arginine-rich splicing factors (SRSFs) were co-immunoprecipitated together with RPB1 (Fig. 6F; Dataset EV4). We then measured relative abundance of proteins associated with RPB1 and discovered that other RNA Pol II subunits (RPB2/3), several MEDs (e.g., MED14/30) and GTFs were less associated with RPB1 upon EZH1 depletion (Fig. 6G,H). Strikingly, ectopic expression of EZH1α could restore or enhance the interaction between these RNA Pol II-associated components with RPB1. Upon ectopic expression of EZH1α in shEzh1$^{EZH1a}$ condition, many proteins like RPB2/3, MED17/23 and TAF9 exhibited stronger association with RPB1 (Fig. 6F). These data further prove that stoichiometry of RNA Pol II machinery-associated components are precisely regulated by EZH1, which could also explain partial recovery of periodicity of nascent circadian transcripts (Fig. 5E). Among all those affected co-immunoprecipitated proteins, MED14, a central scaffold component which required for the formation of RNA Pol II clusters (Jaeger et al, 2020), in the absence of EZH1α was found less associated with RPB1 (Fig. 6H). Relative amounts of serine/arginine-rich splicing factors (SRSFs) and integrators (INTs) were not affected in comparison with WT and EZH1 depleted condition (Fig. 6F). These data collectively indicate the crucial and specific role of EZH1 in stabilizing RNA Pol II preinitiation complex integrity and nascent transcription process.

## The distinct profile of PcG complex and H3K27me3 at activated and repressive targets

In order to further investigate the integrated direct role of EZH1α as both an activator and a repressor in mediating RNA Pol II activity and H3K27me3 deposition, we conducted genome-wide profiling of EZH1α, SUZ12, and H3K27me3 occupancy using ChIP-seq analysis at six different time points. To ensure robust EZH1 ChIP-seq data, we employed a stringent biotin-ChIP method, as used in previous studies to profile EZH1α occupancy states (Mousavi et al, 2012; Vo et al, 2018). For this purpose, C2C12 cells were engineered to express BirA ligase and Avi-tagged Ezh1α (Ezh1α-Avi) simultaneously. After confirming the successful biotinylation and expression of the biotinylated EZH1α (Appendix Fig. S15A,B), we obtained high-quality EZH1α ChIP-seq data, using C2C12 cells expressing only BirA ligase as the control. To accurately quantify the relative intensities of SUZ12 and H3K27me3, we performed calibrated ChIP-seq by incorporating spike-in chromatin and antibody (Appendix Fig. S15C–E). The specificity of PcG and H3K27me3 ChIP-seq was also evident from the unique peak enrichment observed around the HoxD Cluster loci (Appendix Fig. S15F). Similar to the previous study (Mousavi et al, 2012), we observed significant enrichment of EZH1α, SUZ12, and H3K27me3 peaks around the transcription start sites (TSS) of target genes (Fig. 7A–C). Using RNA-seq data, we identified Cluster I and II genes, which represent the mature state of total RNA and involve both transcriptional and post-transcriptional regulation and could not reflect the RNA Pol II activity directly. In order to better understand the primary and direct function of EZH1α in the regulation of nascent transcription processes, we selected genes that exhibited the significantly highest expression levels at 16 h and the lowest levels at 24 h, as determined by TT-seq analysis of synchronized wild-type (WT) cells. We classified these genes as rhythmic activation and repressive targets, respectively (Fig. 7D,E; Dataset EV5). Interestingly, the activated and repressive rhythmic targets determined using TT-seq data were enriched in distinct biological processes (Appendix Fig. S15G,H).

Next, when intersecting EZH1α, SUZ12, and H3K27me3 ChIP-seq data with the TT-seq-defined cyclically expressed activation and repressive targets. In order to identify the precise and exclusive targets of EZH1α, SUZ12, and H3K27me3, we conducted a thorough analysis of the common peaks within three biological

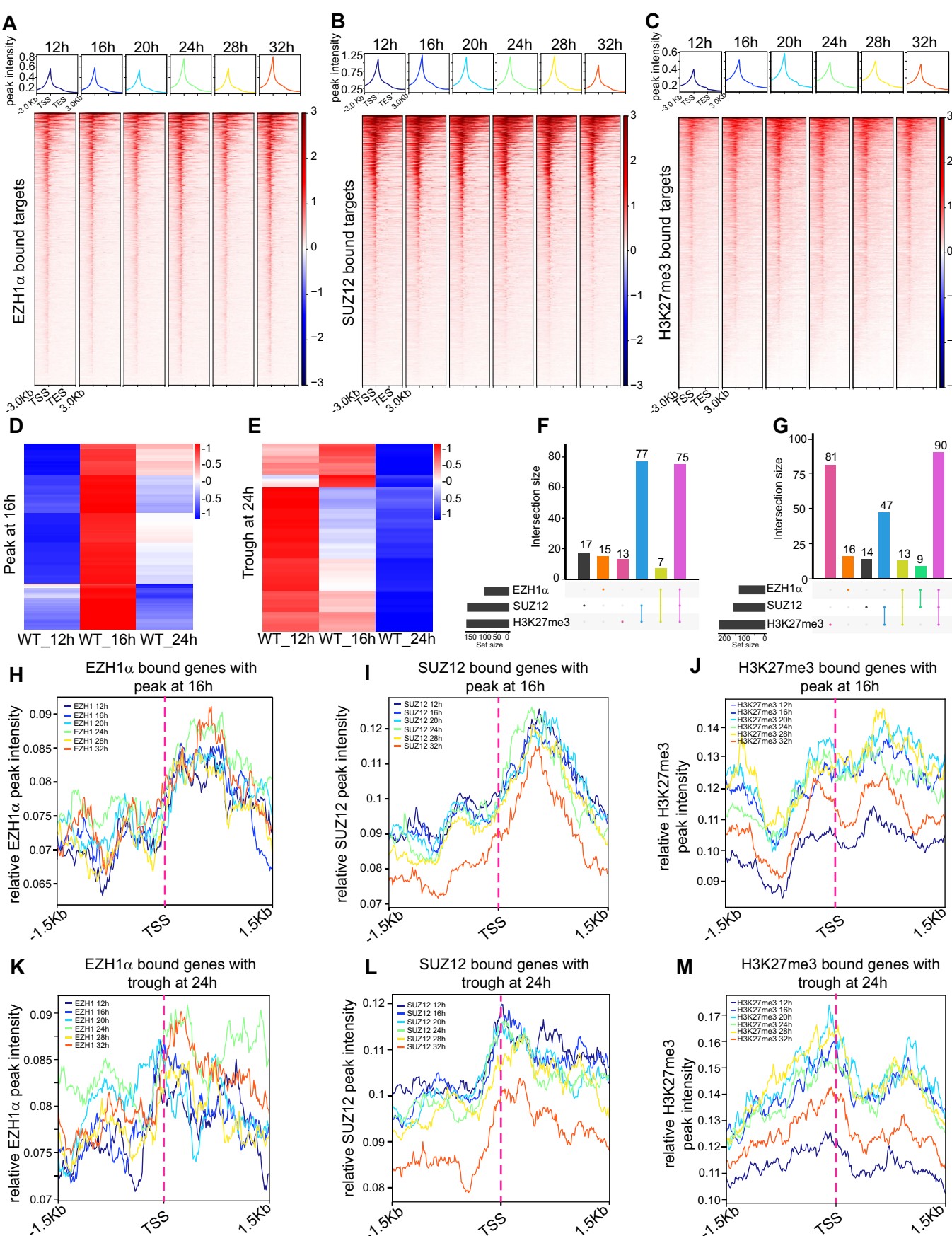

◄

**Figure 7. Differential binding patterns of EZH1α, SUZ12, and H3K27me3 at activated and repressed circadian targets.**

(A–C) Heatmap showing EZH1α (A), SUZ12 (B), and H3K27me3 (C). ChIP-seq signal around transcription start site (TSS), gene body, and transcription termination site (TES) in wild-type (WT) synchronized C2C12 myotube cells at indicated time point. (D, E) Clustering of genes expressed with the highest level (peak) or lowest level (trough) at 16 h (D) and 24 h (E), as determined by TT-seq in wild-type (WT) synchronized C2C12 cells. (F, G) Upset plot illustrating the intersection of EZH1 α, SUZ12, and H3K27me3 co-occupancy status around genes that exhibit peak expression at 16 h (F) or trough expression at 24 h (G), as determined in (D, E). (H–J) Metaplot analyses of EZH1α (H), SUZ12 (I), and H3K27me3. (J) ChIP-seq peak signals at various time points surrounding the transcription start sites (TSSs) of genes exhibiting the highest expression peak at 16 h (peak at 16 h), as determined by TT-seq in wild-type (WT) synchronized C2C12 cells. (K–M) Metaplot analyses of EZH1α (K), SUZ12 (L), and H3K27me3 (M) ChIP-seq peak signals at various time points surrounding the transcription start sites (TSSs) of genes exhibiting the lowest expression peak at 24 h (trough at 24 h), as determined by TT-seq in wild-type (WT) synchronized C2C12 myotube cells.

replicates for each time point, ensuring statistical significance with an adjusted p-value threshold of less than 0.01. Specifically, we focused on those peaks that were annotated within promoter regions (Dataset EV6). We found that the majority of EZH1α-bound activation and repressive targets were also occupied by SUZ12 and possessed H3K27me3 marks (Fig. 7F,G; Dataset EV7). Notably, the peaks of EZH1, SUZ12, and H3K27me3 were specifically enriched around the TSS of repressive targets, consistent with the repressive role of H3K27me3. Interestingly, we also observed that both EZH1 and SUZ12 are directly bound to activated targets; however, H3K27me3 enrichment was not observed around the TSS of activated targets (Fig. 7H–M). Analyzing the intensity of EZH1α, SUZ12, and H3K27me3 across one nucleosome upstream and downstream of both activated and repressive targets' TSSs, we found no significant changes in the average intensity of these marks around activation and repressive targets, except for the intensity of H3K27me3 around repressive targets (Appendix Fig. S16). Interestingly, the peak phase of H3K27me3 occupancy around repressive targets coincided with the peak phase of EED shuttle and canonical PRC2–EZH1α complex assembly (Fig. 4A–C). In conclusion, our findings suggest that EZH1 and SUZ12 co-occupy activated targets constitutively, while the trafficking of EED from the cytosol to the nucleus may facilitate the assembly of the canonical PRC2–EZH1 complex and modulate the deposition of H3K27me3 marks specifically around repressive circadian targets that are expected to be repressed at this particular time point.

## Discussion

Our current research has identified the PRC2–EZH1 complex, specifically EZH1, as a novel and crucial player in the regulation of the circadian clock. Mechanistically, we have investigated how the coordination between RNA Pol II activity and H3K27me3 epigenome plasticity regulates circadian gene expression through the precise spatial-temporal partitioning of EZH1α into different complexes within postmitotic skeletal muscle cells.

Our hypothesis suggests that throughout the entire circadian cycle, EZH1α, possibly in conjunction with SUZ12, is constantly associated with the RNA Pol II machinery, via stabilization of the RNA Pol II initiation complex, ensuring the basal transcription of of several tissue-specific genes as well as central clock genes including *Bmal1, Per1*, and *Per2*. At certain phases, EED translocate from the cytosol to the nucleus, facilitating the assembly of the PRC2–EZH1α repressive complex and the deposition of H3K27me3 marks around these targets. This idea is supported by

the dynamic interaction observed between EZH1α and other components of the PRC2–EZH1α complex, as well as the differential occupancy of H3K27me3 marks around the transcription start site (TSS) of these targets. Regarding the enhanced activation of a subset of circadian targets, our analysis of EZH1α binding profiles reveals that rhythmicity is not observed around these targets. Furthermore, there is a lower enrichment of H3K27me3 marks around the transcription start site (TSS) of these activation targets, with a higher enrichment observed in the gene body region downstream of the TSS. This specific enrichment pattern may be involved in the process of H3K27me3 demethylation. Once a distinct H3K27me3 pattern is established around activated targets, cyclic interactions between EZH1α and transcriptional key regulators, such as Mediators, facilitate the activation process with RNA Pol II. A previous study has reported that two subunits, CHD4 and MTA2, within the Mi-2/nucleosome remodeling and deacetylase (NuRD) transcriptional corepressor complex promote the transcriptional activity of the BMAL1-CLOCK complex through their constitutive association (Kim et al, 2014b). During the onset of the negative feedback phase, the PER complex aids in the reconstitution of the full NuRD corepressor complex by delivering the remaining components, thereby acquiring full repressor activity and targeting BMAL1-CLOCK (Kim et al, 2014b). In a similar vein, our study not only uncovers the mechanism by which EED coordinates the H3K27me3 patterns surrounding EZH1 repressive targets but also presents an instance where certain components of traditionally recognized repressor complexes play a positive role in regulating specific circadian targets during particular phases.

In our current study, we made an intriguing discovery that the EZH1β protein exhibits a 12-h rhythm, specifically in skeletal muscle tissue. However, this 12-h rhythm is not observed in cell-autonomous systems. These findings collectively suggest the presence of potential endogenous physiological signals, such as metabolic signals or inter-organ communication signals, in vivo, which are absent in synchronized in vitro cell cultures. Previous reports have indicated that XBP1 acts as a major transcription factor governing the 12-hour cycling pattern (Zhu et al, 2017), and regulates various lipid and metabolic pathways associated with this rhythm (Ballance and Zhu, 2021). Interestingly, the expression of *Ezh1β* transcripts shows a 24-h rhythm, suggesting a regulation at the post-transcriptional, translational, or post-translational levels that is independent of XBP1-mediated transcriptional regulation. In our previous study, we demonstrated that ubiquitin E3 ligases, such as HUWE1 and NEDD4 (Liu et al, 2019) interacts with EZH1β protein. Ubiquitin-mediated proteasome degradation has been extensively studied and plays an essential role in regulating

protein rhythmic homeostasis (Patke et al, 2020a). It is worth investigating whether ubiquitin-mediated degradation could govern the 12-hour rhythm of EZH1β in a circadian pattern by sensing potential endogenous signals, such as metabolic signals. However, establishing a direct connection between ubiquitin E3 ligase and EZH1β through depletion of HUWE1 or NEDD4 would be challenging, given the multiple downstream targets of both HUWE1 and NEDD4. For example, HUWE1 has been shown to be involved in the degradation of REV-ERBα (Yin et al, 2010). In our rescue cell line shEzh1^EZH1α, where EZH1β is downregulated, we observed that transcriptional rhythmicity could not be fully restored. This indicates the essential role of EZH1β in orchestrating circadian gene expression, possibly by modulating the rhythmic shuttle of EED and the assembly of the PRC2–EZH1 repressive complex. In addition to the metabolic signals that may drive EZH1β degradation, nuclear compartment-specific metabolism could also influence circadian chromatin remodeling, such as the crucial role of AHCY in regulating cyclic H3K4me3 states through the S-adenosylmethionine (SAM)-dependent pathway (Greco et al, 2020). Notably, we have also identified disruptions in H3K4me3 and H3K27me3 balance following EZH1α depletion. Therefore, investigating the potential link between rhythmic nuclear metabolic states and EZH1α function could provide valuable insights into how the PcG complex coordinates metabolic states and epigenome alterations within the context of circadian rhythm.

Our data indicate EZH1α playing a key role in maintaining the stability of the RNA Pol II preinitiation complex and facilitating active transcription. However, the exact mechanism by which EZH1α achieves this stability remains to be investigated. Previous research has suggested that Elongin A (Ardehali et al, 2017) and RNA Pol II CTD (C-terminal domain) (Dias et al, 2015) can undergo methylation, which in turn affects the activity of RNA Pol II. EZH1 may be involved in non-histone methylation of the RNA Pol II machinery or its associated components. Future work will be required to determine the precise methylation targets of EZH1α within the RNA Pol II complex and to understand how these modifications influence the integrity and activity of the RNA Pol II machinery.

Taken together, our study provides novel mechanistic insights into the rhythmicity of circadian physiology by unveiling the coordinated action of EZH1 on both the epigenome and the basal transcriptional machinery.

# Methods

**Reagents and tools table**

| Reagent/resource | Reference or source | Identifier or catalog number |
|---|---|---|
| **Experimental models** | | |
| C2C12 cell (*M. musculus*) | ATCC | CRL-1772 |
| HEK293T cell line (*H. sapiens*) | ATCC | CRL-3216 |
| C57BL/6J (*M. musculus*) | JAX | N/A |
| **Recombinant DNA** | | |
| pOZ-C-FH-EZH1α-FH | Liu et al, 2019 | N/A |

| Reagent/resource | Reference or source | Identifier or catalog number |
|---|---|---|
| pCW57-MCS1-P2A-MCS2 (Neo)-EZH1α-FH | This study | #89180 |
| pMY-fb-Ezh1-T2A-GFP | Prof. Vittorio Sartorelli Lab (NIAMS, NIH) | N/A |
| pMY-BirA-T2A-mOrange | Pr. Vittorio Sartorelli Lab (NIAMS, NIH) | N/A |
| PB53x EF1-Dendra2-RPB1Amr | Addgene | #81228 |
| **Antibodies** | | |
| Anti-BMAL1 | Abcam | Ab93806 |
| Anti-CRY1 | Bethyl | A302-614A |
| Anti-PER2 | Alpha diagnostic | PER21-A |
| Anti-PER1 | Abcam | ab136451 |
| Anti-Rev-ERBα | Cell Signaling Technology (CST) | 13418 |
| Anti-EZH1α, for WB | Active Motif | 61583 |
| Anti-EZH1α, for ChIP-qPCR | Abcam | 64850 |
| Anti-EZH1β | Homemade antibody provided by Orlando's lab | N/A |
| Anti-SUZ12 (DR9F6 XP) | Cell Signaling Technology (CST) | 13701 |
| Anti-ACTIN | Santa Cruz | Sc-47778 |
| Anti-Histone 3 | Abcam | ab1791 |
| Anti-p84 | Abcam | ab487 |
| Anti-EED | Millipore | 17-663 |
| Anti-H3K27me3 | Millipore | 07-449 |
| Anti-H3K4me3 | Abcam | ab8580 |
| RNA Pol II (8WG16), unmodified | Abcam | ab26721 |
| RNA Pol II S5P | Abcam | ab5131 |
| RNA Pol II S2P | Abcam | ab24758 |
| RPB1 (clone 4H8) | Cell Signaling Technology (CST) | 2629 |
| Anti-a-tubulin | Cell Signaling Technology (CST) | 3873 |
| Anti-HA | Abcam | ab9110 |
| **Oligonucleotides and other sequence-based reagents** | | |
| For oligos, shRNA and siRNA sequence information, please refer to the relevant EV table | N/A | N/A |
| **Chemicals, enzymes, and other reagents** | | |
| DMEM, high glucose, GlutaMAX™ Supplement | Thermo Fisher Scientific | 61965026 |
| Fetal Bovine Serum | Thermo Fisher Scientific | 10100147 |
| Horse Serum | Thermo Fisher Scientific | 26050070 |
| FluoroBrite™ DMEM | Thermo Fisher Scientific | A1896701 |
| DPBS (−/−) | Thermo Fisher Scientific | 14190094 |
| Trypsin-EDTA (0.25%), phenol red | Thermo Fisher Scientific | 25200056 |
| Direct-zol RNA Miniprep Kit | Zymo Research | R2050 |

| Reagent/resource | Reference or source | Identifier or catalog number |
|---|---|---|
| iScript™ cDNA Synthesis Kit | Bio-Rad | 170-8891 |
| SsoAdvanced Universal SYBR Green Supermix | Bio-Rad | 172-5270 |
| DharmaFECT 1 Transfection Reagent | Dharmacon | T-2005-01 |
| Lipofectamine™ 2000 Transfection Reagent | Thermo Fisher Scientific | 11668500 |
| 3rd Generation Packaging Mix Kit | ABMgood | LV053 |
| Opti-MEM Reduced Serum Medium | Thermo Fisher Scientific | 31985062 |
| ANTI-FLAG® M2 Affinity Gel | Merck-Sigma | A2220 |
| Flag peptide | Merck-Sigma | F3290 |
| Pierce™ Anti-HA Agarose | Thermo Fisher Scientific | 26182 |
| HA peptide | Thermo Fisher Scientific | 26184 |
| Protein G Dynabeads | Thermo Fisher Scientific | 10003D |
| Pierce™ 16% Formaldehyde (w/v), Methanol-free | Thermo Fisher Scientific | 28906 |
| Pierce™ DSG, No-Weigh™ Format | Thermo Fisher Scientific | A35392 |
| Dynabeads™ MyOne™ Streptavidin T1 beads | Thermo Fisher Scientific | 65602 |
| Spike-in antibody | Active Motif | 61686 |
| Spike-in chromatin | Active Motif | 53083 |
| 4-thiouridine (4sU) | Merck-Sigma | T4509 |
| EZ-Link™ HPDP-Biotin, No-Weigh™ Format | Thermo Fisher Scientific | A35390 |
| µMACS Streptavidin Kit | Miltenyi Biotec | 130-133-282 |
| TruSeq Stranded Total RNA Library Prep Gold | Illumina | RS-122-2201 |
| ChIP-DNA Clean & Concentrator | Zymo | D5205 |
| NEBNext® Ultra™ II DNA Library Prep Kit | NEB | E7645L |
| NEBNext® Multiplex Oligos for Illumina® (Dual Index Primers Set 1) | NEB | E7600S |
| **Software** | | |
| Spectronaut | https://biognosys.com/software/spectronaut/ | N/A |
| GraphPad Prism 10 | https://www.graphpad.com | N/A |
| ImageJ | https://imagej.nih.gov/ij/index.html | N/A |
| Artificial Intelligence RNA-seq Software Service (SaaS) platform | https://transcriptomics.cloud | N/A |
| BBDuk | https://jgi.doe.gov/data-and-tools/bbtools/bb-tools-user-guide/bbduk-guide/ | N/A |
| STAR | Dobin et al, 2013 | N/A |
| RSeQC | Wang et al, 2012 | N/A |

| Reagent/resource | Reference or source | Identifier or catalog number |
|---|---|---|
| featureCounts | Liao et al, 2014 | N/A |
| JTK CYCLE | Hughes et al, 2010 | N/A |
| BIO_CYCLE | https://circadiomics.igb.uci.edu/biocycle | N/A |
| DESeq2 | Love et al, 2014 | N/A |
| deepTools | Ramirez et al, 2014 | N/A |
| SAMtools | Li et al, 2009 | N/A |
| MACS2 | Zhang et al, 2008 | N/A |
| ChIPseeker | Yu et al, 2015 | N/A |
| Localizer | Dedecker et al, 2012 | N/A |
| **Other** | | |
| Novaseq6000 platform | Illumina | N/A |

## Muscle tissue samples preparation

Age-matched, male C57BL/6J mice (JAX, 00064) were maintained on a 12 h light/12 h dark cycle. At 6 weeks of age, animals were placed on a normal chow diet (Prolab RMH 2500), for 10 weeks. After 10 weeks, Gastrocnemius muscle tissue was dissected following the circadian cycle every 4 h (ZT0, ZT4, ZT8, ZT12, ZT16, ZT20, and ZT24) where ZT0 corresponds to light ON and ZT12 to lights-off in the animal facility (5 mice /time point). Samples were snap-frozen in liquid nitrogen. All experiments were performed upon approval by the Institutional Animal Care and Use Committee at the University of California at Irvine.

## C2C12 cell culture and synchronization

C2C12 mouse myoblasts cells (ATCC, CRL-1772™) were cultured in Dulbecco's modified Eagle medium (4.5 g/l D-glucose and Glutamax) (GIBCO) supplemented with 100 U/ml penicillin/streptomycin and 10% fetal bovine serum (FBS) (GIBCO). Differentiation was induced when cells reached ~90–95% confluence using DMEM supplemented with penicillin and streptomycin and containing 2% horse serum (HS) (GIBCO). C2C12 myotubes were synchronized using 50% horse serum shock for 1 h (Chatterjee et al, 2013; Gréchez-Cassiau et al, 2008; Peek et al, 2015). Then, cells were washed with warm PBS and replaced in 2% HS. Cells were collected at MT4-5 every 4 h following the circadian time for 24 h (12 h, 16 h, 20 h, 24 h, 28 h, 32 h) (3 biological replicates/time point). Cell lines were tested negative for mycoplasma contamination using the MycoFluor™ Mycoplasma Detection Kit (Thermo Fisher).

## RNA isolation and gene expression analyses

For muscle tissue, ≈40 mg of tissues from all circadian time points were used for RNA extraction. RNA was extracted by resuspending frozen tissues in 1 ml of TRIzol and homogenized using the TissueLyser II (QIAGEN) for 2 min at 30 Hz, two times. RNA concentration was measured using nanodrop and purified via

RNeasy spin columns (Qiagen). For C2C12, cells were harvested and lysed with Trizol Reagent. Total RNA was isolated by Direct-zolTM RNA MiniPrep Kit (Zymo Research). cDNA was prepared with 1 μg of total RNA using iScript cDNA synthesis kit (Bio-Rad), following the manufacturer´s protocol. cDNA was diluted 1:10 for downstream applications. Quantitative real-time PCR was performed in BioRad CFX96 real-time PCR Machine using cDNA template, primer mix (Dataset EV1), and SYBR green (Bio-Rad) in a final volume of 20 μl PCR reaction.

## siRNA knockdown

Knockdown of *Bmal1* was accomplished by using Silencer Select siRNAs targeting *Bmal1* (s62620, s62621, s62622, Thermo Fisher Scientific). Silencer™ Negative Control No. 1 siRNA (Thermo Fisher Scientific) was used as the control for the knockdown experiments. Cells were transfected with siRNAs and DharmaFECT reagent (Dharmacon) at a final concentration of 5 nM following the manufacturer's instructions. Gene silencing was monitored 48 h after transfection, and cells were harvested and lysed with Trizol for total RNA extraction.

## Construct generation

A retroviral vector expressing EZH1-Avi tag (pMY-fb-Ezh1-T2A-GFP) or BirA ligase (pMY-BirA-T2A-mOrange) was the gift from Pr. Vittorio Sartorelli Lab (NIAMS, NIH). To construct the inducible EZH1 rescue construct, a full-length CDS sequence of *Ezh1α* without stop codon was amplified with corresponding specific primers: EZH1α-F-Kozak-GCA AgeI and EZH1α-R-NLS-FH EZH MluI (Dataset EV1) and ligated into pJET1.2 (Thermo Fisher Scientific) vector for Sanger sequencing. Sanger sequencing confirmed inserts were cut with AgeI/MluI and finally ligated into retroviral vector pCW57-MCS1-P2A-MCS2 (Neo) vector (Plasmid #89180), producing sh*Ezh1*$^{\text{TET-EZH1α}}$ vector. Cloning of EZH1α-FLAG-HA has been described previously (Liu et al, 2019), full length or truncated CDS without stop codon were amplified with corresponding specific primers and ligated into pJET1.2 (Thermo Fisher Scientific) vector for Sanger sequencing. Sanger sequencing confirmed inserts were cut with XhoI/NotI and finally ligated into retroviral vector pOZ-C-FH vector (sh*Ezh1*$^{\text{EZH1α}}$ vector).

## Stable sh*Ezh1*, sh*Ezh1* $^{\text{EZH1α}}$, sh*Ezh1*$^{\text{TET-EZH1α}}$, sh*Ezh1*$^{\text{BirA}}$, and sh*Ezh1*$^{\text{BirA+EZH1α-Avi}}$ cell line generation

For sh*Ezh1*, at 50% confluency, C2C12 cells were incubated with shRNA lentivirus targeting both *Ezh1α and Ezh1β* for 18–20 h at 37 °C (SHCLNV, Sigma). Then, cells were washed and replaced with fresh media with fresh puromycin antibiotics till resistant colonies could be identified after 4–5 days. Lentiviral Particles pLKO.1-puro was used as a negative control shRNA (SHC001V, Sigma).

For sh*Ezh1*$^{\text{EZH1α}}$, sh*Ezh1*$^{\text{BirA}}$, and sh*Ezh1*$^{\text{BirA+EZH1α-Avi}}$, to package retrovirus, Phoenix-Eco (ATCC, CRL-3214) was transfected by sh*Ezh1*$^{\text{EZH1α}}$, pMY-fb-Ezh1-T2A-GFP) or BirA ligase pMY-BirA-T2A-mOrange vector using Lipofectamine 2000 (Thermo Fisher Scientific) according to standard protocol. Transfection medium was changed to virus collection medium (DMEM plus 5% FBS) after 8 h of lipofectamine transfection. After 48 h, virus collection medium containing retrovirus was filtered with 0.45-μm filter and be ready

for titer assay or transduction. The validated retroviral vector containing GFP protein was used as a positive control during lipofectamine-mediated transfection. All freshly prepared and tittered retrovirus was used to infect *Ezh1* knockdown C2C12 mouse skeletal myoblasts stable cell line (ATCC, CRL-1772), 8 μg ml$^{-1}$ polybrene was added during infection procedure. After 8 h, fresh growth medium was added to replace infection medium, after that, C2C12 was allowed to grow another 24–48 h before they reach 80% confluence. sh*Ezh1*$^{\text{EZH1α}}$ positive cells were selected using Anti-CD25 beads (Invitrogen). For sh*Ezh1*$^{\text{BirA}}$ and sh*Ezh1*$^{\text{BirA+EZH1α-Avi}}$, we isolated GFP$^{+}$ & mOrange$^{+}$ double-positive (sh*Ezh1*$^{\text{BirA+EZH1α-Avi}}$) or mOrange$^{+}$ (sh*Ezh1*$^{\text{BirA}}$) positive cells from the infected samples at day 2 after retroviral transfection. We first treated the cells with Trypsin enzyme for 5 min at 37 °C to dissociate them and then spun them down at $300 \times g$ for 5 min. The cell pellets were resuspended in 2 ml of sorting buffer (PBS with 1% FBS), and FCSAriaTM Fusion (BD company) was used to sort the double-positive or single-positive cells for further experiments. After sorting, isolated GFP$^{+}$ &mOrange$^{+}$ or mOrange$^{+}$ cells were analyzed by FACS to confirm the efficiency of cell sorting.

For sh*Ezh1*$^{\text{TET-EZH1α}}$, lentivirus production was performed in HEK293T (ATCC, CRL-3216) and sh*Ezh1*$^{\text{TET-EZH1α}}$ vector using the 3rd Generation Packaging Mix kit (Abmgood) following commercially provided protocol. After 48 h, the virus collection medium containing lentivirus was filtered with 0.45-μm filter and ready for titer assay or transduction. The validated lentiviral vector containing GFP protein was used as a positive control during lipofectamine-mediated transfection. All freshly prepared and tittered lentivirus was used to infect *Ezh1* knockdown C2C12 mouse skeletal myoblasts stable cell line (ATCC, CRL-1772), 8 μg ml$^{-1}$ polybrene was added during infection procedure. After 8 h, fresh growth medium was added to replace infection medium, after that, C2C12 was allowed to grow another 24–48 h before they reach 80% confluence. All positive cells were screened by G418 antibiotics.

## Chromatin fractionation

To obtain total extracts, cells were lysed in a total extraction buffer of 50 mM Tris-HCl (pH 8), 150 mM NaCl, 1% NP-40, and 1× protease inhibitor. The lysate was then subjected to sonication and centrifugation at $16,300 \times g$ and 4 °C for 10 min. The nuclei were isolated from the cytosolic fraction using ice-cold 0.1% NP-40 in PBS. The supernatant containing the cytosolic fraction was retained. For the extraction of nuclei and chromatin fractions, a salt concentration gradient was utilized (Della Valle et al, 2022; Herrmann et al, 2017; Teves and Henikoff, 2012). Briefly, the nuclei pellet was resuspended in strip buffer composed of 10 mM Tris-Cl (pH 7.4), 1 mM EGTA, 1.5 mM KCl, 5 mM MgCl$_2$, 290 mM sucrose, 0.1% Triton X-100, and 1 mM DTT for nucleosol extraction. Subsequently, the chromatin-bound fraction was prepared by using a medium salt buffer containing 20 mM HEPES (pH 7.9), 25% glycerol, 500 mM KCl, 1.5 mM MgCl$_2$, 0.2 mM EDTA, 1 mM DTT, 1× Complete mini EDTA-free (Roche), and subjected to sonication (Bioruptor, 30 s ON/30 s OFF, 10 cycles) to isolate chromatin fractions sequentially.

## Immunoblotting analysis

Roughly 40 mg of the muscle tissues were used for total extract. and lysed in 1 ml of modified RIPA buffer (50 mM Tris-cl [pH 8],

150 mM NaCl, 5 mM EDTA, 15 mM MgCl$_2$ and 1% NP-40) supplemented with 1 mM PMSF, 1× Protease Inhibitor Cocktail (Roche), 20 mM NaF (phosphatase inhibitor), 10 mM nicotinamide (sirtuin inhibitor) + 330 nM TSA (Class I and II HDAC inhibitor). Tissues were homogenized using the TissueLyser II (QIAGEN) for 2 min at 30 Hz, two times. For C2C12, cells were harvested and washed twice with PBS. Total protein was extracted and lysed completely in total extraction buffer (50 mM Tris-HCl [pH 8], 150 mM NaCl, 1% NP-40, 0.5 mM PMSF, and protease inhibitory cocktail (Roche). After 30 min incubation on ice, samples were sonicated and centrifuged at 13,000 rpm for 10 min and total protein was collected. Protein concentration was determent by using Bradford assay (BioRad). In total, 1–4 µg of quantified proteins were denatured and loaded on SDS-PAGE using 8% gel (homemade gels) depending on protein mass and transferred into nitrocellulose membrane through wet transfer. The membrane was blocked (milk 5% in PBS-tween 0.1%) and incubated overnight at 4 °C with primary antibodies (milk 5% in PBS-tween 0.1%), then with the appropriate secondary antibody coupled to HRP (Santa Cruz) (milk 5% in PBS-tween 0.1%) and revealed through chemiluminescence using Amersham ECL Kit (GE Healthcare Life Sciences). Bands were visualized by ChemiDoc imaging system (Biorad). The results were normalized to an internal loading control.

## Protein extraction for tandem affinity purification (TAP)

Cytosolic and nuclear extracts were prepared using our previous protocol (Bodega et al, 2017), with minor modifications. Briefly, cells were lysed in cytosolic extraction buffer (50 mM Tris-HCl, pH 8.0, 150 mM NaCl, 0.5 mM EDTA, 0.5% Triton X-100, 5% glycerol). The nuclei were collected at 1500 RCF and 4 °C, and the supernatant was stored as cytosolic extracts. The nuclei were washed three times in cytosolic extraction buffer and suspended in nuclear extraction buffer (50 mM Tris-HCl, pH 8, 50 mM NaCl, 0.5 mM EDTA, 0.5% Triton X-100, 5% glycerol), sonicated (BRANSON A250 with a 3.2 mm tapered microtip; two cycles of 30 s at 20% amplitude, 50% of duty cycle). The debris was pelleted at 16,380 RCF and 4 °C, and the supernatant was used for nuclear fraction extracts. Before IP, the NaCl concentration was adjusted to 150 mM. Extraction buffer was supplemented with fresh 1 × EDTA protease inhibitor cocktail (Roche).

For TAP, tagged proteins were immunoprecipitated with anti-Flag M2-agarose (Sigma), and eluted with Flag peptide (0.2 mg/mL). Further affinity purification was performed with anti-HA antibody-conjugated agarose (Pierce) and eluted with HA peptide (0.2 mg/mL). The HA and Flag peptides were prepared as 5 mg/ml stock in 50 mM Tris-Cl (pH 7.4) and 150 mM NaCl buffer, then diluted in a corresponding concentration in TGEN-150 buffer (20 mM Tris (pH 7.65), 150 mM NaCl, 3 mM MgCl$_2$, 0.1 mM EDTA, 10% glycerol, 0.01% NP-40). Between each step, the beads were washed in TGEN-150 buffer three times.

## Immunoprecipitation of RNA Pol II

Nuclear extracts were prepared using a similar protocol as described in TAP section. NaCl concentration was adjusted 150 mM. RNA Pol II immunoprecipitation was performed through incubation at 4 °C overnight using antibody against Pol II CTD

(clone 4H8, Cell signaling #2629 M), conjugated to Protein G Dynabeads (Thermo scientific, #10004D). After washing with TGEN-150 buffer three times, immunoprecipitated proteins were eluted twice in SDS elution buffer (0.1% SDS, 50 mM Tris-Cl pH 8.0, 1 mM EDTA).

## Protein digestion, peptide fractionation, LC-MS, and DIA data analysis

Both HA peptide eluted samples from the TAP assay and eluted proteins after immunoprecipitation of RNA Pol II were concentrated to 30 µl volume, then diluted in 8 M urea in 0.1 M Tris-HCl, followed by protein digestion with trypsin, according to the FASP protocol (Wisniewski et al, 2009). After overnight digestion, the peptides were eluted from the filters with 25 mM ammonium bicarbonate buffer. The eluted peptides were processed in the desalting step by using Sep-Pag C18 Column (waters) based on the manufacture's instruction.

The DIA-MS analysis of the peptide mixture was conducted on Orbitrap Fusion Lumos mass spectrometer (Thermo Fisher Scientific) coupled with an UltiMate™ 3000 UHPLC (Thermo Fisher Scientific). Approximately 0.5 µg peptide mixtures were separated using a pre-column (Acclaim PepMap, 300 µm × 5 mm, 5-µm particle sizes) and desalted for 15 min with 0.1% FA in water at a flow rate of 5 µL/min. Then, the peptide was eluted into an EasySpray C18 Column (50 cm × 75 µm ID, PepMap C18, 2-µm particles, 100 Å pore size, Thermo Scientific) and separated with a 130-min gradient at constant 300 nL/min at 40 degrees with a flow rate of 300 nl/min. The gradient was established using mobile phase A (0.1% FA in H$_2$O) and mobile phase B (0.1% FA, 95% ACN in H2O): 2.1–5.3% B for 5 min, 5.3–10.5% for 15 min, 10.5–21.1% for 70 min, 21.1–31.6% B for 18 min, ramping from 31.6 to 94.7% B in 2 min, maintaining at 94.7% for 5 min, 4.7% B for 15-min column conditioning. The sample was introduced into the Fusion Lumos mass spectrometer through an EasySpray (Thermo Fisher Scientific) with an electrospray potential of 1.9 kV. The ion transfer tube temperature was set at 270 °C. The MS parameters included application mode as standard for peptide, RF lens as 30%, default charge state of 3 and the use of EASY-IC as internal mass calibration in both precursor ions (MS1) and fragment ions (MS2). Other parameters for MS1 include a resolution of 60,000 (at 200 $m/z$), a maximum ion accumulation time of 50 ms, a target value of 2e5, and data type of profile. For the MS2 by DIA, the mass defect was 0.9995. The normalized HCD collision energy was set to 30% for peptide fragmentation. MS2 spectra were recorded in centroid mode at 30,000 resolutions from 350 to 1500 $m/z$, with an AGC target of 1e6 and a maximum ion accumulation time of 100 ms.

The DIA data were analyzed by Spectronaut (Biognosys, Switzerland) has been reported previously (Zhang and Bensaddek, 2021), using directDIA approach. The protein database used in this study was a combination of the Uniprot Mus musculus (Mouse) and proteome (Proteome ID: UP000000589) sequence. The default settings for the database match consisted of the following parameters: full trypsin cleavage, peptide length of between seven and 52 amino acids and the maximum missed cleavage of two. In addition, lysine and arginine (KR) were used as special amino acids for decoy generation, and N-terminal methionine was removed during the pre-processing of the protein database. In the directDIA, data extraction was based on maximum intensity in both MS1 and

MS2 spectra with relative mass tolerances of 10 and 20 ppm, respectively. Retention time window of Extracted ion Chromatogram (XIC) was set dynamically but with a correction factor of 0.5 to enhance specificity. The Biognosys default settings were applied for both identification and quantification. The *P* value was calculated by the kernel-density estimator. Interference correction was activated and a minimum of three fragment ions and 2 precursor ions were kept for the quantitation. Peptide (stripped sequence) quantity was measured by the mean of 1–3 best precursors, and protein quantity was calculated accordingly by the mean of 1–3 best peptides. Local normalization strategy and *q*-value sparse selection were used for cross run normalization. To determine the differential abundance between samples, the major group (quantification settings) was used for the differential abundance grouping, and the precursor ion (quantification settings) was chosen in the smallest quantitative unit. In addition, an unpaired *t* test with assuming equal variance, group-wise testing correction and clustering was carried out. Proteins with a fold change of higher than 1.5 and a *q*-value of less than 0.01 were considered as differentially expressed proteins.

### Chromatin immunoprecipitation (ChIP)

For muscle tissue, minced frozen tissues (3 biological replicates) were cross-linked with 1% formaldehyde (Thermo Fisher Scientific, 28906) for 10 min followed by Glycine (0.125 M final concentration) at RT for 10 min. Tissues were lysed, homogenized, and sonicated using Branson (BRANSON A250 with a 3.2-mm tapered microtip; 4–5 cycles of 2 min at 20% amplitude, 50% of duty cycle) to generate 200–500 base pairs fragments. Each IP reaction was performed using 100 μg of chromatin DNA equivalents and incubated with 5–8 μg of specific antibodies overnight at 4 °C. To monitor ChIP assay specificity, control samples were immunoprecipitated with a specific-antibody isotype-matched control immunoglobulin (IgG). For H3K4me3 and H3K27me3 ChIP in C2C12, chromatin was performed as previously described. Cells were cross-linked with 1% of formaldehyde and lysed in lysis buffer 1 (50 mM Hepes KOH [pH 7.5], 10 mM NaCl, 1 mM EDTA, 10% glycerol, 0.5% NP-40, 0.25% Triton X-100). Nuclei were collected, washed in Lysis buffer 2 (10 mM Tris-HCl [pH 8], 200 mM NaCl, 1 mM EDTA, 0.5 mM EGTA) and lysed in Lysis buffer 3 (10 mM Tris-HCl [pH 8], 100 mM NaCl, 1 mM EDTA, 0.5 mM EGTA, 0.1% Na-Deoxycholate, 0.5% N-laurylsarcosine). Protease inhibitors (Protease cocktail inhibitor and PMSF (phenylmethane sulfonyl fluoride) were added to all lysis buffers. Chromatin was sheared IP was performed as above. The immune complexes, for both tissues and cells, were recovered using magnetic Dynabeads (Protein A or G Invitrogen) for 3 h. Then, beads were recovered, washed with low salt wash buffer (0.1% SDS, 2 mM EDTA, 1% Triton X-100, 20 mM Tris-HCl [pH 8], 150 mM NaCl), and high salt wash buffer (0.1% SDS, 2 mM EDTA, 1% Triton X-100, 20 mM Tris-HCl [pH 8], 500 mM NaCl) respectively, then suspended in elution buffer (50 mM Tris-HCl [pH 8], 10 mM EDTA, 1% SDS). Eluted beads were treated with 0.2 mg/ml RNase cocktail, then 0.2 mg/ml proteinase K (Invitrogen) at 65 °C for decross-linking overnight. DNA was extracted and purified using Qiagen kit (Cat. 28106) for library preparation or RT-qPCR using specific primers (Dataset EV1).

For EZH1α and SUZ12 ChIP in C2C12, we performed double-cross-linked ChIP-seq. Briefly, sh*Ezh1*[BirA] and sh*Ezh1*[BirA+EZH1α-Avi] cells were firstly cross-linked with 2 mM disuccinimidyl glutarate (Thermo Scientific) while rotating for 45 min at room temperature

(25 °C) and then with 1% formaldehyde (methanol-free, Thermo Scientific) for a further 15 min. Reactions were quenched by addition of 125 mM glycine. All other following lysis steps were performed as previously described in the previous paragraph. In particular, 300 μg sheared chromatin from sh*Ezh1*[BirA+EZH1α-Avi] was used for SUZ12 antibody incubation at 4 °C O/N. The immune complexes were recovered using magnetic Protein G Dynabeads (Invitrogen) for 3 h. For EZH1α ChIP, 300 μg sheared chromatin from both sh*Ezh1*[BirA] and sh*Ezh1*[BirA+EZH1α-Avi] were used to incubate with Dynabeads™ MyOne™ Streptavidin T1 beads to immunoprecipitate biotinylated EZH1α by incubation at 4 °C O/N. Subsequent washing, elution, and de-crosslinking steps were carried out as previously described to extract DNA for library preparation.

For H3K27me3 and SUZ12 ChIP-seq, 1 μg spike-in antibody (Active motif #61686) and 25 ng spike-in chromatin (Active motif #53083) were used for ChIP assay according to the manufacturer's guidelines.

### TT-seq, total RNA-seq, and ChIP-seq library preparation

TT-seq was performed as described with minor modifications[50]. Specifically, two biological replicates of C2C12 cells were labeled for 10 min with 500 μM of 4-thiouridine (4sU, Sigma) at 37 °C and 5% $CO_2$. Each replicate was grown on three Petri dishes with a diameter of 15 $cm^2$. The cells from each plate were lysed and collected with 5 ml of Qiazol Lysis Reagent (Qiagen, # 79306), and lysates belonging to the same replicate were pooled and stored at −80 °C. Before total RNA extraction, 60 μl mixture of 6 ng/μl spike-in mix was added into thawed lysates according to the manufacturer's protocol. Spike-in sequences and their synthesis has been reported previously (Schwalb et al, 2016). In total, 600 μg of RNA in batches of 200 μg of RNAs were sonicated to obtain fragments of <6 kb using AFA micro tubes in Covaris S220 Focused-ultrasonicator. 4sU-labeled RNA was purified from 200 μg batches of the fragmented RNAs. Biotinylation and purification of 4sU-labeled RNAs was performed as described (Schwalb et al, 2016). Libraries for sequencing were prepared using 100 ng of labeled RNA (for both, total and labeled RNA-seq libraries) and TruSeq Stranded Total RNA Library Prep Gold (Illumina, #20020598). Whole transcriptome sequencing was performed using Novaseq6000 platform (100 bp, pair end (PE)), while TT-seq sequencing was done with Hiseq X platform (150 bp PE).

The extracted 20 ng ChIP-DNA (using ChIP-DNA Clean & Concentrator, Zymo) was processed for ChIP-seq high-throughput library preparation (NEBNext® Ultra™ II DNA Library Prep Kit, NEBNext® Multiplex Oligos for Illumina® (Dual Index Primers Set 1), NEB). The ChIP libraries were purified using Ampure XP beads and were quantified using qubit dsDNA HS assay kit. ChIP-seq sequencing was performed using Novaseq6000 platform (150 bp PE).

### Total RNA-seq and TT-seq bioinformatics analysis

The Artificial Intelligence RNA-seq Software as a Service (SaaS) platform (https://transcriptomics.cloud) was used to analyze the total RNA-seq data (Vara et al, 2019). AIR accepts raw next-generation sequencing Illumina FASTQ data as input, performs the quality check and trimming steps with online source BBDuk (minimum read length: 35, minimum Phred quality score: 20) (https://jgi.doe.gov/data-and-tools/bbtools/bb-tools-user-guide/bbduk-guide/) and maps against the

reference genome with STAR (end-to-end mode) (Dobin et al, 2013). Afterward, strand-specificness is assessed by RSeQC (Wang et al, 2012) and gene expression quantification is performed with feature-Counts (Liao et al, 2014). DESeq2 (Love et al, 2014) was the algorithm chosen in the platform for differential gene expression analysis. The genome version used was mm10 and the gene annotation for expression quantification was GENCODE vM23. Differentially expressed genes were those with an FDR lower than 0.05. For the identification of genes following a circadian pattern, the algorithm JTK CYCLE (version 3.1) (Hughes et al, 2010) was used with TMM-normalized read counts as input.

For the analysis of TT-seq data, a custom reference genome including both the mouse genome (version mm10) and the spike-in sequences (ERCC-00043, ERCC-00170, ERCC-00136, ERCC-00145, ERCC-00092-5, ERCC-00002) was required. The data initially underwent a quality check and trimming step with BBDuk (version December 10, 2015) (minimum read length: 35, minimum Phred quality score: 20) (https://jgi.doe.gov/data-and-tools/bbtools/bb-tools-user-guide/bbduk-guide/). The quality-checked reads were then mapped against the reference genome with STAR (version 2.7.3a) (end-to-end mode) (Dobin et al, 2013). The strand specificity was assessed by RSeQC (version 3.0.1) and gene expression quantification was performed with featureCounts (version 2.0) (option -t gene) (Liao et al, 2014). DESeq2(Love et al, 2014) was used for differential gene expression analysis, only considering the 4s-UTP labeled spike-in (ERCC-00043, ERCC-00136, ERCC-00092-5) for the size-factor estimation and the antisense bias corrected counts as input. The antisense bias correction was carried out, as described previously (Michel et al, 2017). The gene annotation for expression quantification was GENCODE vM23. Differentially expressed genes were those with an FDR lower than 0.05.

In both total RNA-seq and TT-seq data, for metaplot analysis and generation of bigWig files, the software deepTools (version 3.4.3) (Ramirez et al, 2014) was used.

## ChIP-seq bioinformatics analysis

Raw data was quality-checked and trimmed with BBDuk (version December 10, 2015) (https://jgi.doe.gov/data-and-tools/bbtools/bb-tools-user-guide/bbduk-guide/) with a minimum read length of 35 bp and a minimum Phred quality score of 25. Since the datasets contained a spike-in from *Drosophila melanogaster*, the high-quality reads were mapped against the mouse genome (version mm10) and against the *D. melanogaster* genome (version BDGP6) with the software BWA (version 0.7.17) (Li and Durbin, 2009). The reads were classified as mouse-derived or Drosophila-derived according to the algorithm disambiguate (Ahdesmaki et al, 2016). Afterward, from the mouse-derived mapping reads, PCR duplicates were removed with Picard MarkDuplicates (version 2.8.1) (http://broadinstitute.github.io/picard/) and only reads being properly paired with a mapping quality equal or higher than 30 were kept with SAMtools (version 1.7) (Li et al, 2009). Peak calling was performed using MACS2 (version 2.2.4) (Zhang et al, 2008) using a $q$-value cutoff of 0.05. Peak annotation was carried out using the R packages ChIPseeker (version 1.26.2) (Yu et al, 2015).

## Immunofluorescence staining and quantification

To precisely quantify relative distribution of EED between cytoplasm and nucleus, synchronized C2C12 myotube cells at the indicated time

point were fixed with 4% formaldehyde in PBS for 10 min at 4 °C, followed by permeabilization with 0.5% Triton X-100 in PBS for 10 min. Cells were then blocked for nonspecific binding with 3% BSA solution for an hour at room temperature, and incubated with the indicated antibody (1:200 dilution for EED antibody; 1:200 dilution for MyHC antibody) for 1 h at room temperature, followed by incubation with Alexa Fluor-labeled secondary antibody (Alexa Fluor 488 and 568 for EED and MyHC respectively) for 45 min at room temperature. Coverslips were mounted on slides using anti-fade mounting medium with DAPI. Immunofluorescence images were acquired on a Zeiss LSM710 confocal microscope with 100X oil-immersion objective. For each channel, all images were acquired with the same settings. Myotube and the nuclear area was defined using MyHC and DAPI signal, and selected for ROI (Region of Interest) to represent total and nuclear fraction, respectively. Fluorescence-based quantitative analysis of nucleocytoplasmic signal was calculated using ImageJ software.

## Super-resolution imaging and tcPALM analysis of RNA Pol II clustering dynamics

In order to generate stable expression of the Dendra2-RPB1Amr construct in our cell lines, Wild Type (WT) C2C12 cells, *Ezh1* knockdown cell line (sh*Ezh1*) and Rescue cell line (sh*Ezh1*^EZH1α) were transfected with the piggyback vector (PB53x EF1-Dendra2-RPB1Amr, Addgene#81228) along with a plasmid expressing the super Piggybac Transposase using an Amaxa nucleofector (Lonza). Cells expressing Dendra2-RPB1-Amr were then selected with G418 (1.2 mg/ml, GIBCO) and α-Amanitin (2 μg/mL, Sigma-Aldrich) for 2 weeks, starting 2 days post-transfection. After the 2-week drug-selection period, positive cells (green fluorescence from Dendra2) were sorted using flow cytometry.

For STORM imaging of Dendra2-RPB1Amr in C2C12 cells, synchronization of stable C2C12 cell lines expressing Dendra2-RPB1Amr was performed as described previously (Peek et al, 2015). The cells at indicated time points were washed twice with DPBS and transferred into FluoroBrite™ DMEM (GIBCO) containing 10% FBS (GIBCO) before image acquisition. We acquired images using a custom-built setup (Abadi et al, 2018) with minor modifications (Cho et al, 2018). Briefly, cells were illuminated with 1.3 W cm⁻² near-UV light (405 nm) for photoconversion of Dendra2 and 3.2 kW cm⁻² (561 nm) for fluorescence detection with an exposure time of 10 ms. We acquired a total of 10,000 frames of fluorescence images of Dendra2-RPB1Amr (image acquisition time of 100 s) to conduct tcPALM analysis. Super-resolution images were reconstructed using Localizer software (Dedecker et al, 2012). The number of bursts detected by using a maximum blinking cutoff of 75 frames and below were less than 10 bursts. This number of bursts was not enough for a reliable fitting of the data to the cumulative distribution function. Thus, we choose 100 frames as a maximum blinking cutoff for our analysis. To distinguish small clustering of RNA Pol II from fluorescence blinking or intensity fluctuations of Dendra2, we set the maximum blinking cutoff at 100 frames. We then collected the correlation time obtained after fitting to the cumulative distribution function with R-squared values ≥ 0.98. As shown in Fig. S12A, the events due to single molecule photo physics in fixed cells showed R-squared values below the set threshold of 0.98. We normalized the data based on the number of bursts where we considered 2500 bursts in in live and in fixed cells to calculate the correlation time. As shown in Fig. S12b, our

analysis (maximum blinking cutoff = 100 frames and R-squared value ≥ 0.98) showed that the correlation time obtained from live cells can be distinguished from fixed cells by considering correlation time values below 20 s. To further remove any contribution of the photophysical phenomenon, we removed the correlation time values if their frequencies is smaller than 20% of the maximum frequency. Thus, we only considered the frequencies that lie in the cyan-shaded area for further analysis.

## Antibodies

The following antibodies were used for western blot or Immuno-fluorescence assay: anti-BMAL1 (Abcam, ab93806); anti-PER2 (Alpha Diagnostic, PER21-A); anti-EZH1(Active Motifs, 61583); anti-EZH1α (Abcam, 64850); anti-EZH1β (homemade antibody provided by Orlando's lab, described previously (*39*)); anti- SUZ12 (Cell Signaling, DR9F6 XP); anti-EED, clone AA19 (Millipore, 05-1320); Actin (Santa Cruz, Sc-47778); anti-Histone-3 (ab1791, Abcam); anti-p84 (Abcam, ab487), Anti-HA (Roche, 3F10), Streptavidin-HRP Conjugate (Cytiva, RPN1231). For ChIP and protein IP: Dynabeads™ MyOne™ Streptavidin T1 beads were used for bio-EZH1 ChIP; anti-H3K27me3 (Millipore, 07-449); anti-SUZ12 (Cell Signaling, DR9F6 XP); Anti-H3K4me3 (Millipore, 7473); anti-BMAL1 (Abcam, 93806); RNA Pol II antibody; unmodified (8WG16); RNA Pol II S5P (ab5131, Abcam) phospho-serine-2 (S2P) (ab24758); EED (Merck, 17-663).

## Statistical analysis

Data are expressed as mean ± SEM and mean ± SD. The significance of differences was analyzed by Student's *t* test or one-way ANOVA for multiple group comparison. The test was considered significant with a *P* value < 0.05. A circadian significance test was performed using the BIO_CYCLE test (Agostinelli et al, 2016), a system to robustly estimate which signals are periodic in high-throughput circadian experiments, producing estimates of amplitudes, periods, phases, as well as several statistical significance measures (http://circadiomics.igb.uci.edu). The test was considered significant with a *P* value < 0.05.

## Data availability

The RNA-seq, TT-seq, and ChIP-seq data have been deposited in the SRA with PRJNA774105. All proteome data have been deposited in PRIDE/ProteomeXchange with accession number: PXD035453. All image source data have been deposited under BioImages accession number S-BIAD1317.

The source data of this paper are collected in the following database record: biostudies:S-SCDT-10_1038-S44318-024-00267-2.

## Peer review information

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

## Acknowledgements

The authors would like to begin by dedicating this paper to the memory of Paolo Sassone-Corsi, an immensely inspiring, heartful colleague and friend. The authors are grateful for the support received from Hounjun Wang (NIAMS, NIH) and Vittorio Sartorelli (NIAMS, NIH), who provided the plasmid and offered valuable suggestions for EZH1 ChIP-seq. The authors also extend our appreciation to Kevin Koronowski (UCI) and Jacob Smith (UCI) for their insightful comments on the manuscript. The authors would like to express our thanks to Huoming Zhang (KAUST Core Lab) for providing assistance with proteome sequencing and analyses. We express our gratitude to Guoxin Cui (KAUST) for generating the ternary plot. In addition, we would like to acknowledge Riccardo Aiese Cigliano (Sequentia Biotech) for his guidance in analyzing RNA-seq, TT-seq, and ChIP-seq data. The authors are grateful for support from KAUST-UCI Strategic Research Joint Program "Epigenetics". (VO and PSC). KAUST baseline grant BAS/1/1037-01-01 (VO). KAUST Competitive Research Grant (CRG) URF/1/1042-01-01 (VO and PSC). KAUST baseline grant BAS/1/1028-01-01 (SH). NextGenerationEU Italian Ministry of University and Research M4.C2 -PNRR YOUNG MSCA_0000081 iNsPIReD (PT). Grant FORDECYT-PRONACES/15758/2020 from the National Council of Humanities, Science and Technology (CONAHCyT) (LAA). National Institutes of Health grants R01CA244519 and R01CA259370 (SM). KAUST baseline grant BAS/1/1080-01 (ML).

## Author contributions

**Peng Liu**: Conceptualization; Data curation; Methodology; Writing—original draft; Writing—review and editing. **Seba Nadeef**: Conceptualization; Data curation; Methodology; Writing—original draft; Writing—review and editing. **Maged F Serag**: Data curation; Methodology; Writing—review and editing. **Andreu Paytuví-Gallart**: Data curation; Writing—review and editing.

**Maram Abadi**: Data curation; Writing—review and editing. **Francesco Della Valle**: Data curation; Writing—review and editing. **Santiago Radio**: Data curation; Writing—review and editing. **Xenia Roda**: Data curation. **Jaïr Dilmé Capó**: Data curation. **Sabir Adroub**: Data curation. **Nadine Hosny El Said**: Data curation. **Bodor Fallatah**: Data curation. **Mirko Celii**: Data curation. **Gian Marco Messa**: Data curation. **Mengge Wang**: Data curation; Writing—review and editing. **Mo Li**: Supervision; Funding acquisition; Writing—review and editing. **Paola Tognini**: Data curation; Supervision; Funding acquisition; Writing— review and editing. **Lorena Aguilar-Arnal**: Data curation; Supervision; Writing— review and editing. **Satoshi Habuchi**: Supervision; Funding acquisition; Writing —review and editing. **Selma Masri**: Supervision; Writing—review and editing. **Paolo Sassone-Corsi**: Conceptualization; Supervision; Funding acquisition. **Valerio Orlando**: Conceptualization; Supervision; Funding acquisition; Writing —original draft; Project administration; Writing—review and editing.

Source data underlying figure panels in this paper may have individual authorship assigned. Where available, figure panel/source data authorship is listed in the following database record: biostudies:S-SCDT-10_1038-S44318-024-00267-2.

## Disclosure and competing interests statement

The authors declare no competing interests.

