## [Peer Review File · The EMBO Journal]

PRC2- EZH1 contributes to circadian gene expression by orchestrating chromatin states and RNA polymerase II complex stability

Valerio Orlando, Peng Liu, Seba Nadeef, Maged Serag, Andreu Paytuví-Gallart, Maram Abadi, Francesco Della Valle, Santiago Radio, Xenia Roda, Jaïr Dilmé Capó, Sabir Adroub, Nadine Hosny El Said, Bodor Fallatah, Mirko Celii, Gian Marco Messa, Mengge Wang, Mo Li, Paola Tognini, Lorena Aguilar Arnal, Satoshi Habuchi, Selma Masri, and Paolo Sassone-Corsi

Corresponding author(s): Valerio Orlando (valerio.orlando@kaust.edu.sa) , Peng Liu (peng.liu@kaust.edu.sa)

Review Timeline:

Submission Date:	2nd Jun 24
Editorial Decision:	24th Jul 24
Revision Received:	4th Sep 24
Accepted:	27th Sep 24

Editor: Cornelius Schneider

Transaction Report:

Please note that the manuscript was transferred from another journal where it was originally reviewed. Since the original reviews are not subject to EMBO's transparent review process policy, they cannot be published.

1st round peer review

REVIEWER COMMENTS

Reviewer #1 (Remarks to the Author):

In this manuscript, entitled “EZH1 Orchestrates Rhythmicity of Circadian Gene Expression Acting on Transcription Machinery Integrity”, the authors present work suggesting that EZH1 plays an important role in circadian clock regulation in post mitotic skeletal muscle cells. More specifically, the authors suggest that EZH1 can act as either a positive or negative modulator of circadian gene expression.

For the convenience of reviewers, we have included figures in the rebuttal letter, some of which are also presented in our revised manuscript or supplemental files. To distinguish these figures, we have highlighted them with different colors when referencing them.

Major points:

1. The authors should use two siRNA/shRNAs for each respective gene in the their experiments, as is standard practice. This applies for *Bmall* in Figure 2 and *Ezh1* in Figures 3, 4, 6 and 7.

We thank the reviewer for raising this question. We utilized two additional siRNA targeting *Bmall* and confirmed their knock-down efficiency of *Bmall* through qPCR and WB assay. As evident in supplemental Fig. S2 and also shown below for the convenience of the reviewer, our results were consistent with previous findings, revealing that the rhythmicity and amplitude of *Ezh1 α* , *Suz12*, and *Eed* were disrupted upon *Bmall* down-regulation using different siRNAs (Fig.1) [revised Supplemental Fig.S2]. These data collectively support the conclusion that BMAL1 plays a consistent role in modulating the circadian expression of *Ezh1 α* , *Suz12*, and *Eed*.

In our previous manuscript, we demonstrated the specificity of *Ezh1* knock-down by restoring *Ezh1* expression constitutively in the *shEzh1* knock-down background. Indeed, another study has also used this approach to overcome potential off-target effects from the shRNA strategy¹. However, repeating all circadian experiments in Fig. 3, 4, 6, and 7 with additional siRNA/shRNA is challenging. To address the reviewer's concern, we generated a new rescue cell line, *shEzh1*^{TET-EZH1 α} , by expressing EZH1-HA fusion construct driven by the Tetracycline-Responsive Element (TRE) promoter. Both transcript and protein analyses confirmed that the levels of the exogenous EZH1-HA were similar to the endogenous EZH1 version (Fig.2a, b, d) [revised Supplemental Fig.S10a, b, d]. Subsequently, we repeated the TT-seq and measured the nascent transcriptome features of WT, *shEzh1* knock-down, and *shEzh1*^{TET-EZH1 α} after synchronization 16h, a time point when nascent transcription exhibits its highest level. We found that the majority of differentially regulated genes could be restored to WT due to the inducible expression of EZH1 α (Fig.2e, f) [revised Supplemental Fig.S10e, f]. These new rescue experiments further confirm the direct and specific role of EZH1 in regulating transcription, unveiling a specific role in RNA Pol II initiation complex stability. We believe this alternative approach offers a practical solution instead of using more siRNA/shRNAs targeting *Ezh1*, which would be challenging for studying circadian rhythms under different conditions. We hope the reviewer will appreciate this approach.

Figure 1. Efficiency of siRNAs targeting *Bmal1*. a-b RT-qPCR and immunoblotting of knock-down efficiency of *Bmal1*/BMAL1 in C2C12 treated with different siRNAs against *Bmal1* during the circadian period. Error bars represent \pm SEM from three independent experiments. c-e Circadian pattern of *Ezh1* α , *Suz12* and *EED* was measured from synchronized C2C12 cells treated with scramble or siRNAs (#2 and #3) targeting *Bmal1* at indicated time points and analyzed by quantitative real-time PCR using specific oligos (n= 3 per time point).

Figure 2. Expression levels of differentially expressed genes were restored by transiently inducing the expression of EZH1α. **a** The protein levels of Ezh1α were assessed through Western blot (WB) analysis in various C2C12 cell lines, including WT, *shEzh1*, and *shEzh1*^{TET-EZH1α} (treated with DMSO or DOX) transgenic cells. In the WB analysis, the band observed in the WT lane corresponds to the endogenous EZH1α protein. In contrast, the band in the DOX lane represents the HA-EZH1α. **b** Different cell lines were utilized to immunoprecipitate the HA-EZH1α protein using HA-conjugated agarose beads. Subsequently, the HA-tagged proteins were visualized using anti-HA. **c** Experimental design illustrating TT-seq samples collection at indicated conditions. **d** FPKM value of EZH1 in WT, *shEzh1*^{TET-EZH1α} treated with DMSO (DMSO), and *shEzh1*^{TET-EZH1α} treated with DOX (DOX). **e** Absolute fold changes of significantly differential genes in comparison with indicated biological conditions at 16h. **f** Volcano plot representation of differential expression profile of nascent transcripts in comparison with different biological conditions at 16h.

2. Central to the paper is the finding that *Ezh1* plays a rhythmic role in regulating both repressive and active state of circadian gene expression pattern through H3K27me3 and RNA Pol II regulation. However, the expression changes observed following *Ezh1* knockdown could be due to secondary effects of the shRNA knockdown. As such, it is imperative that the authors show direct binding of EZH1 (and preferably also other core PRC2 components such as either SUZ12 or EED) at their cluster 1 and cluster 2 genes using ChIP-seq across their time course and conditions. While, an attempt to define the cluster 1 genes as Polycomb target genes were made in Figure 4. Some improvements are needed. For example, they used wild-type, non-synchronized C2C12 cells and not the EZH1 knockdown synchronized cells, which would have been more informative. In Figure 4, no positive or negative controls are included to give confidence in their ability to map EED and EZH1. For example, the authors should feature all clusters of genes in panel b (Cluster 1, 2 negative control genes etc etc) and show a more zoomed out view of the *Hoxc13* locus in Figure 4d to demonstrate the specificity of their ChIP-seq.

We appreciate the reviewers' concerns regarding the potential secondary effects of shRNA knockdown and their suggestions to perform more comprehensive ChIP-seq experiments across different time courses.

To address the concern, we generated a new rescue cell line, *shEzh1*^{TET-EZH1 α} , where we expressed an EZH1-HA fusion construct driven by the Tetracycline-Responsive Element (TRE) promoter. As we described in replying to Point 1, our transcript and protein analyses confirmed that the levels of exogenous EZH1-HA were similar to the endogenous EZH1 version. TT-seq experiments by measuring the nascent transcriptome features of wild-type (WT), *shEzh1* knockdown, and *shEzh1*^{TET-EZH1 α} cells after synchronization at the 16-hour time point proved that down-regulated genes could be restored to WT levels due to the inducible expression of EZH1 α . These new rescue experiments provide further confirmation of the direct and specific role of EZH1 in regulating nascent gene expression (Fig.2) [revised Supplemental Fig.S10]. Additionally, in total RNA-seq analyses focusing on common circadian genes, we observed that the altered amplitude caused by *Ezh1* knockdown could be rescued back to WT levels [revised Fig.3d]. Similar findings were observed when examining cluster I and cluster II genes in total RNA-seq, where constitutive rescue of EZH1 also restored the differential transcriptome signature [revised Fig.3g, h]. These data strongly support the specific and direct function of EZH1. While we cannot fully exclude the possibility of some secondary effects introduced upon *Ezh1* depletion, we did observe that many genes regained their circadian profile, which is unique to the *shEzh1* background [revised Supplemental Fig.S6].

To further consolidate the direct contribution of EZH1 to gene expression, we adopted the modified biotin-mediated ChIP (bioChIP-seq) system shared with us by Prof. Sartorelli (NIH). This approach involves the biotinylation of EZH1-Avi-tag through co-expression with BirA ligase endogenously in the same cell line (Fig.3) [revised Supplemental Fig.S13]. This method has been successfully utilized by various research groups to generate high-quality EZH1 ChIP-seq datasets in different model systems^{1,2}. Following this strategy, we generated new EZH1 ChIP-seq data using synchronized cells at different circadian time points. Additionally, we also performed SUZ12 and H3K27me3 ChIP-seq experiments across different circadian time points, allowing us to investigate the occupancy profiles of these key Polycomb group (PcG) factors and their modifications in conjunction with EZH1 throughout the circadian cycle (Fig.4a-c) [revised Fig.7a-c].

In our initial manuscript, we identified cluster 1 and cluster 2 genes as potential targets of EZH1 for both repression and activation based on total RNA-seq data. However, since total RNA-seq primarily captures the mature state of RNA, encompassing various regulatory processes such as transcription, post-transcriptional modifications, and RNA turnover, it may not fully reflect the direct impact of EZH1 knockdown. To address this, we performed TT-seq analysis at different time points (12h, 16h, and 24h) to specifically examine the cyclic nascent transcription process. Through the TT-seq analysis, we identify potential activation and repression targets of EZH1. Notably, we observed significant gene expression peaks at 16h (referred to as TT-seq 16h peak) and troughs at 24h (referred to as TT-seq 24h trough) during the 24-hour circadian cycle. These peaks and troughs were considered representative of the genes activated and repressed by EZH1, respectively, in a more direct and time-dependent manner compared to the total RNA-seq data (Fig.4d, e) [revised Fig.7d, e].

Finally, we conducted an intersection analysis to compare the ChIP-seq data of EZH1 α , SUZ12, and H3K27me3 with the TT-seq 16h peak and 24h trough genes. This analysis unveiled that a significant portion of the rhythmic genes bound by EZH1, both in terms of repression and activation, were also co-bound by SUZ12 and H3K27me3. Specifically, among the EZH1 α -bound TT-seq 16h targets, 75 out of 97 genes exhibited co-occupancy with SUZ12

and H3K27me3. Similarly, among the EZH1 α TT-seq 24h targets, 90 out of 128 genes showed co-occupancy with SUZ12 and H3K27me3 (**Fig.4f, g**) [**revised Fig.7f, g**]. Interestingly, a careful examination of the binding profiles of EZH1, SUZ12, and H3K27me3 around these activation targets revealed a distinct pattern compared to the repressive targets. Notably, H3K27me3 deposition occurred around the transcription termination site (TTS) region for repressive targets, while it was more enriched within the gene body for activation targets (**Fig.4h-m**) [**revised Fig.7h-m**]. Moreover, H3K27me3 levels were partially depleted around the transcription start site (TSS) of activation targets, suggesting the involvement of other demethylation processes in establishing this profile. These distinct binding profiles of PcG proteins, including EZH1 and SUZ12, along with the distribution of H3K27me3, indicate that the relative levels of H3K27me3 around the TSS region play a crucial role in modulating RNA Pol II activity for both activation and repression targets regulated by EZH1.

As suggested by the reviewer, we have included an IGV screenshot of the binding profiles of EZH1, SUZ12, and H3K27me3 around the *Hoxd* cluster genes based on our new ChIP-seq data across different time points. The observed co-bound profile of these different epigenetic marks around the canonical Polycomb group (PcG) targets provides strong evidence of the specificity of our ChIP-seq experiments (**Fig.3f**) [**revised Supplemental Fig. S13f**].

Figure 3. Genomic features of EZH1 α , SUZ12, and H3K27me3 ChIP-seq. **a** Protein levels of Ezh1 α were assessed through Western blot (WB) analysis in various C2C12 cell lines, including WT, *shEzh1*, *shEzh1*^{BirA}, and *shEzh1*^{BirA+EZH1 α -Avi} transgenic cells. In the WB analysis, the band observed in the WT lane corresponds to the endogenous EZH1 α protein. In contrast, the band in the BirA+EZH1 α -Avi Lane represents the biotinylated form of EZH1 α . **b** Different indicated cell lines were utilized to immunoprecipitate the biotinylated EZH1 α protein using streptavidin resin. Subsequently, the biotin-labeled proteins were visualized using Streptavidin-HRP. **c-e** Genomic features were annotated for the ChIP-seq peaks of EZH1 α (c), SUZ12(d), and H3K27me3(e). **f** IGV track of EZH1 α , SUZ12, and H3K27me3 peak distribution around *Hoxd* genes cluster loci. **g-h** Gene ontology (GO) enrichment analysis of genes that expressed with the highest level (peak) or lowest level (trough) at 16h (g) and 24h (h).

Fig. 4: Differential Binding Patterns of EZH1 α , SUZ12, and H3K27me3 at Activated and Repressed Circadian Targets. a-c Heatmap showing EZH1 α (a), SUZ12 (b), and H3K27me3 (c) ChIP-seq signal around transcription start site (TSS), gene body, and transcription termination site (TES) in wild-type (WT) synchronized C2C12 myotube cells

at indicated time point. **d-e** Clustering of genes expressed with the highest level (peak) or lowest level (trough) at 16h (**d**) and 24h (**e**), as determined by TT-seq in wild-type (WT) synchronized C2C12 cells. **f-g** Upset plot illustrating the intersection of EZH1 α , SUZ12, and H3K27me3 co-occupancy status around genes that exhibit peak expression at 16 hours (**f**) or trough expression at 24 hours (**g**), as determined in panel (**d**) and (**e**). **h-j** Metaplot analyses of EZH1 α (**h**), SUZ12 (**i**), and H3K27me3 (**j**) ChIP-seq peak signals at various time points surrounding the transcription start sites (TSSs) of genes exhibiting the highest expression peak at 16 hours (peak at 16h), as determined by TT-seq in wild-type (WT) synchronized C2C12 cells. **k-m** Metaplot analyses of EZH1 α (**k**), SUZ12 (**l**), and H3K27me3 (**m**) ChIP-seq peak signals at various time points surrounding the transcription start sites (TSSs) of genes exhibiting the lowest expression peak at 24 hours (trough at 24h), as determined by TT-seq in wild-type (WT) synchronized C2C12 myotube cells.

3. There is no attempt made by the authors to evaluate if EZH1 does or doesn't bind at the cluster 2 genes, in line with the claimed activating role of EZH1 at these targets. The authors refer to a prior study (Mousavi et al. 2012: PMID: 22196887) that reported EZH1 binding at active genes – however, it is important to address that that paper remains controversial as the specificity of the EZH1 antibody used is questionable. At minimum, the authors here should show if EZH1 and EED bind at Cluster 2 genes. My view, based on reading many published studies profiling PRC2 in various cell types, is that PRC2 does not bind at active genes.

We appreciate the reviewer's comment regarding the investigation of direct EZH1 binding to activated circadian targets. In our initial manuscript, we identified cluster 1 and cluster 2 genes based on total RNA-seq data, which served as representative examples of rhythmic targets showing both repression and activation. It was hypothesized that these genes might also be bound by EZH1 α . However, total RNA-seq primarily captures the mature RNA state, encompassing multiple regulatory processes such as transcription, post-transcriptional modifications, and RNA turnover. As a result, it may not fully reflect the direct and primary role of EZH1 α in the transcriptional process.

Based on the IP-MS anticipating a role in RNA Pol II initiation complex stability, we conducted TT-seq analysis at different time points (12h, 16h, and 24h) to specifically examine cyclic nascent transcription. In response to Point 2, we have provided a detailed description outlining the redefinition of potential rhythmic targets of EZH1 α involved in both activation and repression, as determined by TT-seq. Specifically, we selected candidate genes that exhibited significant peaks at 16h (referred to as TT-seq 16h peaks) and/or significant troughs at 24h (referred to as TT-seq 24h troughs) during the 24-hour circadian cycle. The selection of genes with peaks at 16h and troughs at 24h was based on the observed peak interactions between EZH1 α and the RNA Pol II machinery or other PRC2-EZH1 α components at these respective time points. Gene Ontology (GO) analyses of these two sets of targets revealed their involvement in distinct biological pathways, suggesting the dynamic nature of their regulation (**Fig.3g, h**) [**revised Supplemental Fig. S13g, h**]. Collectively, these identified peaks and troughs were considered indicative of genes activated and repressed by EZH1 in a more direct and time-dependent manner compared to the total RNA-seq data.

To address the reviewer's concerns regarding the specificity of the EZH1 antibody used in our study, we conducted pilot ChIP-seq experiments using alternative commercial EZH1 antibodies in both wild-type (WT) and *shEzh1* knockdown (KD) cells. Unfortunately, we found that the efficiency and specificity of the tested antibodies were not sufficient for generating high-quality EZH1 α ChIP-seq data. Additionally, we sought advice from Prof. Vittorio Sartorelli (NIH), who confirmed that, based on their experience and as mentioned in their latest work³, there are no other commercially available EZH1 antibodies suitable for ChIP-seq experiments. To overcome this challenge, we devised a solution by adopting the modified biotin-mediated ChIP (bioChIP-seq) system, which was kindly shared by Prof. Sartorelli. In this bioChIP-seq system, we employed the biotinylation of EZH1 α -Avi-tag through co-expression with BirA ligase endogenously in the same cell line. This approach has been successfully utilized by various research groups to generate high-quality EZH1 α ChIP-seq datasets in different model systems^{1,2}. Following this strategy, we have generated new EZH1 α ChIP-seq data using synchronized WT C2C12 myotube cells at different circadian time points. Additionally, we have also generated SUZ12 and H3K27me3 ChIP-seq data across different circadian time points, allowing us to investigate the occupancy profiles of these key PcG factors and their modification in conjunction with EZH1 throughout the circadian cycle (**Fig. 4a-c**) [**revised Fig.7a-c**].

Rather than overlapping with cluster 2 genes from total RNA-seq data, we performed an intersection analysis to compare the ChIP-seq data of EZH1 α , SUZ12, and H3K27me3 with the TT-seq 16h peak genes. This analysis unveiled that a significant portion of the rhythmic genes bound by EZH1 (**Fig.5a, b**) [**revised Supplemental Fig.S14a, b**]. Subsequently, we investigated whether the occupancy of EZH1 on these rhythmic targets exhibits a time-dependent

pattern. We specifically examined the peak intensity around the transcription start site (TSS) spanning one nucleosome (± 147 bp) for both activation and repressive targets. However, we did not observe significant rhythmic changes in the binding intensity (Fig. 5c, d) [revised Supplemental Fig.S14c, d]. As explained in point 2, a careful examination of the binding profiles of EZH1, SUZ12, and H3K27me3 around these activation targets revealed a distinct pattern compared to the repressive targets. These distinct binding profiles of PcG proteins, including EZH1 and SUZ12, along with the distribution of H3K27me3, indicate that the relative levels of H3K27me3 around the TSS region play a crucial role in modulating RNA Pol II activity for both activation and repression targets regulated by EZH1.

To further discuss the role of EZH1 α in gene activation or repression, it is important to mention that we do not negate the repressive function of EZH1. PRC2-EZH1 complex still plays a critical role in mediating the deposition of H3K27me3 around canonical Polycomb group (PcG) targets, thereby repressing their expression. Instead, we highlight the diverse function of EZH1 by providing additional mechanistic evidence to demonstrate its involvement in gene activation processes. Numerous studies have reported similar activation roles for EZH1 α in various cell types. For instance, EZH1 has been observed to associate broadly with active H3K4me3 mark^{1,2,4}. Additionally, the subcomplex EZH1 α -SUZ12 has been identified as a crucial complex mediating this activation function^{4,5}. In support of these findings, we also have observed co-occupancy of EZH1 α and SUZ12 at rhythmic activation targets. It is worth noting that EZH1 is not the only Polycomb protein involved in gene activation; other members of the Polycomb group have also been implicated in this process⁶. Collectively, these findings emphasize the significance of cellular context and genetic programs in determining the function of EZH1 and underscore the complexity of Polycomb-mediated transcriptional regulation, highlighting the plasticity of chromatin states and transcriptional processes.

Figure 5. Intersection between EZH1 α bound targets and TT-seq determined rhythmic genes. a-b Venn diagram of overlap between EZH1 ChIP-seq and genes expressed with the highest level (peak) or lowest level (trough) at 16h (a) and 24h (b), as determined by TT-seq in wild-type (WT) synchronized C2C12 cells. c-d Violin plots illustrate the peak intensity of EZH1 ChIP-seq peaks surrounding the transcription start site of genes expressed with the highest level (peak) or lowest level (trough) at 16h (c) and 24h (d) co-bound by EZH1 α , SUZ12 and H3K27me3 encompassing upstream and downstream regions spanning one nucleosome. Each dot on the plot represents the average peak intensity around each gene.

Minor points:

1. The inclusion of a BMAL1 western blot in Figure 1 would be helpful.

We have added the circadian profile of BMAL1 both in skeletal muscle and synchronized C2C12 myotube cells in our revised Figure 1 [revised Fig.1b, e].

2. In Figure 2b, the authors use “Relative BMAL1 binding” for the y-axis instead of percentage input (as in Figure 2c and 2d). This should be changed to percentage input so that the quality and strength of the BMAL1 ChIPs can be appreciated.

Thanks to the reviewer for pointing out this discrepancy; we have corrected the description in our **revised Fig.2** following the reviewer's suggestion.

3. Text in lines 206-212 referencing the Venn diagram in Figure 3c does not clearly or intuitively correlate with the figure thereby making it difficult for the reader. An alternate form of presentation should be used to make these analyses more easily and readily comprehensible.

We have presented this data in another alternative way by specifying the exact number of genes for each category **[revised Fig.3c]**.

4. In Figure 2d, the authors should include some positive controls (well-established Polycomb target genes such as HOX and other developmental genes) for their EZH1 ChIPs to evaluate if the EZH1 binding is or is not oscillating on these other targets.

In our experiment, we incorporated positive controls, namely *Hoxd10* and *Neurog1*, to demonstrate the specificity of EZH1 in our ChIP-qPCR analysis. Moreover, we observed no significant circadian occupancy pattern of EZH1 near the transcription start sites (TSS) of these control targets **[revised Supplemental Fig.S3e, f]**.

Reviewer #2 (Remarks to the Author):

In this manuscript entitled “EZH1 Orchestrates Rhythmicity of Circadian Gene Expression Acting on Transcription Machinery Integrity” by Drs. Orlando and Sassone-Corsi and colleagues, the authors show that the PRC2 components EZH1, SUZ12 and EED, are regulated by the clock transcription factor BMAL1 in skeletal muscle tissue and synchronized C2C12 cells. Interestingly, they also found that EZH1 directly regulates the expression of the core clock machinery. The authors also show that a cluster of 167 genes circadian genes are repressed by EZH1-PRC2. Notably, a cluster of 313 genes seems to be positively regulated by EZH1, and by IP-MS, they show physical interaction with subunits of RNA Pol II and Mediator. Following on IP-MS experiments, they demonstrate that assembly of PRC2-complexes is regulated during the circadian cycle, around 24h/28h post synchronization, and suggest that EED cytoplasm to nucleus shuttling facilitates assembly of PRC2 complexes. Finally, they propose that EZH1 regulates circadian transcription through stabilization of RNA Pol II complex.

I think this is a quite interesting study that is suitable for publication in Nature Communications. The authors did a great deal of work, and almost all conclusions are well supported by the experiments, therefore my comments are mainly curiosity driven. My remarks are only intended to improve the message of the paper, and my recommendation for publication is not contingent upon performing more experiments.

We thank the reviewer for his positive feedback and for considering our study suitable for publication in Nature Communications. We also thank the reviewer for the supportive remarks and comments aimed at improving the manuscript.

For the convenience of reviewers, we have included figures in the rebuttal letter, some of which are also presented in our revised manuscript or supplemental files. To distinguish these figures, we have highlighted them with different colors when referencing them.

Abstract: I think the sentence “the intermingle between epigenome and RNA Pol II rhythmicity remains to be investigated” is not well connected with the following sentence of the abstracts. My impression is that the first question of the study was not this one but if PRC2 subunits have a circadian expression pattern and a role in the circadian rhythm.

Thank you for your comment regarding the connection between the sentence "the intermingle between epigenome and RNA Pol II rhythmicity remains to be investigated" and the following sentence in our abstract. We have now updated the text in the revised version on lines 34-35 as follows:

“However, the role of Polycomb group (PcG) complex PRC2 in coordinating epigenome and RNA Pol II rhythmicity remains to be extensively investigated.”

The observation that EZH1 isoforms have a different protein pattern in figures 1b and 1e is interesting and could be discussed in more depth. Also, what is the pattern of H3K27me3 in the experiments from figure 1b and 1e?

We appreciate the request by the reviewer to discuss in more detail the reported different protein patterns observed in comparison between skeletal muscle tissue and synchronized cells. We have discussed this with further details in our revised manuscript Discussion section, which was shown on lines 477-497 as follows:

“In our current study, we made an intriguing discovery that the EZH1 β protein exhibits a 12-hour rhythm specifically in skeletal muscle tissue. However, this 12-hour rhythm is not observed in cell-autonomous systems. These findings collectively suggest the presence of potential endogenous physiological signals, such as metabolic signals or inter-organ communication signals, in vivo, which are absent in synchronized in vitro cell cultures. Previous reports have indicated that XBP1 acts as a major transcription factor governing the 12-hour cycling pattern⁷, and regulates various lipid and metabolic pathways associated with this rhythm⁸. Interestingly, the expression of Ezh1 β transcripts shows a 24-hour rhythm, suggesting a regulation at the post-transcriptional, translational, or post-translational levels that is independent of XBP1-mediated transcriptional regulation. In our previous study, we demonstrated that ubiquitin E3 ligases, such as HUWE1 and NEDD4⁹ interacts with EZH1 β protein. Ubiquitin-mediated proteasome degradation

has been extensively studied and plays an essential role in regulating protein rhythmic homeostasis¹⁰. It is worth investigating whether ubiquitin-mediated degradation could govern the 12-hour rhythm of EZH1 β in a circadian pattern by sensing potential endogenous signals, such as metabolic signals. However, establishing a direct connection between ubiquitin E3 ligase and EZH1 β through depletion of HUWE1 or NEDD4 would be challenging, given the multiple downstream targets of both HUWE1 and NEDD4. For example, HUWE1 has been shown to be involved in the degradation of REV-ERB α ¹¹. In our rescue cell line *shEzh1^{EZH1 α}* , where EZH1 β is down-regulated, we observed that transcriptional rhythmicity could not be fully restored. This indicates the essential role of EZH1 β in orchestrating circadian gene expression, by modulating the rhythmic shuttle of EED and the assembly of the PRC2-EZH1 repressive complex.”

We have examined the pattern of H3K27me3 in both in vivo and in vitro systems, as shown in revised Supplemental Fig. S2. Interestingly, we did not observe a significant circadian pattern in either system (see Fig. 6 below for convenience). However, our genome-wide H3K27me3 ChIP-seq analysis did identify the occupancy of H3K27me3 marks around rhythmic repressive target genes [revised Supplemental Fig.S14i]. These findings suggest the importance of the rhythmic shuttle of EED that orchestrates the circadian epigenome landscape of H3K27me3 around specific circadian repressive targets.

Figure 6. Global levels of H3K27me3 in skeletal muscle and synchronized C2C12 cells. **c** Levels of H3K27me3 in mouse gastrocnemius muscle tissue were analyzed by immunoblotting. Three independent experiments were performed, and representative data was shown. Global H3 was used as an internal control. **d** Quantitative analysis of protein abundance of H3K27me3 shown in panel (c) in gastrocnemius muscle tissue. Error bars represent \pm SEM from three independent experiments. **e** Levels of H3K27me3 from synchronized C2C12 cells at indicated time points were analyzed by immunoblotting. Three independent experiments were performed, and representative data was shown. Global H3 was used as an internal control. **f** Quantification of protein abundance of H3K27me3 shown in panel (e). Error bars represent \pm SEM from three independent experiments.

If the experiments in figure 1d would be extended to 48h, do the levels of Ezh1 isoforms will go back to 1? Why is the oscillatory effect on Ezh1 much milder in figure 2a than in figure 1?

We appreciate your intriguing question. In original Figure 1d, it is theoretically expected that the oscillatory profile would exhibit similarity during the same synchronized time points if extended to 48 hours. However, it should be noted that we are using differentiated myotubes, and the relative expression of Ezh1 could undergo slight changes

without impacting the final results. This is because *Ezh1* is primarily expressed in post-mitotic tissues, and the duration of myotube differentiation may slightly influence the expression level of *Ezh1*. Since there was no specific need to examine the profile within 48 hours, we did not conduct this particular experiment.

In original Figure 2a, we show an independent experiment conducted in smaller plates to facilitate the use of si-RNA and achieve the knock-down effect. Both experiments (Figure 1d and Figure 2a) involved the collection of differentiated myotubes on MT4-5. However, the process of myotube formation, contraction, and serum synchronization could potentially affect the expression level of *Ezh1* in both experiments. Nevertheless, these slight variations in expression level do not appear to impact the final results, which focus on the oscillatory profile of EZH1.

I did not understand why *Ezh1* recruitment pattern at *Bmal1* is the opposite in muscle tissue and C2C12 cells. Please clarify.

We appreciate the reviewer for bringing this issue to our attention. During the organization of the different panels using Illustrator, there was an inadvertent misuse of the same plot, intended for the *Per2* locus, in place of the *Bmal1* locus. We have rectified this error by replacing the panel with the correct plot [revised Fig. 2c]. Furthermore, to ensure transparency and support our findings, we have provided all the source data used to generate these plots. We sincerely apologize for any confusion caused by this mistake.

The H3K27me3 results in Figure 3a are counterintuitive. Although the authors discuss this result, I am wondering if the problem here is that the synchronization was not perfect (= cellular heterogeneity). Perhaps EZH1-H3K27me3 re-ChIP would be informative. It is not clear why the EED and EZH1 ChIP-seq in figure 4b were performed in non-synchronized cells. I would say that these ChIPs at 0 and 12h post-synchronization would have been more informative. Please explain in greater detail the rationale of the experiment set up.

We appreciate the reviewer for raising this point. We cannot fully exclude the effects of cellular heterogeneity in the myotube formation process. To address whether the effects we observed upon *Ezh1* depletion were due to direct occupancy of EZH1 around the affected loci, (a similar concern has been raised by Reviewer 1), we have conducted EZH1, SUZ12, and H3K27me3 ChIP-seq experiments across all time points.

We conducted pilot ChIP-seq experiments using alternative commercial EZH1 antibodies in both wild-type (WT) and *shEzh1* knockdown (KD) cells. Unfortunately, we found that the efficiency and specificity of the tested antibodies were not sufficient for generating high-quality EZH1 α ChIP-seq data. Additionally, we sought advice from Prof. Sartorelli (NIH), who confirmed that, based on their experience and as mentioned in their latest work³, there are no other commercially available EZH1 antibodies suitable for ChIP-seq experiments. To overcome this challenge, we devised a solution by adopting the modified biotin-mediated ChIP (bioChIP-seq) system, which was kindly shared with us by Prof. Vittorio Sartorelli. In this bioChIP-seq system, we employed the biotinylation of EZH1 α -Avi-tag through co-expression with BirA ligase endogenously in the same cell line. This approach has been successfully utilized by various research groups to generate high-quality EZH1 α ChIP-seq datasets in different model systems^{1,2}. Following this strategy, we have generated new EZH1 α ChIP-seq data using synchronized WT C2C12 myotube cells at different circadian time points. Additionally, we have also generated SUZ12 and H3K27me3 ChIP-seq data across different circadian time points, allowing us to investigate the occupancy profiles of these key PcG factors and their modification in conjunction with EZH1 throughout the circadian cycle (Fig. 7a-c) [revised Fig.7a-c].

In our initial manuscript, we identified cluster 1 and cluster 2 genes as potential targets of EZH1 for both repression and activation based on total RNA-seq data. However, since total RNA-seq primarily captures the mature state of RNA, encompassing various regulatory processes such as transcription, post-transcriptional modifications, and RNA turnover, it may not fully reflect the direct impact of EZH1 knockdown. To address this, we performed TT-seq analysis at different time points (12h, 16h, and 24h) to specifically examine the cyclic nascent transcription process. Through the TT-seq analysis, we identify potential activation and repression targets of EZH1. Notably, we observed significant gene expression peaks at 16h (referred to as TT-seq 16h peak) and troughs at 24h (referred to as TT-seq 24h trough) during the 24-hour circadian cycle. These peaks and troughs were considered representative of the genes activated and repressed by EZH1, respectively, in a more direct and time-dependent manner compared to the total RNA-seq data (Fig. 7d, e) [revised Fig.7d, e].

Finally, we conducted an intersection analysis to compare the ChIP-seq data of EZH1 α , SUZ12, and H3K27me3 with the TT-seq 16h peak and 24h trough genes. This analysis unveiled that a significant portion of the rhythmic genes bound by EZH1, both in terms of repression and activation, were also co-bound by SUZ12 and H3K27me3. Specifically, among the EZH1 α -bound TT-seq 16h targets, 75 out of 97 genes exhibited co-occupancy with SUZ12 and H3K27me3. Similarly, among the EZH1 α TT-seq 24h targets, 90 out of 128 genes showed co-occupancy with SUZ12 and H3K27me3 (**Fig. 7f-g**) [**revised Fig.7f-g**]. Interestingly, a careful examination of the binding profiles of EZH1, SUZ12, and H3K27me3 around these activation targets revealed a distinct pattern compared to the repressive targets. Notably, H3K27me3 deposition occurred around the transcription termination site (TTS) region for repressive targets, while it was more enriched within the gene body for activation targets. Moreover, H3K27me3 levels were partially depleted around the transcription start site (TSS) of activation targets, suggesting the involvement of other demethylation processes in establishing this profile (**Fig. 7h-m**) [**revised Fig.7h-m**]. These distinct binding profiles of PcG proteins, including EZH1 and SUZ12, along with the distribution of H3K27me3, indicate that the relative levels of H3K27me3 around the TSS region play a crucial role in modulating RNA Pol II activity for both activation and repression targets regulated by EZH1.

Figure 7: Differential Binding Patterns of EZH1 α , SUZ12, and H3K27me3 at Activated and Repressed Circadian Targets. a-c Heatmap showing EZH1 α (a), SUZ12 (b), and H3K27me3 (c) ChIP-seq signal around

transcription start site (TSS), gene body, and transcription termination site (TES) in wild-type (WT) synchronized C2C12 myotube cells at indicated time point. **d-e** Clustering of genes expressed with the highest level (peak) or lowest level (trough) at 16h (**d**) and 24h (**e**), as determined by TT-seq in wild-type (WT) synchronized C2C12 cells. **f-g** Upset plot illustrating the intersection of EZH1 α , SUZ12, and H3K27me3 co-occupancy status around genes that exhibit peak expression at 16 hours (**f**) or trough expression at 24 hours (**g**), as determined in panel (**d**) and (**e**). **h-j** Metaplot analyses of EZH1 α (**h**), SUZ12 (**i**), and H3K27me3 (**j**) ChIP-seq peak signals at various time points surrounding the transcription start sites (TSSs) of genes exhibiting the highest expression peak at 16 hours (peak at 16h), as determined by TT-seq in wild-type (WT) synchronized C2C12 cells. **k-m** Metaplot analyses of EZH1 α (**k**), SUZ12 (**l**), and H3K27me3 (**m**) ChIP-seq peak signals at various time points surrounding the transcription start sites (TSSs) of genes exhibiting the lowest expression peak at 24 hours (trough at 24h), as determined by TT-seq in wild-type (WT) synchronized C2C12 myotube cells.

Finally, I think the IP-MS results in figures 5a-c are great and the accumulation of EED at chromatin 24h post synchronization is clear, but the conclusion of the figure is perhaps overstated. Ideally, EED should be depleted to demonstrate that shuttling of EED facilitates assembly of PRC2 complexes, but I understand that this is not an easy experiment to do because EED loss would disassemble all PRC2 complexes. I think the authors should tone down the conclusion of the figure.

Thank you for your positive feedback on our IP-MS results in original Figures 5a-c. We appreciate your thoughtful comment regarding the conclusion of the figure. We understand that ideally, depletion of EED would be necessary to demonstrate that shuttling of EED facilitates the assembly of PRC2 complexes. However, as the reviewer observes, this experiment is challenging to perform, as EED loss would disassemble all PRC2 complexes. We agree with your suggestion to tone down the conclusion of the figure, as shown on lines 291-293.

“Hence, time-dependent shuttling of EED components may facilitate cyclic assembly of PcG complex in order to orchestrate rhythmicity of H3K27me3 deposition around Polycomb direct targets and drive their circadian expression pattern in post-mitotic skeletal muscle.”

Lluís Morey

Reviewer #3 (Remarks to the Author):

This current study by Liu et al presented several novel findings, regarding the circadian regulation of Ezh1, a component of PRC2 complex, its role in modulating circadian gene transcription, and the integrated role in gene repression via H3K27me3 and gene activation activity via Pol II-mediated transcription initiation. Overall the results on reciprocal circadian control of Ezh1 and PRC2 complex assembly and its regulation of circadian gene transcription is strong. However, how Ezh1 may integrate transcription repression and activation to drive circadian oscillation of transcription via H3K27me3 and Pol II interaction, respectively, is not convincing. More work is needed to directly link circadian oscillation of transcription with these mechanisms to be impactful in the field. Below are the major points to be considered regarding this study.

We appreciate the reviewers' positive comments and constructive suggestions to strengthen the proposed conclusions of our manuscript. We conducted a series of new experiments and analyses to address these concerns. We have also updated the main text and figures accordingly, highlighting the changes in our revised manuscript.

For the convenience of reviewers, we have included figures in the rebuttal letter, some of which are also presented in our revised manuscript or supplemental files. To distinguish these figures, we have highlighted them with different colors when referencing them.

1. The role of Ezh1 in H3K27me3 and Pol II-mediated transcription mechanism has been established in prior studies (PMID: 2219688, PMID: 19026781). In fact, the function of Ezh1 in H3K27me3 was found to be negligible on global H3K27me2/3 levels, while may instead directly represses transcription through compacting chromatin (PMID: 19026781). Furthermore, similar as the authors found, previous study in muscle cells revealed “genome-wide association of Ezh1 complex with active epigenetic mark (H3K4me3), RNA polymerase II (Pol II), and mRNA production”. Thus H3K4me3 marks in shEzh1, and shEzh1+EZH1a should be assessed in addition to H3K27me3.

We have incorporated the analysis of H3K4me3 (Fig.8) [revised Supplemental Fig.S5]. Interestingly, we observed that the occupancy of H3K4me3 around the *Bmal1*, *Per1*, and *Per2* loci did not exhibit a circadian pattern. Moreover, the intensity of H3K4me3 signals was upregulated upon *Ezh1* depletion, which is in contrast to the expected H3K27me3 pattern. These findings suggest a coordinated role of bivalent marks around core clock genes, and this coordination may extend to other cyclically expressed genes, indicating a conserved role for these marks.

Figure. 8 Occupancy profile of H3K4me3 surrounding core clock genes following *Ezh1* depletion. a-c ChIP experiment of H3K4me3 at transcription start site (TSS) respectively of *Bmal1* (a), *Per1* (b), and *Per2* (c) during different circadian time points in synchronized C2C12 from WT, *shEZH1* and *shEZH1^{Ezh1 α}* (n= 3 per time point). Immunoprecipitated DNA was quantified by qPCR and normalized to input. IgG represents ChIP experiment performed with an isotype-matched control immunoglobulin (normal rabbit IgG) to H3K4me3.

It also appears that the repressive functions of Ezh1 in circadian transcription is very limited (Fig. 3G cluster 1) that comprised less than 10% of cycling transcripts, which could be due to secondary regulation. Overall the data presented strongly supports a predominantly activating role of Ezh1 in either circadian or global transcription (Fig. 3b & 3g; Fig. 6). Thus the central idea that circadian gene transcription requires the integration of positive and negative transcriptional regulations by Ezh1 was not supported by these findings, particularly the core clock genes appears to be mostly down-regulated (Fig. 3b).

We thank the reviewer raising this point out. In our initial manuscript, we initially identified cluster 1 and cluster 2 genes based on total RNA-seq data, which served as representative examples of rhythmic targets showing both repression and activation. However, total RNA-seq primarily captures the mature RNA state, encompassing multiple regulatory processes such as transcription, post-transcriptional modifications, and RNA turnover. Therefore, it may not fully reflect the direct and primary role of EZH1 α in the transcriptional process. To address this limitation, we conducted TT-seq analysis at different time points (12h, 16h, and 24h) to specifically examine cyclic nascent transcription. By reanalyzing these TT-seq data, our aim was to redefine potential rhythmic targets of EZH1 α involved in activation and repression.

We focused on genes with peaks at 16h and troughs at 24h due to several reasons:

1. We observed peak interactions between EZH1 α and the RNA Pol II machinery or other PRC2-EZH1 α components at these respective time points.
2. The TT-seq data showed the strongest transcriptional activity at 16h.
3. EED shuttle was more enriched in the chromatin fraction at 24h.

Based on these considerations, we selected candidate genes that exhibited significant peaks at 16h (referred to as TT-seq 16h peaks) and/or significant troughs at 24h (referred to as TT-seq 24h troughs) during the 24-hour circadian cycle. In total, we identified 1215 rhythmic transcripts with a significantly highest expression value at 16h compared to 12h and 24h, as well as 944 rhythmic transcripts with a significantly lowest expression value at 24h compared to 16h and 12h (Fig.9d, e) [revised Fig.7d, e]. Further Gene Ontology (GO) analyses of these two sets of targets revealed their involvement in distinct biological pathways, suggesting the spatial partitioning of EZH1 α into different complexes to exert an activation or repressor role (Fig.10) [revised Supplemental Fig.S13g, h].

To further investigate whether the cycling transcripts are a result of secondary regulation or direct targeting by EZH1, we generated new EZH1 α ChIP-seq data using synchronized WT C2C12 myotube cells at different time points. In addition, we also generated SUZ12 and H3K27me3 ChIP-seq data across different time points, allowing us to examine the occupancy profiles of these key PcG factors in conjunction with EZH1 throughout the circadian cycle (Fig.9a-c) [revised Fig.7a-c]. With the availability of new potential activation or repressive rhythmic candidates from both TT-seq and EZH1 ChIP-seq data, we conducted an intersection analysis to compare the ChIP-seq data of EZH1 α , SUZ12, and H3K27me3 with the TT-seq 16h peak and 24h trough genes. This analysis revealed that a significant portion of the rhythmic genes bound by EZH1, SUZ12 and H3K27me3 in terms of both repression and activation (Fig.11a, b, e, f, i, j) [revised Supplemental Fig.S14 a, b, e, f, i, j]. It is noteworthy that there was a higher number of overlapping genes between H3K27me3 ChIP-seq and TT-seq compared to the other two marks, EZH1 and SUZ12. This difference in overlap could be attributed to variations in ChIP efficiency. Among the EZH1 α -bound TT-seq 16h targets, 75 out of 97 genes exhibited co-occupancy with SUZ12 and H3K27me3. Similarly, among the EZH1 α -bound TT-seq 24h targets, 90 out of 128 genes showed co-occupancy with SUZ12 and H3K27me3 (Fig.9f, g) [revised Fig.7f, g]. Interestingly, a closer examination of the binding profiles of EZH1, SUZ12, and H3K27me3 around the activation targets revealed a distinct pattern compared to the repressive targets. For the repressive targets, EZH1, SUZ12, and H3K27me3 deposition occurred around the transcription termination site (TTS) region, while for the activation targets, these proteins were more enriched within the gene body. Moreover, we observed a partial depletion of H3K27me3 levels around the transcription start site (TSS) of activation targets, indicating the possible involvement of other demethylation processes in establishing this profile (Fig.9h-m) [revised Fig.7h-m]. These distinct binding profiles of PcG proteins, including EZH1 and SUZ12, along with the distribution of H3K27me3, suggest that the relative levels of H3K27me3 around the TSS region play a crucial role in modulating RNA Pol II activity for both activation and repression targets regulated by EZH1.

Subsequently, we investigated whether the occupancy of EZH1 and SUZ12 on these targets exhibits a time-dependent pattern. We specifically examined the peak intensity around the transcription start site (TSS) spanning one nucleosome (-/+ 147bp) for both repressive and activation targets. However, we did not observe significant rhythmic changes in

the binding intensity, except the H3K27me3 peak intensity around the rhythmic repressive targets (**Fig.11c, d, g, h, k, l**) [**revised Supplemental Fig.S14c, d, g, h, k, l**]. Notably, the peak occupancy of H3K27me3 around the TSS of repressive targets coincides with the phase when EED is more enriched with chromatin, indicating a consistent association between EED and the regulation of repressive targets.

In summary, EZH1 α plays a crucial role in basal transcription based on TT-seq analysis. However, when it comes to constitutively repressed genes like the *Hox* cluster genes and *Neurog1*, we still observed EZH1 α occupancy around these well-established repressive targets (**Fig.12**). It is important to consider that the removal of the silent state and the reactivation processes may not be fully triggered immediately upon *Ezh1* depletion, especially in non-cycling myotube cells. Thus, TT-seq assay may not fully capture nascent transcription state of these repressed targets.

When it comes to specifically enhancing or repressing rhythmic transcripts, EZH1 α exhibits dynamic partition into different complexes, enabling it to function as either an activator or a repressor. It is noteworthy that SUZ12 appears to co-bind these targets, potentially forming a subcomplex with EZH1 α , as observed in other systems^{4,5}. Furthermore, the recruitment of other factors such as Mediators and EED to the RNA Pol II machinery or PRC2-EZH1 complex is under precise rhythmic regulation (**Fig.13**) [**revised Fig.8**]. To gain a better understanding of this phenomenon, we have created a schematic model to elucidate the underlying mechanisms (**Fig.13**) [**revised Fig.8**].

The reviewer highlighted that upon depletion of *Ezh1*, core clock genes exhibit down-regulation, which was further supported by TT-seq data. An example of this is illustrated in the IGV track of *Per2* (**Fig.14**). Thus, we cannot entirely exclude the possibility of secondary effects resulting from compromised expression of core clock genes. In other words, we suggest that the core circadian transcription factor BMAL1/CLOCK orchestrates the circadian patterns of RNA Pol II activity, while EZH1 α would play a specific role in the integrity of the initiation complex. These observations are supported by various pieces of evidence. Firstly, the major alteration observed in common circadian genes is in their amplitude. TT-seq analysis also confirms that the expression levels can be partially restored, although the rhythmicity remains compromised. This indicates that BMAL1/CLOCK serves as the primary driver for establishing the entire circadian program, while the role of EZH1 α is to precisely modulate a subset of circadian genes to fulfill specific physiological requirements, and all these precisely mediated targets are directly bound by EZH1 α (**Fig.13**) [**revised Fig.8**].

Figure 9: Differential Binding Patterns of EZH1 α , SUZ12, and H3K27me3 at Activated and Repressed Circadian Targets. a-c Heatmap showing EZH1 α (a), SUZ12 (b), and H3K27me3 (c) ChIP-seq signal around

transcription start site (TSS), gene body, and transcription terminaton site (TES) in wild-type (WT) synchronized C2C12 myotube cells at indicated time point. **d-e** Clustering of genes expressed with the highest level (peak) or lowest level (trough) at 16h (**d**) and 24h (**e**), as determined by TT-seq in wild-type (WT) synchronized C2C12 cells. **f-g** Upset plot illustrating the intersection of EZH1 α , SUZ12, and H3K27me3 co-occupancy status around genes that exhibit peak expression at 16 hours (**f**) or trough expression at 24 hours (**g**), as determined in panel (**d**) and (**e**). **h-j** Metaplot analyses of EZH1 α (**h**), SUZ12 (**i**), and H3K27me3 (**j**) ChIP-seq peak signals at various time points surrounding the transcription start sites (TSSs) of genes exhibiting the highest expression peak at 16 hours (peak at 16h), as determined by TT-seq in wild-type (WT) synchronized C2C12 cells. **k-m** Metaplot analyses of EZH1 α (**k**), SUZ12 (**l**), and H3K27me3 (**m**) ChIP-seq peak signals at various time points surrounding the transcription start sites (TSSs) of genes exhibiting the lowest expression peak at 24 hours (trough at 24h), as determined by TT-seq in wild-type (WT) synchronized C2C12 myotube cells.

Figure. 10: GO analyses of rhythmic activation and repressive targets.

Figure 11. Distinct occupancy profile of EZH1α, SUZ12 and H3K27me3 around the 1st nucleosome of activated and repressive targets. a-b Venn diagram of overlap between EZH1 ChIP-seq and genes expressed with the highest level (peak) or lowest level (trough) at 16h (a) and 24h (b), as determined by TT-seq in wild-type (WT) synchronized

C2C12 cells. **c-d** Violin plots illustrate the peak intensity of EZH1 ChIP-seq peaks surrounding the transcription start site of genes expressed with the highest level (peak) or lowest level (trough) at 16h (**c**) and 24h (**d**) co-bound by EZH1 α , SUZ12 and H3K27me3 encompassing upstream and downstream regions spanning one nucleosome. Each dot on the plot represents the average peak intensity around each gene. **e-f** Venn diagram of overlap between SUZ12 ChIP-seq and genes expressed with the highest level (peak) or lowest level (trough) at 16h (**e**) and 24h (**f**), as determined by TT-seq in wild-type (WT) synchronized C2C12 cells. **g-h** Violin plots illustrate the peak intensity of SUZ12 ChIP-seq peaks surrounding the transcription start site of genes expressed with the highest level (peak) or lowest level (trough) at 16h (**g**) and 24h (**h**) bound by EZH1 α , SUZ12 and H3K27me3 encompassing upstream and downstream regions spanning one nucleosome. Each dot on the plot represents the average peak intensity around each gene. **i-j** Venn diagram of overlap between H3K27me3 ChIP-seq and genes expressed with the highest level (peak) or lowest level (trough) at 16h (**i**) and 24h (**j**), as determined by TT-seq in wild-type (WT) synchronized C2C12 cells. **k-l** Violin plots illustrate the peak intensity of EZH1 ChIP-seq peaks surrounding the transcription start site of genes expressed with the highest level (peak) or lowest level (trough) at 16h (**k**) and 24h (**k**) occupied by EZH1 α , SUZ12, and H3K27me3 encompassing upstream and downstream regions spanning one nucleosome. Each dot on the plot represents the average peak intensity around each gene. To identify significant differences in peak intensity across time points, pairwise Wilcoxon rank-sum tests were conducted. The p-values from these tests were adjusted for multiple comparisons using the Benjamini-Hochberg method.

Figure 12. IGV track of EZH1, SUZ12, and H3K27me3 ChIP-seq peaks around *Hoxd10* cluster.

Figure 13: Schematic model of EZH1 function. A conceptual model demonstrates the integrated role of EZH1 α by dynamically partitioning into distinct complexes in space and time. In this model, EZH1 α functions as a catalytic enzyme within the PRC2-EZH1 complex, responsible for depositing H3K27me3 marks near the transcription start site (TSS) of repressive targets involved in pathways like myofibril assembly. Remarkably, this deposition process coincides with the translocation of EED, another essential component of the PRC2-EZH1 complex, from the cytosol to the nucleus, occurring at the same peak phase. Furthermore, during an alternative phase, EZH1 α associates with the RNA Pol II machinery, acting as a transcriptional activator that governs the enhanced activation of targets, such as metaphase plate congression.

Figure 14: IGV track of TT-seq signals around *Per2* loci.

2. The use of stable cell lines shEzh1, and shEzh1+EZH1a will not allow to distinguish direct vs. indirect effects of Ezh1 in transcription control and PRC2 complex stoichiometry, particularly there are global changes in transcripts affected by Ezh1. This could be a key confounding issue in interpretation of the results presented.

We express our gratitude to the reviewer for raising this crucial concern. Of course, while EZH1 controls a variety of tissue-specific genes, not all of them undergo circadian regulation and vice versa. In order to thoroughly investigate the direct and indirect effects of EZH1, we considered employing the Degron system to assess the immediate impact of EZH1 depletion on transcriptional activity. However, implementing the degradation of EZH1 within the context of a synchronized cell-autonomous circadian system presents significant challenges.

Furthermore, when considering the "*direct effects of EZH1 in transcriptional regulation*" it is important to note that the main functions of EZH1 α are primarily associated with chromatin or RNA Pol II. Rather than total RNA-seq, to capture the state of nascent transcripts, we opted to utilize TT-seq, which is a suitable assay for measuring the direct effects of EZH1 depletion, in the light in particular of the novel findings regarding the control of RNA Pol II initiation complex integrity.

To address these challenges, we generated an inducible rescue cell line in which EZH1-HA expression is controlled by the Tetracycline-Responsive Element (TRE) promoter in an *Ezh1* knockdown background. This new inducible system allowed us to restore EZH1 α function 24 hours prior to synchronization (**Fig.15a-c**) [**revised Supplemental Fig.S10a-c**].

To assess the primary role of EZH1 α in transcriptional regulation, specifically in terms of direct effects, we performed TT-seq experiments using wild-type, knockdown, and the new inducible rescue cell line at the active phase, specifically at 16 hours. The TT-seq results clearly demonstrated that the transient restoration of EZH1 expression could restore the transcriptional features (**Fig.15e, f**) [**revised Supplemental Fig.S10e, f**].

To further differentiate between the direct and indirect effects of EZH1 α , as mentioned in the previous Point 1, we also conducted new EZH1 α ChIP-seq experiments at different time points. By integrating the EZH1 α ChIP-seq and TT-seq data, we aimed to define the primary effects of EZH1 α and determine whether the binding was direct or indirect. As explained in the reply for Point 1, we revealed that a significant portion of the rhythmic genes are directly bound by EZH1 α in terms of both repression and activation.

Collectively, we firmly believe that the global changes observed in transcripts are directly influenced by EZH1, which is supported by the TT-seq analyses using both constitutive and inducible rescue cell lines. Regarding nascent rhythmic transcripts, some are direct targets of EZH1 α , while others are likely to be directly bound by the BMAL1-CLOCK complex, indicating the involvement of other regulatory mechanisms.

Figure 15. Expression levels of differentially expressed genes were restored by transiently inducing the expression of EZH1 α . **a** The protein levels of Ezh1 α were assessed through Western blot (WB) analysis in various C2C12 cell lines, including WT, *shEzh1*, and *shEzh1*^{TET-EZH1 α} (treated with DMSO or DOX) transgenic cells. In the WB analysis, the band observed in the WT lane corresponds to the endogenous EZH1 α protein. In contrast, the band in the DOX lane represents the HA-EZH1 α . **b** Different cell lines were utilized to immunoprecipitate the HA-EZH1 α protein using HA-conjugated agarose beads. Subsequently, the HA-tagged proteins were visualized using anti-HA. **c** Experimental design illustrating TT-seq samples collection at indicated conditions. **d** FPKM value of EZH1 in WT, *shEzh1*^{TET-EZH1 α} treated with DMSO (DMSO), and *shEzh1*^{TET-EZH1 α} treated with DOX (DOX). **e** Absolute fold changes of significantly differential genes in comparison with indicated biological conditions at 16h. **f** Volcano plot representation of differential expression profile of nascent transcripts in comparison with different biological conditions at 16h.

3. Data presented in Figure 7 does not support circadian modulation of Ezh1 in Pol II transcription activity. There are not sufficient details described to fully understand the data. The only somewhat clear result is Fig 7e and 7f showing loss of Ezh1 attenuated PolII stoichiometry, but this may not be a direct effect of Ezh1. How the stoichiometry value is expressed in these panels are not clear, as all values are normalized to 1, instead of the relative stoichiometry as shown in Fig. 5a-c which indicates relative abundance of subunits being assessed to determine stoichiometry.

We appreciate the reviewer's concern regarding this figure. Our study has provided evidence that depletion of EZH1 leads to disruptions in the circadian expression of nascent transcripts, as observed through TT-seq analysis. Our circadian TT-seq data clearly indicate that the process of nascent transcription follows a circadian pattern, with the highest activity observed at 16 hours. Additionally, it is important to note that the TT-seq assay was performed using bulk cells, which may not fully capture the dynamics of RNA Pol II at the single-molecular level in live cells.

To address this gap in understanding, we employed single-molecule imaging to measure the dynamics of RNA Pol II clustering and predict mRNA output. Additionally, we used IP-qMS to assess the interaction between EZH1 and RNA Pol II subunits. Our results revealed that the interaction between EZH1 and RNA Pol II subunits displayed highest intensity at 16 hours, which is consistent with the strongest TT-seq signal. As a control, we also compared the half-life of RNA Pol II transient clusters at 24 hours. It is important to note that from the beginning, we included this data in the supplemental information, which could have led the reviewer to believe that we only analyzed this data for one time point when evaluating this figure. We have now moved the tcPALM analysis of RNA Pol II transient clusters at 24 hours, clearly demonstrating no significant changes (Fig.16d, e) [revised Fig.7d, e]. This suggests that EZH1 is required to stabilize RNA Pol II specifically during the enhanced transcription phase at 16 hours (Fig.16b, c) [revised Fig.7b, c]. However, when more genes are repressed through the EED shuttle and PRC2-EZH1 complex assembly, the dynamics of RNA Pol II transient clusters show no significant changes (Fig.16d, e) [revised Fig.7d, e]. Additionally, besides the half-life of RNA Pol II transient clusters, we found that the number of RNA Pol II transient clusters was significantly down-regulated specifically at 16h specifically, and this effect could be rescued by constitutive expression of EZH1 (Fig.17) [revised Supplemental Fig.S11 b, c].

Further, we examined the stoichiometry of RNA Pol II machinery in WT, shEzh1, and shEzh1+EZH1a backgrounds at 16 hours post-synchronization. The findings depicted in revised Fig.7 reveal a direct association between EZH1 and various components of RNA Pol II, including subunits, general transcription factors, and mediators. Notably, the reduction of EZH1 levels impacts the stoichiometry of Pol II in the Ezh1 knockdown cell line. This effect primarily affects the formation of the preinitiation complex, involving general transcription factors and mediators. These observations are consistent with the tcPALM data, which indicate significant changes in the transient cluster. Crucially, the reintroduction of EZH1 in the knockdown background exhibits a reverse or enhancing effect, leading to the restoration of Pol II stoichiometry. This outcome reinforces the notion that EZH1 has a specific impact on this process. Additionally, it is worth noting that our findings already revealed the interactions between EZH1 and RNA Pol II subunits and Mediators.

Collectively, these data provide further support for the direct and specific role of EZH1 in preserving the integrity of the RNA Pol II preinitiation complex. However, we acknowledge that we still lack a comprehensive understanding of the mechanistic effects of EZH1 on the integrity of the RNA Pol II machinery, a level of details that we believe would be beyond the scopes of this work.

Figure 16. tcPALM analysis of RNA Pol II transient cluster half-life.

Figure 17. tcPALM analysis of RNA Pol II transient cluster number.

4. The cluster 1 circadian genes identified in Fig. 3g further analyzed in Fig. 4a are not consistent. These genes clearly did not display rhythm in Fig. 4a as shown initially in Fig. 3g. This entire analysis in Fig. 4 thus not relevant to assess circadian regulation by Ezh1.

We apologize for any confusion caused by the initial version of the manuscript. In the first version, the purpose of Figure 3g and Figure 4a served different objectives. Figure 3g presented a heatmap analysis of all circadian genes under WT conditions, utilizing a hierarchical clustering algorithm to identify distinct gene sets within different clusters. This approach allowed for clearer differentiation between the various gene sets. Conversely, Figure 4a provided an alternative visualization of a subset of Figure 3g. It specifically highlighted the circadian expression pattern of cluster 1 genes, which were sorted based on their LAG phase determined by JTK-CYCLE (ordering genes according to the time point with the highest expression).

To address any potential confusion, we have reorganized the main figures by exclusively displaying Cluster 1 and Cluster 2 [revised Fig. 3g, h], and we have removed the original Figure 3g. This revised presentation aims to provide a clearer and more focused depiction of the relevant gene clusters.

5. For Fig. 6e, It is not clear how the circadian time points post-sync at 12, 16 and 24 hours were selected, as they do not represent the peak or trough of steady-state transcript levels based on Fig. 3f and 3g.

The primary advantage of TT-seq is its ability to capture the profile of nascent transcripts, which often differs from that of mature total transcripts. However, determining the appropriate time points for TT-seq can be challenging when relying solely on total RNA-seq data¹². To address this issue, we developed a hypothesis that the interaction intensity between EZH1 and RNA Pol II-associated components could serve as a correlate for transcriptional activity differences. We utilized this hypothesis to guide our selection of time points for the TT-seq experiment.

Our observations revealed a relatively stronger interaction between EZH1 and RNA Pol II-associated components at 16 hours post-synchronization, while 12 and 24 hours displayed less interaction intensity. Additionally, we found that the assembly of the PRC2-EZH1 repressive complex exhibited the strongest interaction at 24 hours, which could serve as proper control. Based on these findings, we selected the time points of 12, 16, and 24 hours post-synchronization for our TT-seq experiment. Our results demonstrated that the highest transcriptional activity, as measured by TT-seq, was observed at 16 hours. This provides support for our hypothesis and highlights the presence of a circadian pattern in transcriptional activity within synchronized C2C12 myotube cells, which correlates with the relatively strong interaction between EZH1 and the RNA Pol II machinery.

Indeed, our tcPALM analysis confirms the concordance between the results obtained from our IP-qMS and TT-seq experiments. Specifically, we observed disruptions in the RNA Pol II transient clusters at 16 hours, which aligns with the overall effects detected in our IP-qMS and TT-seq data. However, it is worth noting that the peak phase of PRC2-EZH1 complex assembly suggests that the majority of EZH1 associates with the PRC2-EZH1 complex, while only a minority of EZH1 exhibits spatial partitioning with RNA Pol II. This finding provides a plausible explanation for the mild effects observed at 24 hours.

6. Fig. 5d: EED levels in individual fractions assayed does not add up to be total protein levels shown for different time points. Cytosol, nucleosomal and chromatin fractions at 16 and 32 hours are all quite low although the total levels were comparable to 24 hours. This result across time points are not consistent.

We appreciate the reviewer's attention to this specific point. The preparation of total protein extracts and fraction-specific protein extracts has been done using different methods. The detailed procedures have been described extensively in the revised manuscript, but for convenience, we will summarize them below, lines 604-615.

"To obtain total extracts, cells were lysed in a total extraction buffer of 50 mM Tris-HCl (pH 8), 150 mM NaCl, 1% NP-40, and 1× protease inhibitor. The lysate was then subjected to sonication and centrifugation at 16,300g and 4

°C for 10 minutes. The nuclei were isolated from the cytosolic fraction using ice-cold 0.1% NP-40 in PBS. The supernatant containing the cytosolic fraction was retained. For the extraction of nuclei and chromatin fractions, a salt concentration gradient was utilized. Briefly, the nuclei pellet was resuspended in strip buffer composed of 10 mM Tris-Cl (pH 7.4), 1 mM EGTA, 1.5 mM KCl, 5 mM MgCl₂, 290 mM sucrose, 0.1% Triton X-100, and 1 mM DTT for nucleosol extraction. Subsequently, the chromatin-bound fraction was prepared by using a medium salt buffer containing 20 mM HEPES (pH 7.9), 25% glycerol, 500 mM KCl, 1.5 mM MgCl₂, 0.2 mM EDTA, 1 mM DTT, 1X Complete mini EDTA-free (Roche), and subjected to sonication (Bioruptor, 30 seconds ON/30 seconds OFF, 10 cycles) to isolate chromatin fractions sequentially.”

Specifically, we employed a salt-based fractionation approach to prepare protein extracts from different fractions. We recognized that a potential problem in our previous methodology was the omission of a sonication step for the last chromatin fraction. This oversight could have introduced technical variations and potentially resulted in the exclusion of some PcG components that remained associated with chromatin, even under high salt concentrations.

To rectify this, we repeated all the assays while incorporating the sonication step for the last chromatin fraction. As a result, our revised Figure 6 now demonstrates that the sum of the protein amounts from all the fractions is more similar to the total protein at each time point. This improvement in the accuracy and reliability of our data is consistent with previous findings, such as the distribution of EED to the chromatin fraction at 24 hours. We have included the representative Western blot image in the revised main figure, and we also provide additional replicate data in **Fig.18** for further validation of these findings. It is important to note that the method utilized for preparing total protein extracts is still distinct from our fractionation method, which may introduce slight inconsistencies.

We are confident that these new data will support the robustness of our findings.

Figure 18. Replicates for fractionation WB

7. Figure 2: It is perplexing that circadian phase shown in muscle between Ezh1 association in Bmal1, Per1 and Per2 promoter (Fig. 2c) are the same (in-phase), while the phase angle in synchronized culture (Fig. 2d) between Bmal1 and Per1/per2 are reversed. Understanding this discrepancy for the in vivo result could be challenging due to the complete antiphase expression between Bmal1 and Per2.

We appreciate the reviewer for bringing this issue to our attention. During the organization of the different panels, there was an inadvertent misuse of the same plot, intended for the *Per2* locus, in place of the *Bmal1* locus. We have rectified this error by replacing the panel with the correct plot [revised Fig. 2c]. Furthermore, to ensure transparency and support our findings, we have provided all the source data used to generate these plots. We sincerely apologize for any confusion caused by this mistake.

8. Figure 1. It is interesting that Ezh1b display two peaks in its protein rhythm (Fig. 1b, 1c) while transcript level does not show two peaks (Fig. 1a). This appears to be in muscle only but not in synchronized culture (Fig. 1e). The authors may need to validate this in vivo finding with additional circadian protein profiles, particularly Bmal1. In addition, if Ezh1a and Ezh1b are expressed at comparable levels in muscle, and the rescue cell line, shEzh1+EZH1a, only restores Ezh1a but not Ezh1b (Fig. S4a), does this suggest Ezh1b does not contribute to the circadian transcription regulation observed even though displays a similar rhythmic control as 1a? This could be an important point for discussion.

We appreciate your valuable feedback on Figure 1 of our manuscript. We acknowledge the observation of two peaks in the Ezh1 β protein rhythm. It has been previously reported that the mammalian 12-hour rhythm is driven by a distinct 12-hour pacemaker that is cell-autonomous and can be influenced by metabolic and ER stress pathways⁷. In light of this, the differential circadian profile of the EZH1 β protein observed in vivo and in vitro may be attributed to the absence of metabolic signals in the in vitro setting. To further validate this profile, as suggested by the reviewer we include circadian protein profiles of BMAL1 (**revised Fig.1b**) from skeletal muscle tissue in our revised manuscript.

The reason why we introduced only the long isoform EZH1 α into the shEzh1 background is that this approach allowed us to partially differentiate the roles of the two EZH1 isoforms. Our previous studies have indicated that the primary function of EZH1 β is to retain EED within the cytosol fraction. In our current study, we show that the interaction between EZH1 α and EED, the association of EED with chromatin, and the rhythmic pattern of H3K27me3 ChIP-seq peaks around rhythmic repressive targets coincided at the same phase. These findings could potentially be explained by the circadian pattern exhibited by EZH1 β , which sequentially governs the EED release in a time-dependent pattern. Moreover, in our TT-seq experiment, although we achieved restoration of the expression levels of target genes by expressing only EZH1 α , we noted that their rhythmicity was not fully restored. This observation suggests that the absence of EZH1 β may be responsible for the incomplete recovery of rhythmic patterns. We believe that EZH1 β plays an important role in the repressive function by regulating the stoichiometry of the complete repressive PRC2-EZH1 complex.

As suggested by the reviewer, we have addressed the potential reasons behind the 12-hour circadian pattern exhibited by EZH1 β specifically in vivo, and discussed the role of EZH1 β in modulating the rhythmic pattern of H3K27me3. For the convenience of the reviewer, we have listed these points below, lines 477-497.

“In our current study, we made an intriguing discovery that the EZH1 β protein exhibits a 12-hour rhythm specifically in skeletal muscle tissue. However, this 12-hour rhythm is not observed in cell-autonomous systems. These findings collectively suggest the presence of potential endogenous physiological signals, such as metabolic signals or inter-organ communication signals, in vivo, which are absent in synchronized in vitro cell cultures. Previous reports have indicated that XBP1 acts as a major transcription factor governing the 12-hour cycling pattern⁷, and regulates various lipid and metabolic pathways associated with this rhythm⁸. Interestingly, the expression of Ezh1 β transcripts shows a 24-hour rhythm, suggesting a regulation at the post-transcriptional, translational, or post-translational levels that is independent of XBP1-mediated transcriptional regulation. In our previous study, we demonstrated that ubiquitin E3 ligases, such as HUWE1 and NEDD4⁹ interacts with EZH1 β protein. Ubiquitin-mediated proteasome degradation has been extensively studied and plays an essential role in regulating protein rhythmic homeostasis¹⁰. It is worth investigating whether ubiquitin-mediated degradation could govern the 12-hour rhythm of EZH1 β in a circadian pattern by sensing potential endogenous signals, such as metabolic signals. However, establishing a direct connection between ubiquitin E3 ligase and EZH1 β through depletion of HUWE1 or NEDD4 would be challenging, given the multiple downstream targets of both HUWE1 and NEDD4. For example, HUWE1 has been shown to be involved in the degradation of REV-ERB α ¹¹. In our rescue cell line shEzh1^{EZH1 α} , where EZH1 β is down-regulated, we observed that transcriptional rhythmicity could not be fully restored. This indicates the essential role of EZH1 β in orchestrating circadian gene expression, by modulating the rhythmic shuttle of EED and the assembly of the PRC2-EZH1 repressive complex.”

9. The discussion portion of the manuscript is lacking. The authors should include sufficient discussions with existing literature and particularly the potential caveats and limitations of the study to better inform the field.

We have carefully revised it to address the points raised and enhance the analysis of potential caveats and limitations in our study. Here are some key highlights of the revised discussion:

1. We have delved deeper into the coordination between RNA Pol II activity and H3K27me3 epigenome plasticity, emphasizing how this interplay regulates circadian gene expression by governing the precise spatial-temporal partitioning of EZH1 α into different complexes within post-mitotic skeletal muscle cells. To aid in understanding this concept, we have included a schematic model that illustrates the implications of our findings [revised Fig.8].
2. We have expanded the discussion on the potential regulatory mechanisms underlying the 12-hour rhythm exhibited by EZH1 β . By considering the existing literature⁹, we explore possible factors and pathways that may contribute to this unique circadian pattern of EZH1 β .
3. In terms of caveats and limitations, we acknowledge that one aspect missing from our study is a mechanistic understanding of how EZH1 α modulates the stability of the RNA Pol II machinery. We recognize the importance of this aspect and its potential implications, and we highlight it as an avenue for future research and investigation.

By incorporating these revisions, we aim to provide a more comprehensive and insightful discussion that integrates our findings with existing knowledge while addressing potential limitations and avenues for further exploration.

References:

1. Vo, L.T. et al. Regulation of embryonic haematopoietic multipotency by EZH1. *Nature* **553**, 506-510 (2018).
2. Mousavi, K., Zare, H., Wang, A.H. & Sartorelli, V. Polycomb protein Ezh1 promotes RNA polymerase II elongation. *Mol Cell* **45**, 255-62 (2012).
3. Feng, X. et al. Polycomb Ezh1 maintains murine muscle stem cell quiescence through non-canonical regulation of Notch signaling. *Dev Cell* **58**, 1052-1070 e10 (2023).
4. Xu, J. et al. Developmental Control of Polycomb Subunit Composition by GATA Factors Mediates a Switch to Non-Canonical Functions. *Molecular Cell* **57**, 304-316 (2015).
5. Su, S.K. et al. The EZH1-SUZ12 complex positively regulates the transcription of NF-kappaB target genes through interaction with UXT. *J Cell Sci* **129**, 2343-53 (2016).
6. Blackledge, N.P. & Klose, R.J. The molecular principles of gene regulation by Polycomb repressive complexes. *Nat Rev Mol Cell Biol* **22**, 815-833 (2021).
7. Zhu, B. et al. A Cell-Autonomous Mammalian 12 hr Clock Coordinates Metabolic and Stress Rhythms. *Cell Metab* **25**, 1305-1319 e9 (2017).
8. Ballance, H. & Zhu, B. Revealing the hidden reality of the mammalian 12-h ultradian rhythms. *Cell Mol Life Sci* **78**, 3127-3140 (2021).
9. Liu, P., Shuaib, M., Zhang, H.M., Nadeef, S. & Orlando, V. Ubiquitin ligases HUWE1 and NEDD4 cooperatively control signal-dependent PRC2-Ezh1 alpha/beta-mediated adaptive stress response pathway in skeletal muscle cells. *Epigenetics & Chromatin* **12**(2019).
10. Patke, A., Young, M.W. & Axelrod, S. Molecular mechanisms and physiological importance of circadian rhythms. *Nat Rev Mol Cell Biol* **21**, 67-84 (2020).
11. Yin, L., Joshi, S., Wu, N., Tong, X. & Lazar, M.A. E3 ligases Arf-bp1 and Pam mediate lithium-stimulated degradation of the circadian heme receptor Rev-erb alpha. *Proc Natl Acad Sci U S A* **107**, 11614-9 (2010).
12. Menet, J.S., Rodriguez, J., Abruzzi, K.C. & Rosbash, M. Nascent-Seq reveals novel features of mouse circadian transcriptional regulation. *Elife* **1**(2012).

2nd round peer review:

Reviewers' comments:

Reviewer #1 (Remarks to the Author):

Unfortunately, the authors have not done enough for me to convince me that their data support their proposed model. In particular, I do not see anything in the new data to convince me that EZH1 binds to genes when they are active.

Reviewer #2 (Remarks to the Author):

The authors have successfully addressed all my suggestions and I fully support publication of this work in Nature Comm. Congratulations on this very interesting paper!

Reviewer #4 (Remarks to the Author):

This study made an interesting discovery that the functions of EZH1 in muscle tissue are controlled by circadian rhythms. It added new understandings of both circadian rhythm and epigenetics regulations. The mechanism may be conserved in other systems. All concerns from reviewer 3 were well addressed.

Reviewer #5 (Remarks to the Author):

I was asked to specifically evaluate whether the genome wide data (ChIP-seq) support the claim that EZH1 together with SUZ12 is bound at active genes.

1) Since the authors claim that EZH1 is bound at active genes involved in circadian rhythm (namely *Per1*, *Per2* and *Arntl* (mouse homolog of *Bmal1*); Lane 174-175 and Figure 2c/d), we checked EZH1, Suz12 and H3K27me3 enrichment on the provided bigwig files. None of these three genes were significantly enriched for any of these features. In the case of EZH1, this is in direct contradiction with data shown in Figure 2d.

2) I extensively browsed through the EZH1, SUZ12 and H3K27me3 ChIP-seq in search of EZH1 peaks (using the bed file of replicate 1, GSM7899978_bio_12_rep1_peaks_filtered.bed) that would not overlap with a region enriched for H3K27me3 and could not find any example supporting the claim of the authors. Actually, the three features appear to almost perfectly colocalize with the exception of some H3K27me3 peaks that are neither significantly enriched for EZH1 nor for SUZ12.

We sincerely appreciate the valuable efforts from all the reviewers and thank those who positively evaluated our revised manuscript. However, we regret that two reviewers (1 and 5) were not convinced by the inclusion of our new and extensive ChIP-seq data, in particular about the evidence that EZH1 binds active gene loci.

Indeed, the present study builds on previous work from our lab^{1,2}, and investigates the dynamics and novel physiological adaptive role of PRC2-EZH1 complex in postmitotic cells and not in differentiating or cycling cells, where the PcG cell memory system everybody agree exerts a different function.

That is why we believe that their conclusion may not be correct as it seems to be based on the assumption that colocalization with H3K27m3 would mean repressed state of the underlying promoter, an argument that we have also openly discussed in our revised manuscript. As we are sure the reviewers are aware of, the adoption of this criteria does not apply to several contexts as broadly recognized in the PcG literature^{3,4}, including the circadian field⁵⁻⁸.

However, we present here a new set of ChIP-seq data from our lab in which the direct binding of EZH1 to circadian loci (*Bmall*, *Per1* and *Per2*) is clearer. These data were obtained by an independent approach namely using a HA-Tag version of EZH1, driven by inducible promoter we used for the TT-seq rescue experiment shown in the manuscript. The additional data, including Supplementary Figure 11 and 12, have been incorporated into our revised manuscript. These findings are thoroughly described in the revised manuscript, specifically in lines 341-350, providing a comprehensive account of the observations.

Finally, being aware of the complexity of RNA Pol II regulation at promoters and the verified role of EZH1, we used the state-of-the-art TT-seq technology and not RNA-seq to define the global transcription state of target genes. This is particularly important as EZH1 has been defined as an activator associated with RNA Pol II dynamics, and the biochemical and tPALM data presented in our manuscript regarding the circadian stability of functional RNA Pol II complex provide interesting novel mechanistic insights in this context.

We hope the following arguments will convince the reviewers to reconsider their conclusion and allow publication of our work.

Reviewer 1:

“Unfortunately, the authors have not done enough for me to convince me that their data support their proposed model. In particular, I do not see anything in the new data to convince me that EZH1 binds to genes when they are active.”

Despite the prevailing understanding that Polycomb complexes predominantly contribute to transcriptional repression, a growing body of evidence suggests that in certain contexts, these complexes can actually control transcriptional activation. For example, EZH1 was also reported to exert activating functions during differentiation of skeletal muscle cells, either through mediating RNA Pol II elongation⁹ or MyoD recruitment¹⁰. The activation mechanism of EZH1 potentially involves an EZH1-SUZ12 subcomplex that does not contain EED. This subcomplex

has the ability to bind to active chromatin and positively regulate transcription^{11,12}. EZH1's activation is also implicated in the initial release of RNAPII pausing, instead of H3K27me3 removal, which is characteristic of bivalent genes^{13,14}. Additionally, EZH1 plays a role in maintaining gene expression of the Notch signaling pathway in muscle stem cells¹⁵. Collectively, these studies confirm a primary positive regulatory role of EZH1 in transcriptional regulation. It is worth noting that EZH1 is not the only Polycomb protein involved in gene activation, as also EZH2 has been implicated in the process of transcriptional activation^{4,16-18}. Indeed, we are sure the reviewers will agree that there are still several aspects of PRC2 role, and PcG in general, in somatic cells that have not been elucidated, yet relevant to both healthy and pathological contexts.

As stated in our previous response, it is crucial to clarify that we do not dismiss the repressive function of PRC2-EZH1. We acknowledge that the PRC2-EZH1 complex still plays a critical role in depositing H3K27me3 marks around canonical Polycomb group (PcG) targets, thereby repressing their expression. Instead, we highlight the diverse function of EZH1 by providing additional mechanistic evidence to demonstrate its involvement in gene activation processes, through association with RNA Pol II machinery.

Taken together, our present study reiterates the importance of cellular context and genetic programs in defining the role of EZH1. Moreover, it underscores the intricate nature of Polycomb-mediated transcriptional regulation, emphasizing the flexibility of chromatin states and transcriptional processes, a key and well documented physiological aspect of epigenome function.

Reviewer 5:

“I extensively browsed through the EZH1, SUZ12 and H3K27me3 ChIP-seq in search of EZH1 peaks (using the bed file of replicate 1, GSM7899978_bio_12_rep1_peaks_filtered.bed) that would not overlap with a region enriched for H3K27me3 and could not find any example supporting the claim of the authors. Actually, the three features appear to almost perfectly colocalize with the exception of some H3K27me3 peaks that are neither significantly enriched for EZH1 nor for SUZ12.”

We appreciate the reviewer for taking the time to cross-check our ChIP-seq data. It appears that there is a misunderstanding between the reviewer's comments and our conclusion regarding the co-localization of EZH1 and H3K27me3. **To clarify, we never claimed that active targets of EZH1 were not occupied by H3K27me3 marks. Instead, in our revised manuscript (lines 421-432), we have explicitly commented the finding that was pointed out by the reviewer. Indeed, and more importantly, we believe that the assumption that H3K27m3 mark is per se reflecting uniquely the silenced state may not be correct if used as firm criteria to define the functional state of EZH1 targets.**

Although EZH1, SUZ12 and H3K27me3 co-localized around both the active and repressive rhythmic targets, a closer examination of the binding profiles of EZH1, SUZ12, and H3K27me3 around the activation targets revealed a distinct pattern compared to the repressive targets. For the repressive targets, EZH1, SUZ12, and H3K27me3 deposition occurred around the transcription termination site (TTS) region, while for the activation targets, these proteins were more enriched within the gene body. Moreover, we observed a partial depletion of H3K27me3 levels around the

transcription start site (TSS) of activation targets, indicating the possible involvement of other demethylation processes in establishing this profile (**Figure 1**). These distinct binding profiles of PcG proteins, including EZH1 and SUZ12, along with the distribution of H3K27me3, suggest that the relative levels of H3K27me3 around the TSS region reflect a different functional state of RNA Pol II activity at active and repressed EZH1 targets.

The presence and dynamics of H3K27m3 on EZH1 active targets is not surprising as this was already and reported in our previous work¹. As a general concept, in somatic non-cycling cells, the presence of “opposing” histone marks indicates the competence to integrate signal of a given PcG target in modulating transcription allowing “plastic regulation of specific classes of target genes in response to environmental signals. opposed to other classes, like cell type specific genes, that are permanently marked by H3K27m3 even in the absence of PRC2. In this work, we analyze non-cycling myotube cells where these different classes are well represented.

We also investigated whether the occupancy of EZH1 and SUZ12 on these targets exhibits a time-dependent pattern. We specifically examined the peak intensity around the transcription start site (TSS) spanning one nucleosome (-/+ 147bp) for both repressive and activation targets. No significant rhythmic changes were observed around the active targets; however, rhythmic changes in H3K27me3 were observed around the rhythmic repressive targets (**Figure 2**). Notably, the peak occupancy of H3K27me3 around the TSS of repressive targets coincides with the phase when EED is more enriched with chromatin, indicating a consistent association between rhythmic EED shuttle and the regulation of repressive targets.

In conclusion, our findings revealed two significant characteristics of H3K27me3 around rhythmic repressive targets. Firstly, H3K27me3 is specifically enriched around the transcription termination sites (TTS) of these target genes. Secondly, we observed a rhythmic occupancy of H3K27me3 around the rhythmic repressive targets. However, it is worth noting that H3K27me3 marks were also detected around rhythmic active targets. In contrast to the repressive targets, these marks were depleted around the transcription start sites (TSS) and did not exhibit a rhythmic pattern. Therefore, in this case, H3K27m3 cannot be used as a firm criteria to define silenced state and the TT-Seq data clearly indicate the circadian active state of the selected EZH1 bound loci.

Figure 1: Differential Binding Patterns of EZH1a, SUZ12, and H3K27me3 at Activated and Repressed Circadian Targets. **a-c** Heatmap showing EZH1a (**a**), SUZ12 (**b**), and H3K27me3 (**c**) ChIP-seq signal around transcription start site (TSS), gene body, and transcription terminator site (TES) in wild-type (WT) synchronized C2C12 myotube cells at indicated time point. **d-e** Clustering of genes expressed with the highest level (peak) or lowest level (trough) at 16h (**d**) and 24h (**e**), as determined by TT-seq in wild-type (WT) synchronized C2C12 cells. **f-g** Upset plot illustrating the intersection of EZH1a, SUZ12, and H3K27me3 co-occupancy status around genes that exhibit peak expression at 16 hours (**f**) or trough expression at 24 hours (**g**), as determined in panel (**d**) and (**e**). **h-j** Metaplot analyses of EZH1a (**h**), SUZ12 (**i**), and H3K27me3 (**j**) ChIP-seq peak signals at various time points surrounding the transcription start sites (TSSs) of genes exhibiting the highest expression peak at 16 hours (peak at 16h), as determined by TT-seq in wild-type (WT) synchronized C2C12 cells. **k-m** Metaplot analyses of EZH1a (**k**), SUZ12 (**l**), and H3K27me3 (**m**) ChIP-seq peak signals at various time points surrounding the transcription start sites (TSSs) of genes exhibiting the lowest expression peak at 24 hours (trough at 24h), as determined by TT-seq in wild-type (WT) synchronized C2C12 myotube cells.

Figure 2. Distinct occupancy profile of EZH1a, SUZ12 and H3K27me3 around the 1st nucleosome of activated and repressive targets. **a-b** Venn diagram of overlap between EZH1 ChIP-seq and genes expressed with the highest level (peak) or lowest level (trough) at 16h (**a**) and 24h (**b**), as determined by TT-seq in wild-type (WT) synchronized C2C12 cells. **c-d** Violin plots illustrate the peak intensity of EZH1 ChIP-seq peaks surrounding the transcription start site of genes expressed with the highest level (peak) or lowest level (trough) at 16h (**c**) and 24h (**d**) co-bound by EZH1a, SUZ12 and H3K27me3 encompassing upstream and downstream regions spanning one nucleosome. Each dot on the plot represents the average peak intensity around each gene. **e-f** Venn diagram of overlap between SUZ12 ChIP-seq and genes expressed with the highest level (peak) or lowest level (trough) at 16h (**e**) and 24h (**f**), as determined by TT-seq in wild-type (WT) synchronized C2C12 cells. **g-h** Violin plots illustrate the peak intensity of SUZ12 ChIP-seq peaks surrounding the transcription start site of genes expressed with the highest level (peak) or lowest level (trough) at 16h (**g**) and 24h (**h**) bound by EZH1a, SUZ12 and H3K27me3 encompassing upstream and downstream regions spanning one nucleosome. Each dot on the plot represents the average peak intensity around each gene. **i-j** Venn diagram of overlap between H3K27me3 ChIP-seq and genes expressed with the highest level (peak) or lowest level (trough) at 16h (**i**) and 24h (**j**), as determined by TT-seq in wild-type (WT) synchronized C2C12 cells. **k-l** Violin plots illustrate the peak intensity of H3K27me3 ChIP-seq peaks surrounding the transcription start site of genes expressed with the highest level (peak) or lowest level (trough) at 16h (**k**) and 24h (**l**) occupied by EZH1a, SUZ12, and H3K27me3 encompassing upstream and downstream regions spanning one nucleosome. Each dot on the plot represents the average peak intensity around each gene. To identify significant differences in peak intensity across time points, pairwise Wilcoxon rank-sum tests were conducted. The p-values from these tests were adjusted for multiple comparisons using the Benjamini-Hochberg method.

“Since the authors claim that EZH1 is bound at active genes involved in circadian rhythm (namely Per1, Per1 and Arntl (mouse homolog of Bmal1); Lane 174-175 and Figure 2c/d), we checked EZH1, Suz12 and H3K27me3 enrichment on the provided bigwig files. None of these three genes were significantly enriched for any of these features. In the case of EZH1, this is in direct contradiction with data shown in Figure 2d.”

We appreciate the reviewer for raising a concern about what we believe is a technical discrepancy between EZH1 ChIP-qPCR and EZH1-BIO ChIP-seq. We would like to provide some explanations for this inconsistency. One possible reason for the discrepancy could be the well-known higher sensitivity of ChIP-qPCR compared to the ChIP-seq approach. A major concern raised by the reviewers in the first manuscript relates to the quality of the EZH1 ChIP-seq data. To address this concern, we generated two different EZH1 ChIP-seq datasets. In addition to the EZH1-BIO ChIP-seq dataset presented in the revised manuscript, we generated another EZH1 ChIP-seq dataset across different time points. In this dataset, an inducible rescue cell line was used, where EZH1-HA expression was driven by a Dox-inducible promoter. This dataset is referred to as EZH1-HA ChIP-seq.

However, due to the relatively low number of peaks detected in the EZH1-HA ChIP-seq datasets, we suspected that this might be due to different antibody efficiency. Therefore, we decided to utilize only the EZH1-BIO ChIP-seq dataset for further integration with the gene expression data.

To ascertain whether Bmal1, Per1, and Per2 are direct targets of EZH1, we conducted a comparison of the visualized peak files using the EZH1-HA, H3K27me3, and EZH1-Bio Bigwig files. As depicted in **Figure 3**, we observed a clear enrichment of EZH1-HA around the transcription start sites (TSS) of the Per1 and Per2 loci, along with a milder enrichment around the Bmal1 locus. This indicates that EZH1 can directly bind to these three clock genes, with the variation in enrichment profiles likely attributed to differences in immunoprecipitation efficiency, such as the efficiency of the HA antibody or streptavidin beads, may contribute to disparities in enrichment. Thus, we believe that the discrepancy observed between the ChIP-qPCR and ChIP-seq strategies is primarily a result of technical factors rather than biological issues. Further, we looked at other clock genes, such as Npas2, Cry2, and Nr1d2, and found EZH1 also bound to those loci (**Figure 4**). To validate the specificity of both EZH1 ChIP-seq datasets, we additionally examined Hoxc13 and Neurog1 as canonical PcG repressive targets. Notably, we observed robust binding of EZH1 and H3K27me3 at these loci (**Figure 5**). Interestingly, the EZH1-Bio ChIP-seq dataset displayed a higher enrichment signal compared to the EZH1-HA ChIP-seq dataset around these two loci. This finding further supports the notion that differential enrichment profiles can arise due to technical variations in ChIP efficiency.

Figure 3. IGV track of EZH1-HA, H3K27me3 and EZH1-BIO ChIP-seq signals around *Bmal1*, *Per1* and *Per2* loci.

Figure 4. IGV track of EZH1-HA, H3K27me3 and EZH1-BIO ChIP-seq signals around *Cry2*, *Npas2* and *Nr1d2* loci.

Figure 5. IGV track of EZH1-HA, H3K27me3 and EZH1-BIO ChIP-seq signals around Hoxc13 and Neurog1 loci.

References:

1. Bodega, B. et al. A cytosolic Ezh1 isoform modulates a PRC2-Ezh1 epigenetic adaptive response in postmitotic cells. *Nature Structural & Molecular Biology* **24**, 444+ (2017).
2. Liu, P., Shuaib, M., Zhang, H.M., Nadeef, S. & Orlando, V. Ubiquitin ligases HUWE1 and NEDD4 cooperatively control signal-dependent PRC2-Ezh1 alpha/beta-mediated adaptive stress response pathway in skeletal muscle cells. *Epigenetics & Chromatin* **12**(2019).
3. Marasca, F., Bodega, B. & Orlando, V. How Polycomb-Mediated Cell Memory Deals With a Changing Environment: Variations in PcG complexes and proteins assortment convey plasticity to epigenetic regulation as a response to environment. *Bioessays* **40**(2018).

4. Blackledge, N.P. & Klose, R.J. The molecular principles of gene regulation by Polycomb repressive complexes. *Nat Rev Mol Cell Biol* **22**, 815-833 (2021).
5. Aguilar-Arnal, L. & Sassone-Corsi, P. Chromatin landscape and circadian dynamics: Spatial and temporal organization of clock transcription. *Proceedings of the National Academy of Sciences of the United States of America* **112**, 6863-6870 (2015).
6. Takahashi, J.S. Transcriptional architecture of the mammalian circadian clock. *Nat Rev Genet* **18**, 164-179 (2017).
7. Koike, N. et al. Transcriptional Architecture and Chromatin Landscape of the Core Circadian Clock in Mammals. *Science* **338**, 349-354 (2012).
8. Papazyan, R., Zhang, Y.X. & Lazar, M.A. Genetic and epigenomic mechanisms of mammalian circadian transcription. *Nature Structural & Molecular Biology* **23**, 1045-1052 (2016).
9. Mousavi, K., Zare, H., Wang, A.H. & Sartorelli, V. Polycomb protein Ezh1 promotes RNA polymerase II elongation. *Mol Cell* **45**, 255-62 (2012).
10. Stojic, L. et al. Chromatin regulated interchange between polycomb repressive complex 2 (PRC2)-Ezh2 and PRC2-Ezh1 complexes controls myogenin activation in skeletal muscle cells. *Epigenetics & Chromatin* **4**(2011).
11. Xu, J. et al. Developmental Control of Polycomb Subunit Composition by GATA Factors Mediates a Switch to Non-Canonical Functions. *Molecular Cell* **57**, 304-316 (2015).
12. Su, S.K. et al. The EZH1-SUZ12 complex positively regulates the transcription of NF-kappaB target genes through interaction with UXT. *J Cell Sci* **129**, 2343-53 (2016).
13. Chen, J.H. et al. Two faces of bivalent domain regulate VEGFA responsiveness and angiogenesis. *Cell Death & Disease* **11**(2020).
14. Henriquez, B. et al. Ezh1 and Ezh2 differentially regulate PSD-95 gene transcription in developing hippocampal neurons. *Mol Cell Neurosci* **57**, 130-43 (2013).
15. Feng, X. et al. Polycomb Ezh1 maintains murine muscle stem cell quiescence through non-canonical regulation of Notch signaling. *Dev Cell* **58**, 1052-1070 e10 (2023).
16. Zovoilis, A., Cifuentes-Rojas, C., Chu, H.P., Hernandez, A.J. & Lee, J.T. Destabilization of B2 RNA by EZH2 Activates the Stress Response. *Cell* **167**, 1788-1802 e13 (2016).
17. Wang, J. et al. EZH2 noncanonically binds cMyc and p300 through a cryptic transactivation domain to mediate gene activation and promote oncogenesis. *Nature Cell Biology* **24**, 384-+ (2022).
18. Kim, J. et al. Polycomb- and Methylation-Independent Roles of EZH2 as a Transcription Activator. *Cell Reports* **25**, 2808-+ (2018).

Dear Dr. Orlando,

Thank you for submitting a revised version of your manuscript which was previously rejected after review by a different Journal. Your study has now been seen by an arbitrating expert, who finds that the concerns raised by the original referees have been addressed and recommends publication of the manuscript. There remain only a few mainly editorial points that have to be addressed before I can extend formal acceptance of the manuscript:

- Please include the "Funding" section in the "Acknowledgements"
- On the abstract page of the manuscript, please include 4-5 general keyword terms to enhance searchability.
- Please rename the "Data and materials availability" section to "data availability".
- Please rename the "conflict of interest" section to "DISCLOSURE AND COMPETING INTERESTS STATEMENT"
- CRediT has replaced the traditional author contributions section because it offers a systematic machine readable author contributions format that allows for more effective research assessment. Please remove the Author Contributions section from the manuscript and use the free text boxes beneath each contributing author's name in our online systems to add specific details on the author's contribution. More information is available in our guide to authors.
- Please adjust the in-text callouts for individual figures and figure panels: e.g. there are callouts for Supplemental Tables S2-S7, but no tables uploaded.
- Please convert the appendix file into PDF format; The nomenclature should be Appendix Figure S1-S16 with the appropriate callouts, and please also add page numbers to the table of content on the title page.
- Please provide the Reagent and Tools Table. For more information, please check <https://www.embopress.org/page/journal/14602075/authorguide#structuredmethods> and download the template for Reagent Table (attached for your convenience)
- Please make sure to provide all the requested Source data files listed in the uploaded and attached Source Data checklist file. Please complete the Source Data checklist and upload it to our online system. Source data files need to be saved in a scheme one figure/folder and then uploaded as .zip files. E.g. all the Source data files for figure 1 need to be saved in a single folder and this needs to be zipped and then uploaded as "SD figure 1.zip" file.
- Please provide suggestions for a short 'blurb' text prefacing and summing up the conceptual aspect of the study in two sentences (max. 250 characters), followed by 3-5 one-sentence 'bullet points' with brief factual statements of key results of the paper; they will form the basis of an editor-written 'Synopsis' accompanying the online version of the article. Please also provide an altered synopsis image, making sure that the aspect ratio conforms to our website's format - it should be exactly 550 pixels wide and between 300-600 pixels high.
- Please adjust the order of the manuscript sections: Title page with complete author information, Abstract, Keywords, Introduction, Results, Discussion, Methods, Data Availability Section, Acknowledgements, Disclosure and Competing Interests Statement, References, Main figure legends, Tables, Expanded Figure Legends.

In addition, our data editors have raised several points regarding the figures and legends:

- Figure Legends (main + EV): 1. Please define the annotated p values ****/*** as well as provide the exact p-values for the same in the legend of figure 4f; as appropriate.
- 2. Please note that the exact p values are not provided in the legends of figures 2a; 3a-b; 4e; 6g-h; EV 2f-g; EV 4a, c-d; EV 5a-c.
- 3. Please indicate the statistical test used for data analysis in the legends of figures 3g-h; 5d; EV 9c-d; EV 10f; EV 15g-h.
- 4. Please note that in figures 6g-h; EV 5a-c; there is a mismatch between the annotated p values in the figure legend and the annotated p values in the figure file that should be corrected.
- 5. Please note that for the figures EV 2c-e, p-values and statistical tests are indicated in the legends. However, comparison for the same, """" has not been represented in the figures. Please rectify this in the figures or legends as applicable.

6. Please note that the box plots need to be defined in terms of minima, maxima, centre, bounds of box and whiskers, and percentile in the legends of figures 5b; EV 10e; EV 13b-c; EV 16c-d, g-h, k-l.
7. Please note that information related to n is missing in the legends of figures 5b; EV 10e; EV 13b-c.
8. Although 'n' is provided, please describe the nature of entity for 'n' in the legends of figures 3b-c; 4b.
9. Please note that the error bars are not defined in the legends of figures 3b-c; b.

With best regards,

Cornelius Schneider

Cornelius Schneider, PhD
Editor | The EMBO Journal
c.schneider@embojournal.org

We realize that it is difficult to revise to a specific deadline. In the interest of protecting the conceptual advance provided by the work, we recommend a revision within 3 months (22nd Oct 2024). Please discuss the revision progress ahead of this time with the editor if you require more time to complete the revisions. Use the link below to submit your revision:

Referee #1:

The manuscript describes a fluctuation in PRC2-EZH1 levels that is directly regulated by BMAL1, and shows an oscillatory circadian behavior. This is definitely novel. Moreover they show that EZH1 becomes essential for circadian gene expression, through stabilization of RNA Pol II preinitiation complexes. This is also very novel. The earlier reviews were unconvinced by PRC2 CHIP data. It is known that Pc complexes chip poorly to DNA in vivo, thus one cannot expect highly convincing enrichment values. I find the rebuttal of the authors to be convincing and given the novelty of the findings, I think the paper should be accepted for publication. Overall they have used a wide range of omics and chromatin monitoring methods to substantiate their claim and thus the weakness of any one method is compensated by the others. I do not think that further revision will change the paper substantially and I would recommend acceptance.

All editorial and formatting issues were resolved by the authors.

Dear Prof. Orlando,

I am pleased to inform you that your manuscript has been accepted for publication in the EMBO Journal.

Yours sincerely,

Cornelius Schneider, PhD
Editor
The EMBO Journal
c.schneider@embojournal.org
